# DNA2 enables growth by restricting recombination-restarted replication

Jessica J. R. Hudson[1,4], Rowin Appanah[1,4], David Jones[1,2], Kathryn Davidson[1], Alice M. Budden[1], Alina Vaitsiankova[1,3], Kok-Lung Chan[1], Keith W. Caldecott[1], Antony M. Carr[1] & Ulrich Rass[1✉]

Nuclease–helicase DNA2 is a multifunctional genome caretaker that is essential for cell proliferation in a range of organisms, from yeast to human[1–4]. Bi-allelic *DNA2* mutations that reduce DNA2 concentrations cause a spectrum of primordial dwarfism disorders, including Seckel and Rothmund–Thomson-related syndromes[5–7]. By contrast, cancer cells frequently express high concentrations of DNA2 (refs. 8–11). The mechanism that precludes cell proliferation in the absence of DNA2 and the molecular aetiology of *DNA2*-linked diseases remain elusive. Here we used yeast and human cells to demonstrate that *DNA2* suppresses homologous recombination-restarted replication and checkpoint activation at stalled DNA replication forks. Loss of this control mechanism upon degradation of DNA2 in human cells causes recombination-dependent DNA synthesis and build-up of RPA-bound single-stranded DNA in the G2 phase of the cell cycle. Consequently, DNA2 deprivation triggers the DNA damage checkpoint and invariably leads to ATR–p21-dependent cell-cycle exit before mitosis. These findings explain why *DNA2* is essential for cell proliferation and reveal that replication fork processing to restrict recombination is indispensable for avoiding cellular senescence. Stochastic entry into senescence stifles the proliferative potential of cells following the expression of a Seckel syndrome patient-derived DNA2 hypomorph or partial degradation of DNA2, providing a conceptual framework to explain global growth failure in *DNA2*-linked primordial dwarfism disorders.

*DNA2* is implicated in DNA repair and replication[12]. A prevailing model posits that a role in Okazaki fragment processing during lagging-strand replication renders *DNA2* essential for cell proliferation[13]. This is consistent with the ability of DNA2 to cleave 5′-flaps bound by the single-stranded DNA (ssDNA)-binding protein RPA, which makes them refractory to the Okazaki fragment processing nuclease FEN1 (ref. 14). Thus, RPA-bound flaps might accumulate in the absence of DNA2, leading to DNA damage checkpoint activation[15], and explain the lethal G2/M checkpoint arrest that occurs in yeast upon loss of *DNA2* (ref. 16). However, whether toxic Okazaki fragment-linked flaps exist in Dna2-defective yeast cells remains uncertain, and evidence for an Okazaki fragment processing role for human DNA2 has not been found[12,17].

An alternative hypothesis[18] suggests that a function in mediating the recovery of stalled DNA replication forks (RFs)[2,19–22] may explain the essential requirement for *DNA2*. Human DNA2 degrades nascent DNA at stalled RFs, which correlates with replication restart[19] or pathological hyperresection if fork protection factors, such as FANCD2 (ref. 23), BRCA2 (ref. 24) and BOD1L[25], are compromised. By contrast, depletion of DNA2 in yeast and human cells results in increased RF reversal, in which the nascent leading and lagging strands dissociate and anneal to each other to generate a four-way DNA intermediate known as a chicken-foot structure[2,19,21]. This indicates a conserved mechanism whereby the controlled processing of stalled RFs by DNA2 counteracts

RF reversal and promotes fork reactivation. Persistent chicken-foot structures can give rise to homologous recombination-restarted replication (HoRReR), which provides a pathway to recover stalled RFs at which conventional replication cannot be reactivated[26–29]. HoRReR uses the DNA end at the small toe of a chicken-foot structure as a substrate for RAD51-dependent strand invasion into the parental template, in which a displacement loop (D-loop) is formed[30]. Within the D-loop, the invading 3′-end reinitiates templated DNA synthesis while displacing the non-complementary strand as ssDNA[29]. D-loops can give rise to substantial DNA synthesis tracts, thereby contributing to replication completion. However, D-loop DNA synthesis is error-prone and unstable, which can lead to template switching, ectopic recombination and genomic instability[31]. Whether HoRReR is elevated when *DNA2* function is compromised, and whether this is related to toxic DNA damage checkpoint activation in *DNA2*-mutated cells, remains unknown.

Here we demonstrate that the actions of Dna2 suppress HoRReR at a site-specific replication barrier in fission yeast. In human cells, we found that DNA2 deprivation results in recombination-dependent DNA synthesis at persistent replication intermediates, an accumulation of RPA-bound ssDNA in G2, sustained activation of the DNA damage response and, invariably, ATR-dependent withdrawal from the cell cycle before mitosis. These results answer the long-standing question of why DNA2 is indispensable for eukaryotic cell proliferation and have

[1]Genome Damage and Stability Centre, School of Life Sciences, University of Sussex, Brighton, UK. [2]Present address: Centre de Recherche en Cancérologie de Marseille, U1068 Inserm, UMR7258 CNRS, Institut Paoli-Calmettes, Aix Marseille Université, Marseille, France. [3]Present address: Department of Genetics and Development, Institute for Cancer Genetics, Herbert Irving Comprehensive Cancer Center, Columbia University Irving Medical Center, New York, NY, USA. [4]These authors contributed equally: Jessica J. R. Hudson, Rowin Appanah. ✉e-mail: U.W.Rass@sussex.ac.uk

important implications for the use of DNA2 inhibitors as anti-cancer agents, as well as providing a molecular model for *DNA2*-linked primordial dwarfism.

## DNA2 suppresses HoRReR

The *RTS1* replication barrier in the fission yeast *Schizosaccharomyces pombe* provides an opportunity to observe HoRReR at stalled RFs[26,27,29]. Using a reporter strain harbouring direct repeat *ade6⁻* heteroalleles near *RTS1*, HoRReR can be measured by quantifying *ade⁺* recombinants arising by template switching[26] (Extended Data Fig. 1a). Given that *dna2⁺* is essential[2], we measured HoRReR in cells harbouring Dna2 variant R1132Q (Extended Data Fig. 1b), which is equivalent to the well-characterized ATPase/helicase-defective[20] Dna2 hypomorph R1253Q (*dna2-2*) in *Saccharomyces cerevisiae*. The frequency of *ade⁺* recombinants increased by an order of magnitude in *dna2-2* cells compared with *dna2⁺* cells (Extended Data Fig. 1c). Plasmid-based expression of wild-type Dna2 reduced direct repeat recombination in *dna2⁺* and *dna2-2* cells (Extended Data Fig. 1d).

Across organisms, recombination-dependent DNA synthesis requires DNA polymerase-δ (Polδ) subunit POLD3 (Cdc27 and Pol32 in fission and budding yeast, respectively)[32–34]. *S. pombe* cells harbouring the truncated *cdc27-D1* version[35] of the essential *cdc27⁺* gene are strongly defective for break-induced replication[34], a mechanism[36] to recover broken RFs by recombination-dependent DNA synthesis. We determined that HoRReR-mediated DNA synthesis (where Polδ synthesizes the Watson and Crick strands)[37] at *RTS1* was markedly reduced in *cdc27-D1* cells (Extended Data Fig. 2). The expression of *Cdc27-D1* (Extended Data Fig. 3a) largely abolished the formation of *ade⁺* recombinants at *RTS1* (Extended Data Fig. 3b). Combining *dna2-2* with *cdc27-D1* suppressed the dramatic increase in *RTS1*-dependent *ade⁺* recombinants associated with Dna2 dysfunction, confirming that HoRReR underpins excessive *ade⁺* recombinants in Dna2-defective cells (Extended Data Fig. 3b). Moreover, although *dna2-2* and *cdc27-D1* are each associated with chronic checkpoint activation leading to cell elongation compared with the wild type, the introduction of the *dna2-2* allele into the *cdc27-D1* background did not result in a further increase in cell length beyond the baseline provided by *cdc27-D1* cells (Extended Data Fig. 3c). Finally, *cdc27-D1* restored the diminished cell viability of *dna2-2* cells (Extended Data Fig. 3d). Similar results were obtained for interactions between Dna2 and Pfh1, a helicase required for efficient D-loop progression[36]. Thus, *pfh1-mt**, encoding a nuclear-excluded version[38] of Pfh1, suppressed excessive recombination at *RTS1*, checkpoint activation and poor viability in *dna2-2* cells (Extended Data Fig. 3a–d).

Taken together, these findings provide direct evidence that *DNA2* restricts recombination and D-loop DNA synthesis at stalled RFs, and that this function of *DNA2* is required to avoid checkpoint-mediated cell-cycle arrest.

## ATR blocks mitosis upon loss of DNA2

Although previous studies[4,8,17,19,39] on human *DNA2* using RNA interference-mediated gene silencing or conditional knockout in cancer cell lines described DNA damage and rampant genomic instability in the absence of DNA2, a clear view of the molecular events that underpin the strict requirement for *DNA2* in proliferating cells has not yet emerged. Here we have taken a rapid protein degradation approach to study the role of DNA2 in non-transformed human cells. We introduced coding sequences for a mini auxin-inducible degron (mAID)[40] and a small-molecule-assisted shutoff (SMASh) tag[41] into human epithelial RPE-1 cells to replace the stop codon at each genomic copy of *DNA2* (Extended Data Fig. 4a). Cells were additionally modified for doxycycline-induced expression of *Oryza sativa TIR1* (*OsTIR1*) to enable controlled target protein degradation through mAID[42]. The resulting double-degron-tagged DNA2 protein (DNA2[dd]) can be degraded by adding doxycycline in combination with the auxin indole-3-acetic acid (IAA) and/or asunaprevir (Asv) to activate SMASh-tag-mediated protein degradation. Upon adding a doxycycline/IAA/Asv (DIA) drug cocktail to RPE-1 *OsTIR1 DNA2[dd]* cells, DNA2[dd] was no longer detectable within 4 h (Fig. 1a).

The addition of DIA to RPE-1 *OsTIR1 DNA2[dd]* cells abolished colony formation in a clonogenic cell survival assay (Fig. 1b) and blocked cell proliferation (Fig. 1c). To validate that the observed growth defects were causally linked to loss of DNA2, we complemented RPE-1 *OsTIR1 DNA2[dd]* cells with wild-type DNA2 complementary DNA (cDNA), which restored cell proliferation and colony formation after DNA2[dd] degradation (Extended Data Fig. 4b,c).

Analysis by flow cytometry showed that RPE-1 *OsTIR1 DNA2[dd]* cells undergo bulk DNA synthesis and traverse S phase after the addition of DIA but accumulate with a 4N DNA content over a period of 24 h (Fig. 1d). ATR inhibitor (ATRi) VE-821, but not ATM inhibitor (ATMi) KU55933, alleviated the block to cell division in a dose-dependent manner, indicating the involvement of the apical checkpoint kinase ATR in the failure of DIA-treated RPE-1 *OsTIR1 DNA2[dd]* cells to progress through mitosis (Fig. 1e). By live-cell imaging, we observed that newborn daughter cells with degraded DNA2[dd] failed to undergo even one subsequent mitosis; this invariable block to cell division could be suppressed by the addition of ATRi, allowing cells to progress through mitosis and form live daughter cells (Fig. 1f and Extended Data Fig. 4d,e).

Thus, DNA2 is dispensable for bulk DNA replication but essential to avert an immediate ATR-dependent halt to cell-cycle progression in the space of a single cell cycle in non-transformed human cells.

## DNA2 averts RF persistence

To better understand the consequences of DNA replication without DNA2, we quantified the genotoxic stress marker γH2AX (histone H2AX S139 phosphorylation) in RPE-1 *OsTIR1* control cells and *DNA2[dd]* cells. The DIA treatment had no effect in G1 or S phase, but γH2AX intensity increased significantly in late S/G2 phase in RPE-1 *OsTIR1 DNA2[dd]* cells (Extended Data Fig. 5a). Next, we degraded DNA2, allowed RPE-1 *OsTIR1 DNA2[dd]* to transition through the S phase and quantified γH2AX foci 2 h, 4 h and 8 h after cells had traversed the late S phase (cells selected on the basis of EdU incorporation patterns; Methods) (Fig. 2a). In the presence of DNA2, cells exhibited a moderate number of γH2AX foci (median: 5) per cell at the 2-h time point. These foci quickly diminished and had disappeared at the 8-h time point, in which cells had progressed into the G1 phase of the next cell cycle. Upon DNA2 degradation, the cells showed a marked increase in γH2AX foci (median: 32) 2 h after the late S phase. Over time, γH2AX foci per cell decreased, but DNA2-deprived cells still exhibited elevated foci numbers (median: 13) 8 h after completing the S phase (Fig. 2b). DNA2-deprived cells remained in the G2 phase of the cell cycle and failed to undergo mitosis. Thus, although the loss of DNA2 does not result in obvious replication stress or DNA damage phenotypes during the S phase, it causes an accumulation of DNA intermediates decorated by γH2AX in the late S/G2 phase of the cell cycle. This phenotype was highly penetrant across the asynchronous population of DNA2-deprived cells (Extended Data Fig. 5b).

PARP1 acts as a sensor of unligated Okazaki fragments, and S-phase poly(ADP-ribose) (PAR) is strongly elevated upon inhibition of the canonical Okazaki fragment processing nuclease FEN1 (refs. 43,44). To address whether γH2AX foci in DNA2-deprived cells might be linked to perturbed Okazaki fragment maturation[13,16,21], we analysed PAR concentrations in DIA-treated RPE-1 *OsTIR1 DNA2[dd]* cells. The addition of DIA did not increase PAR signals, nor did it synergize with FEN1 inhibitor treatment to elevate PAR concentrations above those detected upon the inhibition of FEN1 (Extended Data Fig. 6). This is in line with previous reports of proficient Okazaki fragment processing in DNA2-depleted human cells[17].

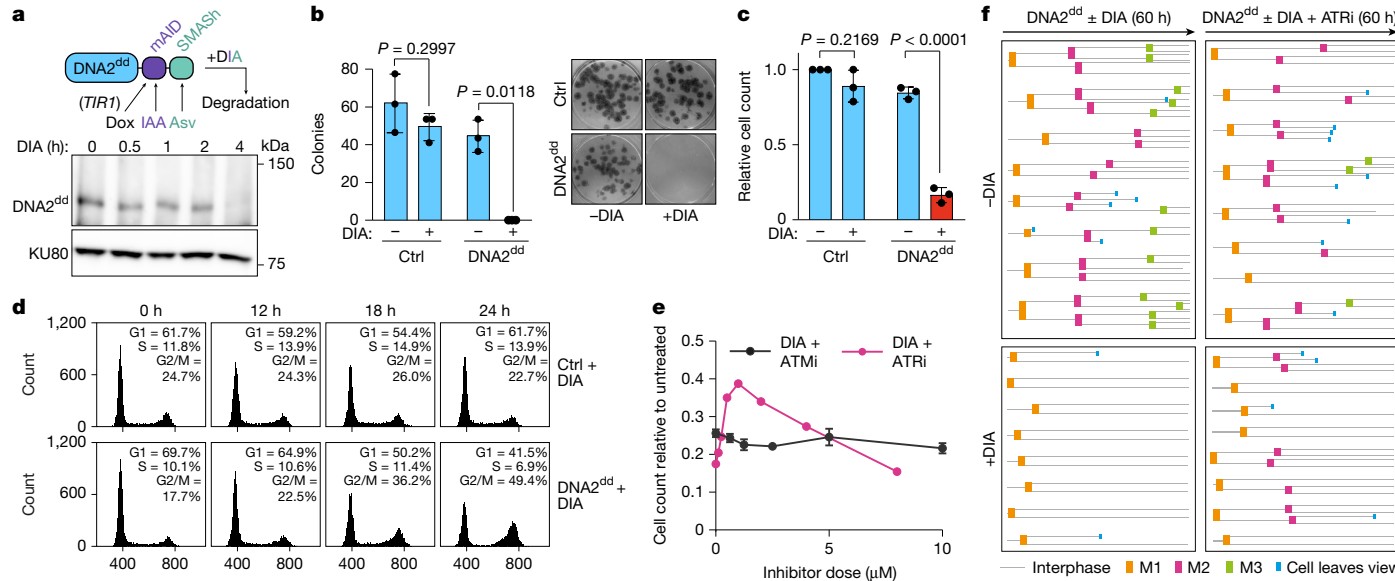

**Fig. 1 | Degradation of human DNA2 results in immediate ATR-dependent cessation of cell proliferation. a**, Top, schematic of DNA2[dd] with mAID and SMASh tags for controlled degradation. Bottom, western blot analysis of cell lysates subjected to DNA2 immunoprecipitation showing loss of DNA2[dd] in RPE-1 *OsTIR1 DNA2[dd]* cells after DIA addition over a period of 4 h (representative of two independent experiments). KU80, input control. **b**, Colony-forming ability of RPE-1 *OsTIR1* (Ctrl) and RPE-1 *OsTIR1 DNA2[dd]* cells after plating 100 cells in the presence or absence of DIA (*n* = 3 independent experiments). Data are presented as mean ± s.d. and significance was determined by means of two-sided *t*-test with Welch's correction. **c**, Cell proliferation in the presence or absence of DIA over 3 days; 3,500 cells were plated (*n* = 3 independent experiments). Data are presented as mean ± s.d. and significance was determined by means of two-sided *t*-test with Welch's correction after normalization to untreated RPE-1 *OsTIR1* (Ctrl) cells. **d**, Flow cytometric analysis of the indicated cell lines following growth in DIA-containing medium for the indicated times (representative of three independent experiments). a.u., arbitrary units. **e**, Cell proliferation assay of 1,500 RPE-1 *OsTIR1 DNA2[dd]* cells plated and treated with DIA in combination with ATMi KU55933 or ATRi VE-821 (*n* = 3 technical replicates). Data are presented as mean ± s.d. **f**, Live-cell imaging over 60 h of RPE-1 *OsTIR1 DNA2[dd]* cells treated or not with DIA and ATRi (1 μM), as indicated. For each condition, 8 individual cells that underwent mitosis (M1) within the first 12 h were monitored for subsequent mitoses (M2 and M3).

To test the possibility that DIA-induced γH2AX foci in RPE-1 *OsTIR1 DNA2[dd]* cells relate to unresolved RFs[45], we probed for FANCD2, which binds stalled RFs, mediates RF protection and marks persistent replication intermediates through the G2 phase and into mitosis[46]. Untreated RPE-1 *OsTIR1 DNA2[dd]* cells showed a moderate number of FANCD2 foci (median: 7.5), which resolved quickly over time as cells completed DNA replication and underwent mitosis. After DIA treatment, cells contained more FANCD2 foci in G2 (median: 29), and many foci (median: 11) persisted 8 h after cells traversed the late S phase (Fig. 2c). Co-localization analysis showed almost complete congruence of γH2AX and FANCD2 foci in RPE-1 *OsTIR1 DNA2[dd]* cells both in the presence and absence of DIA (Fig. 2d).

Together, these data indicate that the actions of DNA2 are dispensable for Okazaki fragment processing but are strictly required to avoid an accumulation of stalled replication intermediates in the late S/G2 phase of the cell cycle.

## DNA2 prevents HoRReR-mediated RPA–ssDNA

To address whether persistent replication intermediates in DIA-treated RPE-1 *OsTIR1 DNA2[dd]* cells undergo HoRReR, we first probed for RPA to detect ssDNA. In the absence of DIA treatment, very few cells showed any observable RPA signals in G2. By stark contrast, the DIA-treated cells exhibited numerous RPA foci (median: 25) 2 h after the S phase. Although the median number of RPA foci per cell subsequently decreased, mirroring the trends we observed for γH2AX and FANCD2 foci, a subset of RPA foci grew into very bright, large RPA foci (with a signal intensity ten times or more than that of the dimmest detectable RPA foci; hereafter referred to as RPA bodies), which were observed 8 h after the S phase and beyond (Fig. 3a and Extended Data Fig. 7a). Co-localization analysis showed a very strong overlap of RPA and

FANCD2 foci in DNA2-deprived cells (Extended Data Fig. 7b). Next, we induced mild replication stress[47] by aphidicolin (APH) treatment in RPE-1 *OsTIR1 DNA2[dd]* cells to increase RF stalling. Co-treatment with DIA and APH significantly induced RPA focus formation (Fig. 3b). Finally, we treated RPE-1 *OsTIR1 DNA2[dd]* cells with DIA at varying time points before the late S phase and observed that the number of FANCD2 and RPA foci persisting 8 h after the late S phase correlated with the proportion of the S phase that cells had to traverse without DNA2. Even if DIA was added only 2 h before the late S phase, cells exhibited accumulations of RPA–ssDNA, and approximately 30% of cells failed to progress to the next G1 phase (Extended Data Fig. 8). Thus, DNA2 is required throughout the S phase to avoid a build-up of stalled replication intermediates accumulating RPA–ssDNA, a trigger[15] for ATR-dependent DNA damage response signalling, in the G2 phase of the cell cycle.

HoRReR initiates at reversed RF intermediates[30]. Two main fork reversal pathways[48] in human cells are defined by SNF2 translocases, including SMARCAL1, HLTF and ZRANB3, and by the UvrD helicase FBH1. Depletion of FBH1, but not SNF2 translocases, significantly reduced RPA foci in DIA-treated RPE-1 *OsTIR1 DNA2[dd]* cells (Fig. 3c,d). A subset of DNA2-deprived cells in which FBH1 depletion had apparently fully suppressed persistent RPA–ssDNA accumulations was able to undergo mitosis (Fig. 3d). HoRReR further depends on the strand exchange activity of recombination factor RAD51. We found that RAD51 marked most RPA foci in DNA2-deprived cells (Fig. 3e). To directly probe for recombination-dependent DNA synthesis, we analysed DIA-treated RPE-1 *OsTIR1 DNA2[dd]* cells for EdU incorporation in G2. EdU incorporation was detected at a small number of discrete sites, which coincided with DIA-induced RPA foci (Fig. 3f). Over time, the intensity of EdU foci at RPA–ssDNA sites increased gradually, revealing prolonged G2 DNA synthesis at unresolved replication intermediates in DNA2-deprived cells. Suppressing RAD51 activity with inhibitor[49] B02 significantly

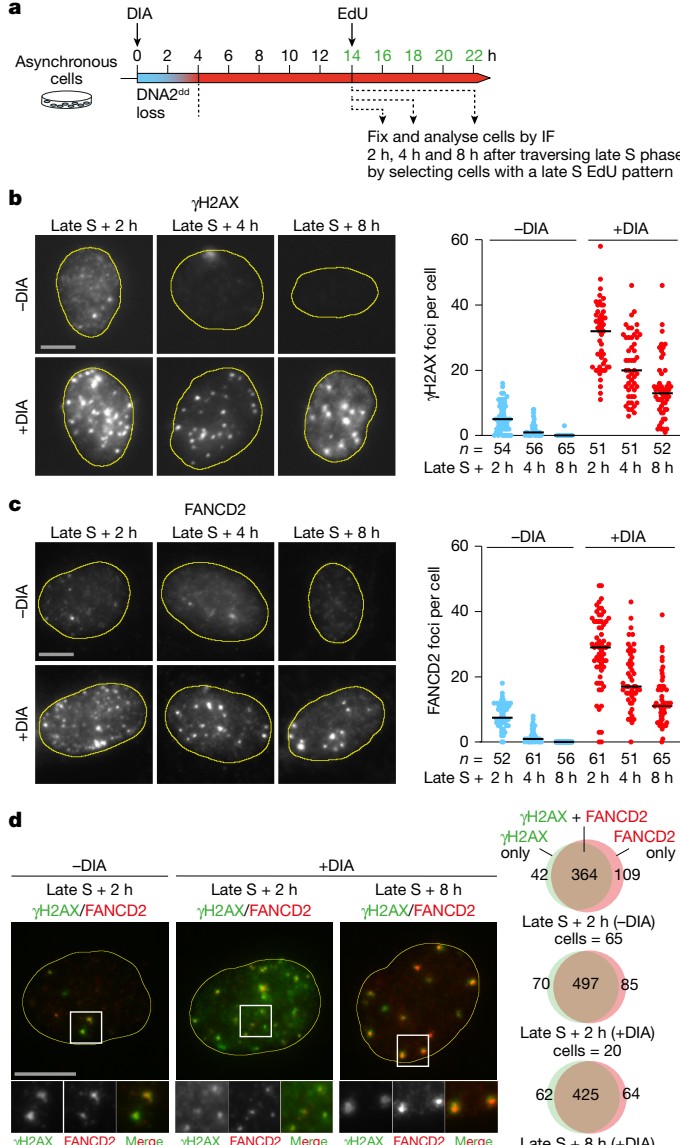

**Fig. 2 | Persistent replication intermediates after degradation of DNA2[dd].**
**a**, Experimental set-up. DIA was added to asynchronous RPE-1 *OsTIR1 DNA2[dd]* cells, inducing DNA2[dd] loss over a period of 4 h. At the 14-h time point, EdU was added and cells with a late S/G2 EdU staining pattern were selected for immunofluorescence (IF) analyses of relevant marker proteins at the 16-h, 18-h and 22-h time points (2 h, 4 h and 8 h after traversing late S phase). **b**, Representative images of cells 2 h, 4 h and 8 h after late S phase. Note that all cells are in G2, except those not treated with DIA, which have progressed to G1 at the 8-h time point. Quantification of γH2AX foci was performed for the indicated number of cells (*n*) from two independent experiments. Bars, median. **c**, As described in **b**, but cells were analysed for FANCD2. **d**, Images (representative of three independent experiments) of γH2AX and FANCD2 co-localization and quantification of foci across the indicated number of cells. Scale bars, 10 μm.

reduced the intensity of RPA and EdU foci in DIA-treated RPE-1 *OsTIR1 DNA2[dd]* cells (Fig. 3g). By analysing DNA2-deprived cells in G2 for BrdU incorporation under non-denaturing conditions, we found that RPA foci contain nascent ssDNA. Nascent ssDNA marked by BrdU at RPA foci was significantly reduced in the presence of B02 (Fig. 3g). Finally, DNA synthesis at RPA foci in DNA2-deprived cells was strongly suppressed by short interfering RNA (siRNA)-mediated depletion of Polδ subunit POLD3 (Fig. 3h). These results are consistent with homologous recombination-dependent DNA synthesis occurring in the context of

a migrating D-loop with the spooled-out nascent ssDNA strand bound by RPA in its wake.

Overall, the data indicate a shift in RF recovery towards HoRReR when DNA2-dependent fork processing and reactivation[19] are not available.

## DNA2 loss elicits direct cell-cycle exit

Exogenous DNA damage shortly before mitosis can result in ATR-dependent cell-cycle exit from antephase[50,51]. We reasoned that RAD51-dependent RPA–ssDNA accumulation in DNA2-deprived cells might similarly trigger cell-cycle exit from G2 in the absence of exogenous DNA damage. To monitor ATR activation in RPE-1 *OsTIR1 DNA2[dd]* cells upon DIA treatment, we scrutinized CHK1, a target of ATR kinase activity. CHK1 phosphorylation was very prominent 12 h after the addition of DIA, before CHK1 protein was gradually lost (Fig. 4a), a phenotype previously observed[52] upon G2 cell-cycle exit following exogenous DNA damage. CHK1 phosphorylation coincided with a significant and durable accumulation of p21 (Fig. 4a), which can trap cyclin B1 in its inactive form within the nucleus, ultimately leading to its degradation and making cell-cycle withdrawal and senescence irreversible[53]. Consistently, we observed an induction of cyclin B1 translocation from the cytoplasm to the nucleus in DIA-treated RPE-1 *OsTIR1 DNA2[dd]* cells (Extended Data Fig. 9a) before an ATR-dependent enlargement of cell nuclei, a hallmark feature of cellular senescence[50,52] (Fig. 4b). Moreover, DIA-treated RPE-1 *OsTIR1 DNA2[dd]* cells stained for increasing concentrations of β-galactosidase and showed an enlarged cell morphology over a period of 14 days, confirming that DNA2-deprived cells exiting the cell cycle from G2 enter a senescent state (Fig. 4c). Similar to ATR inhibition, siRNA-mediated depletion of p21 alleviated senescence and mitotic block experienced by DNA2-deprived cells (Extended Data Fig. 9b,c). Notably, the cell-cycle exit override in DIA-treated RPE-1 *OsTIR1 DNA2[dd]* cells by p21 depletion was accompanied by the formation of micronuclei, indicating that chromosomal replication is not faithfully completed in the absence of DNA2 if cells are afforded unbridled progression to mitosis (Extended Data Fig. 9d).

By probing DNA2-deprived cells with DIA-induced RPA foci for cyclin B1, we observed that RPA focus formation in G2 correlated first with cyclin B1 translocation from the cytoplasm to the nucleus, and then with cyclin B1 loss in the majority of RPA-positive cells 24 h after DIA addition (Fig. 4d). Only a small proportion of cells (approximately 5% after 18 h of DIA addition) showed DNA double-strand break-specific phosphorylation of RPA32 on S4 and S8 (pRPA)[54], and this always coincided with nuclear or absent cyclin B1 (Extended Data Fig. 9e). This indicates that break-induced replication[36], which mediates recombination-dependent DNA synthesis after fork breakage in interphase or mitosis (known as mitotic DNA synthesis)[55] is not significantly involved, further confirming that RAD51–POLD3-dependent RPA–ssDNA formation and commitment to cell-cycle exit arise from stalled, unbroken, replication intermediates along the HoRReR pathway.

The expression of wild-type DNA2, but not of nuclease-deficient (D277A), helicase-deficient (K654E) and double-deficient (D277A/K645E) versions[56] of DNA2, suppressed DNA2-loss phenotypes in DIA-treated RPE-1 *OsTIR1 DNA2[dd]* cells (Extended Data Fig. 10), indicating that the coordinated action of the nuclease and helicase activities is required for DNA2 to suppress HoRReR.

We conclude that DNA2-dependent fork processing is required to suppress the accumulation of stalled replication intermediates, HoRReR and cell-cycle exit, providing a new explanation for why DNA2 is essential for cell proliferation (Fig. 4e).

## Stochastic cell-cycle exit with low DNA2

Case reports indicate low concentrations of wild-type DNA2 in *DNA2*-linked microcephalic primordial dwarfism, Seckel syndrome and Rothmund–Thomson-related syndrome, and senescent cells have been

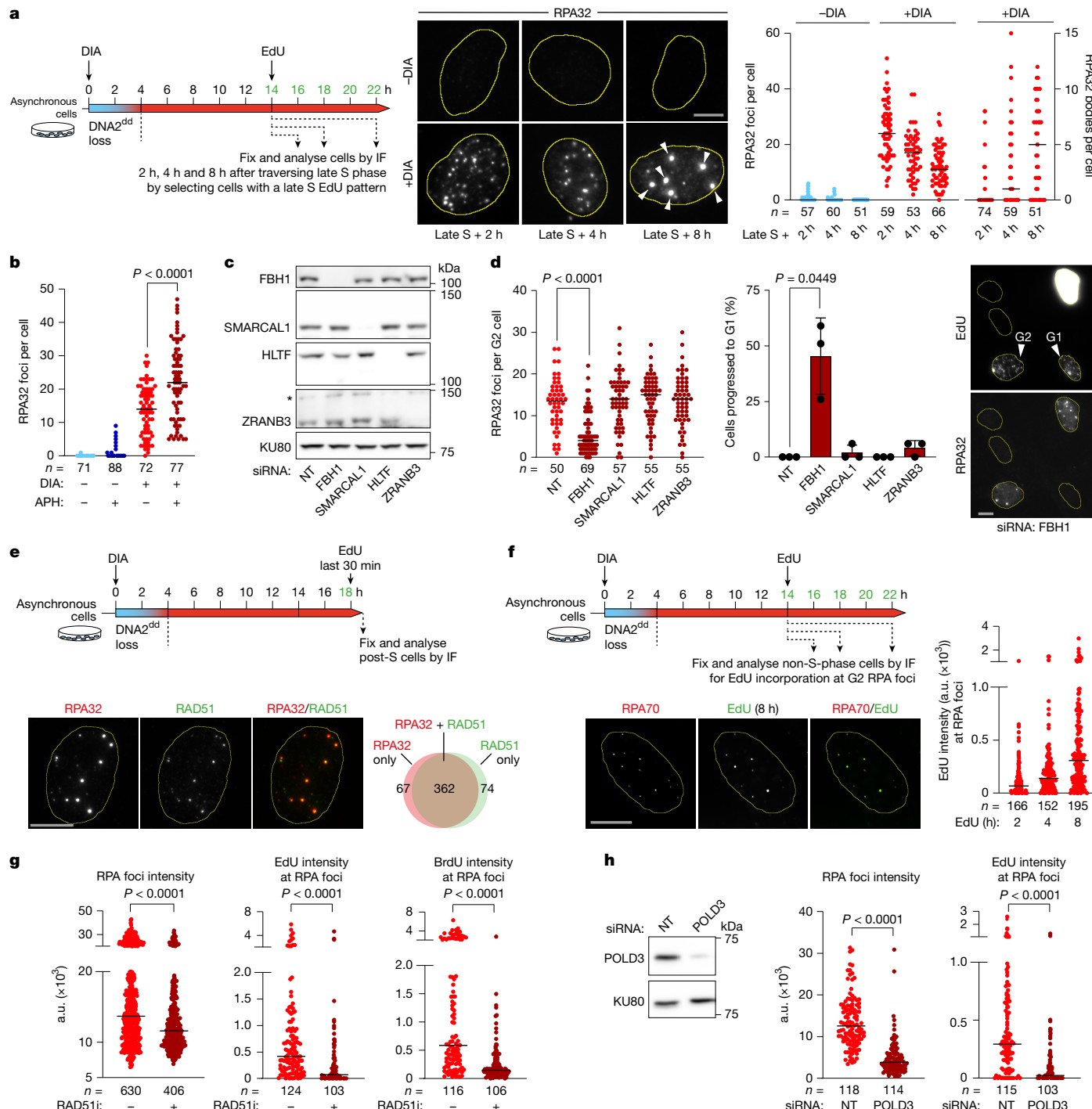

**Fig. 3 | RAD51–POLD3-dependent RPA–ssDNA at persistent replication intermediates in DNA2-deprived cells. a**, Experimental set-up and representative images of G2 RPA foci and large RPA bodies (white arrowheads) in the indicated numbers (*n*) of RPE-1 *OsTIR1 DNA2^{dd}* cells from two independent experiments. **b**, As described in **a**, but APH (0.4 μM) was added with DIA and removed upon EdU addition. RPA foci in the indicated number (*n*) of cells from two independent experiments. **c**, Western blots for siRNA-treated RPE-1 *OsTIR1 DNA2^{dd}* cells. KU80, loading control. **d**, RPA foci analysis, as described in **a**, after siRNA treatment. Percentage of cells progressed into G1 after traversing the late S phase from *n* = 3 independent experiments with representative images. **e**, Experimental set-up and images representative of three independent experiments showing the detection of RPA and RAD51 across 65 DIA-treated RPE-1 *OsTIR1 DNA2^{dd}* cells in G2 (no S-phase EdU incorporation). **f**, Detection of

G2 DNA synthesis in DIA-treated RPE-1 *OsTIR1 DNA2^{dd}* cells. Representative of two independent experiments with foci (*n*) analysed over 30 (2 h and 4 h) or 37 (8 h) cells. Representative image 8 h after the addition of EdU. **g**, As described in **f**, with EdU or BrdU (non-denaturing conditions) and RAD51i B02 (6.25 μM) added at *t* = 0 h. Foci (*n*) analysed over 50 (RPA) or 21 (4 h EdU) cells (representative of three independent experiments), and 26 (4 h BrdU − RAD51i) or 25 (BrdU + RAD51i) cells (representative of two independent experiments). **h**, Western blot analysis of siRNA-treated (48 h) RPE-1 *OsTIR1 DNA2^{dd}* cells. POLD3-depleted cells treated, as described in **f**. RPA and EdU foci (*n*) analysed, as described in **g**, over 20 (siNT) and 23 (siPOLD3) cells (representative of three independent experiments). Significance was determined by two-sided Mann–Whitney *U*-test or two-sided *t*-test with Welch's correction (**d**). Bars, median. Scale bars, 10 μm. NT, non-targeting.

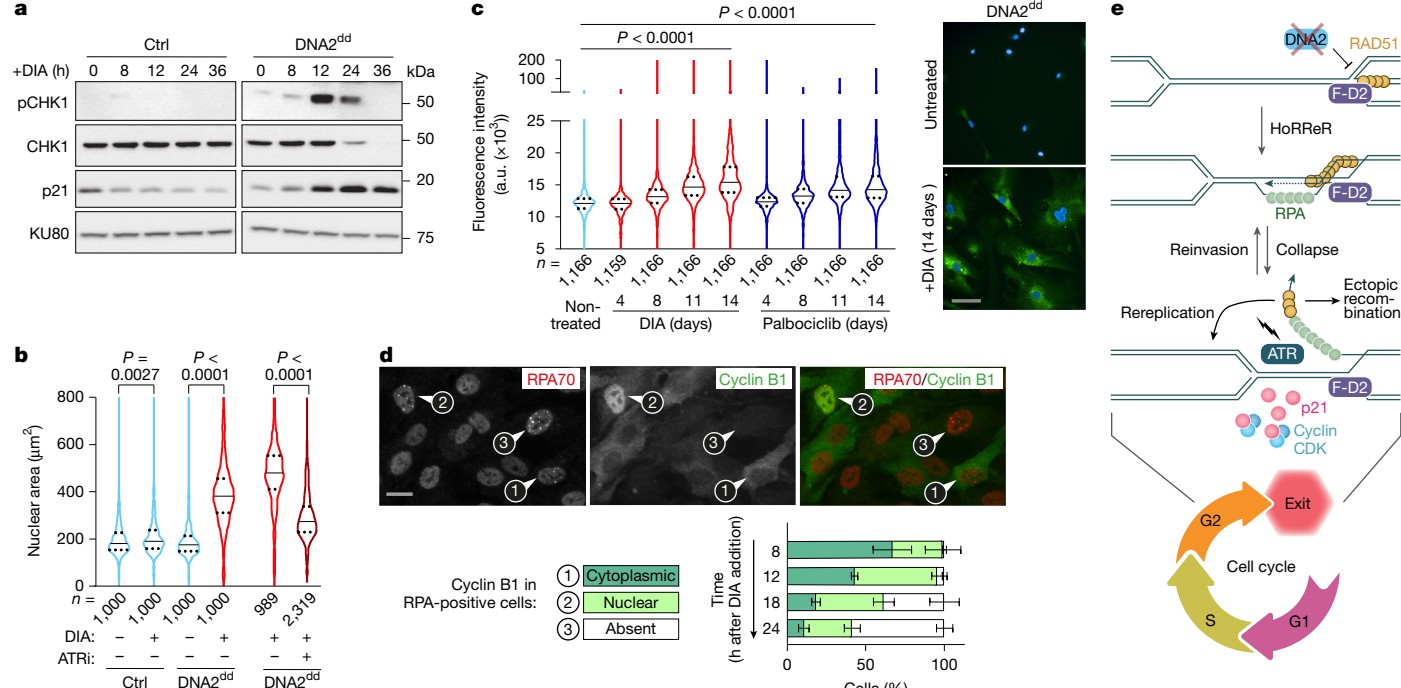

**Fig. 4 | Loss of DNA2 triggers ATR–p21-dependent cell-cycle exit from G2.**
**a**, Western blots (representative of three independent experiments) for DIA-treated RPE-1 *OsTIR1* (Ctrl) and RPE-1 *OsTIR1 DNA2$^{dd}$* cells. KU80, loading control. **b**, Nuclear area for the indicated number (*n*) of RPE-1 *OsTIR1* (Ctrl) and RPE-1 *OsTIR1 DNA2$^{dd}$* cells treated as indicated for 3 days. The median and quartiles are marked. ATRi, 1 µM VE-821. Two-sided Mann–Whitney *U*-test was applied. **c**, Representative images and β-galactosidase activity measurements for the indicated number (*n*) of RPE-1 *OsTIR1 DNA2$^{dd}$* cells treated with DIA or palbociclib (5 µM) for the indicated times over two independent experiments. The median and quartiles are marked, and one-way analysis of variance was

applied. **d**, Representative images and quantification of subcellular cyclin B1 localization in DIA-treated RPA-foci-positive RPE-1 *OsTIR1 DNA2$^{dd}$* cells (*n* = 3 independent experiments). Error bars represent the mean ± s.d. **e**, Model of the essential role of DNA2. At stalled RFs, DNA2-dependent fork processing counteracts fork reversal, thereby promoting fork reactivation and faithful DNA replication. Upon the loss of DNA2, reversed forks persist and give rise to HoRReR. Excessive recombination-dependent DNA synthesis in DNA2-deprived cells results in the accumulation of ssDNA, and cells invariably respond by ATR-dependent checkpoint signalling, p21-mediated sequestration of cyclin B1 and permanent cell-cycle exit before mitosis. Scale bars, 100 µm (**c**), 20 µm (**d**).

observed among cultured patient fibroblasts[5–7]. To model the effect of reduced DNA2 concentrations, we treated RPE-1 *OsTIR1 DNA2$^{dd}$* cells with doxycycline and titrated IAA to gradually induce the mAID degradation tag (Extended Data Fig. 11a). At low IAA concentrations, cell proliferation was largely unaffected, and nuclei showed no enlargement. High IAA concentrations blocked proliferation, and cells showed enlarged nuclei, indicating cell-cycle exit, as expected from the complete loss of DNA2. Intermediate IAA concentrations allowed RPE-1 *OsTIR1 DNA2$^{dd}$* cells to proliferate, although at a reduced rate (Fig. 5a). At these intermediate IAA concentrations, cells segregated into two distinct populations with either normal or enlarged nuclear size (Fig. 5b). Thus, the propensity of individual cells to exit the cell cycle grows as DNA2 concentrations decrease.

Next, we examined the fate of individual RPE-1 *OsTIR1 DNA2$^{dd}$* cells with intermediate concentrations of DNA2 by analysing the RPA and γH2AX foci 8 h after completion of the late S phase. As expected, the cells either persisted in G2 or had progressed through mitosis and into the next G1 phase (Fig. 5c,d). Lower IAA doses led to fewer RPA and γH2AX foci, and an increasing subset of cells avoided the RPA–ssDNA accumulation in G2 altogether, allowing cells to divide without overt DNA damage (Fig. 5d). Consistently, we did not detect elevated levels of micronuclei for cells treated with intermediate IAA doses (Extended Data Fig. 11b). A stochastic cell-cycle exit phenotype was recapitulated upon expression of the Seckel syndrome patient-derived variant[6] DNA2 T655A, with a mutation in the helicase domain in DNA2-deprived cells (Fig. 5e–h). By contrast, expression of DNA2 L48P, a variant linked to Rothmund–Thomson-related syndrome[7], failed to rescue cell proliferation, cell-cycle exit or γH2AX/RPA foci formation in DNA2-deprived cells (Fig. 5e–h and Extended Data Fig. 11c). These data validate the

pathogenicity of the DNA2 T655A and L48P patient mutations and indicate that they result in partial and complete loss of DNA2 function, respectively. The phenotypic dichotomy exhibited by cells with reduced DNA2 concentrations or upon expression of the pathophysiologically relevant DNA2 hypomorph T655A suggests that when cells reach a critical level of DNA2 activity, they either avoid ATR-mediated cell-cycle exit and continue to proliferate, or they exhibit a DNA2-loss phenotype and enter senescence. This switch-like behaviour reduces the proliferative potential of a cell population with low DNA2 activity, providing a molecular rationale for the hypocellularity and primordial dwarfism phenotypes of patients with *DNA2* mutations.

## Discussion

Our data revealed the essential role of *DNA2* in eukaryotes. Previously, we have shown that Dna2 dysfunction in budding yeast exposes cells to a terminal G2/M checkpoint arrest after transient RF arrest[20,22]. The block to M phase entry was alleviated by disruption of the DNA damage checkpoint or mutations compromising recombination-dependent DNA synthesis; in either case, cells progressed into mitosis with incompletely replicated chromosomes[20,22]. These findings led us to propose that elevated HoRReR in lieu of Dna2-mediated RF recovery results in a fatal checkpoint response triggered by recombination-dependent fork restart intermediates[18]. Here we demonstrate, using a site-specific replication barrier in fission yeast, that Dna2 dysfunction results in dramatically increased HoRReR. Thus, Dna2-dependent fork processing is indispensable to prevent the accumulation of recombinogenic chicken-foot structures, consistent with our findings[2] of elevated fork reversal in fission yeast upon loss of Dna2. We further show that genetic

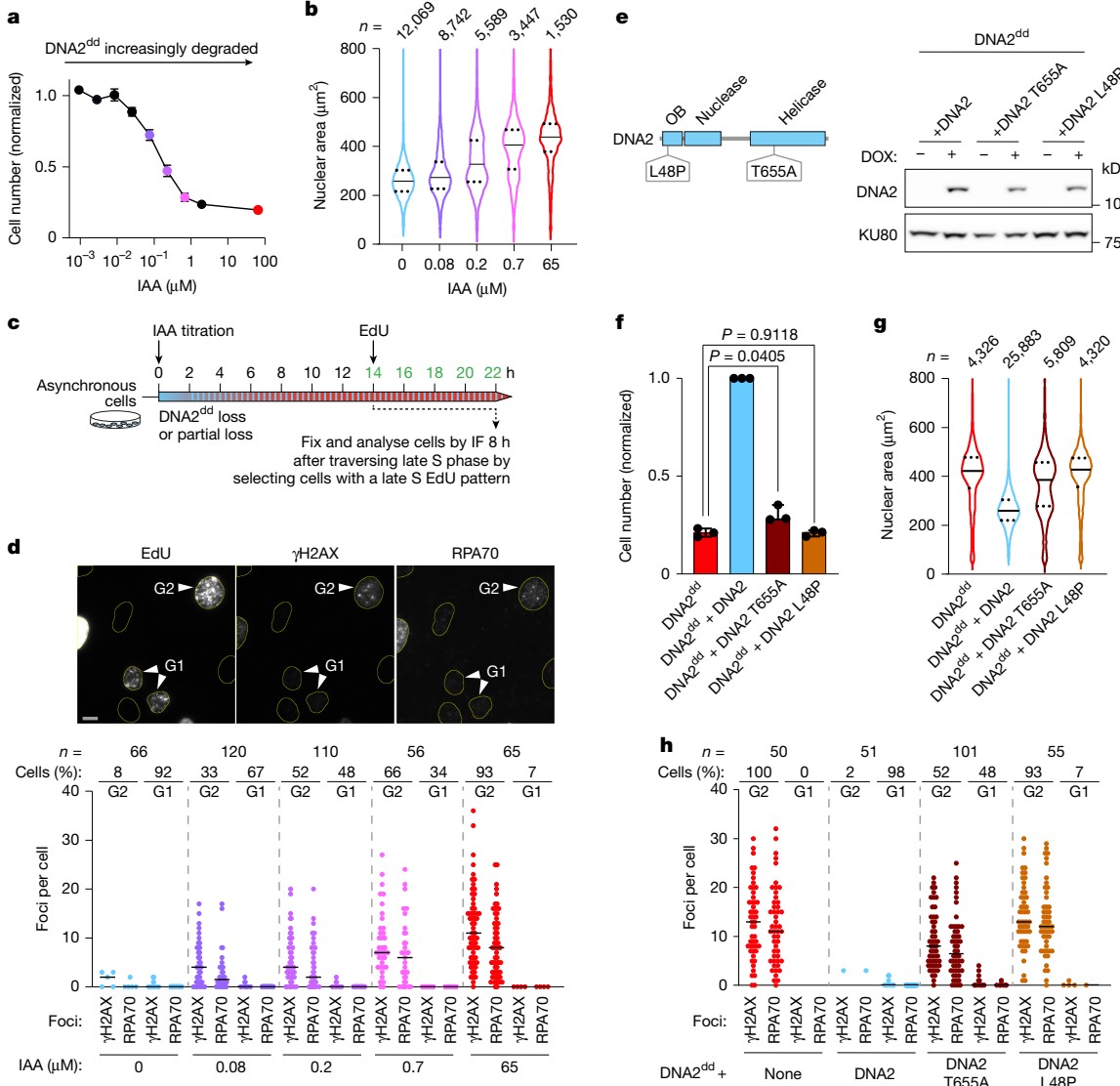

**Fig. 5 | Cells with critically low concentrations of DNA2 undergo stochastic cell-cycle exit. a**, Proliferation of 1 μg ml⁻¹ doxycycline-treated RPE-1 *OsTIR1 DNA2*^dd cells titrated with IAA for 3 days. A total of 1,500 cells were plated, and data were normalized to no-IAA samples and presented as mean values ± s.d. for *n* = 3 technical replicates. **b**, Nuclear area for the indicated number (*n*) of RPE-1 *OsTIR1 DNA2*^dd cells, treated as described in **a** (representative of three technical replicates). **c**, Experimental set-up for the analysis of RPA and γH2AX foci in IAA-titrated RPE-1 *OsTIR1 DNA2*^dd cells 8 h after the late S phase. **d**, Representative images and quantification with determination of cell-cycle stage for the indicated number (*n*) of cells from two independent experiments. Bars, median. Scale bar, 10 μm. **e**, Schematic showing primordial dwarfism-associated mutations in DNA2; western blot (*n* = 1) showing the

doxycycline-induced (1 μg ml⁻¹; 8 h) expression of wild-type and mutant DNA2 in RPE-1 *OsTIR1 DNA2*^dd cells. DOX, doxycycline; KU80, loading control. **f**, Proliferation of RPE-1 *OsTIR1 DNA2*^dd cells complemented with wild-type or mutant DNA2, as indicated after 3 days of DIA treatment. A total of 1,500 cells were plated; normalized data are presented as mean values ± s.d. for *n* = 3 independent experiments. Statistical analysis by two-sided *t*-test with Welch's correction. **g**, Nuclear area for the indicated number (*n*) of cells treated, as described in **f**. Representative example of three independent experiments, median and quartiles marked. **h**, Quantification of RPA foci, γH2AX foci and cell-cycle stage for the indicated number (*n*) of cells from two independent experiments; analysis as described in **d** and cells treated as described in **c**, with DIA treatment replacing IAA titration.

perturbation of recombination-dependent DNA synthesis rescues excessive HoRReR, checkpoint activation and cell-cycle arrest in Dna2 mutant cells. Thus, Dna2 dysfunction causes inviability by durable G2/M checkpoint arrest in distantly related yeast species through unrestricted HoRReR.

In DNA2-deprived human cells, we detected persistent replication intermediates, which give rise to substantial RPA–ssDNA in G2 in a recombination-dependent manner. This is invariably associated with checkpoint-mediated cell-cycle exit from G2 into senescence, a phenotype akin to the sudden and permanent G2/M arrest upon the induction of Dna2 deficiency in yeast[21,57]. Disruption of the DNA damage checkpoint in DNA2-deprived cells enabled mitotic progression,

although at the expense of micronuclei formation indicative of unfaithful chromosome replication in the absence of DNA2. Similarly, limiting the availability of HoRReR substrates by suppressing FBH1-mediated RF reversal diminished RPA–ssDNA accumulation upon DNA2 loss and allowed a subset of cells to progress through mitosis. These striking parallels between the yeast and human systems and the absence of a detectable Okazaki fragment processing defect in DNA2-deprived cells lead to a unified model, in which *DNA2*-dependent fork processing has a critical role in RF recovery and acts as an essential gatekeeper to HoRReR-induced cessation of cell-cycle progression across eukaryotes.

The recombination-dependent reinitiation of DNA synthesis at stalled or collapsed RFs can promote replication completion[29]. Consistently,

most ssDNA accumulations at stalled RFs in DNA2-deprived were transient. However, prominent RPA bodies in DNA2-deprived cells indicate that a subset of HoRReR events result in extended recombination-dependent DNA synthesis. The determinants for these dead-end recombination events remain to be elucidated but may include local origin scarcity precluding D-loop resolution by fork convergence. Moreover, D-loop instability could result in cycles of expulsion, reinvasion and extension of the nascent strand along contiguous or non-contiguous (template switch) templates. Finally, we consider the possibility that HoRReR or D-loop collapse at a stalled RF may coincide with the arrival of an oncoming RF, in which case strand invasion might target the newly replicated sister chromatids in its wake, causing localized re-replication (Fig. 4e). To our knowledge, HoRReR at stalled RFs has not previously been recognized as a trigger for senescence, which is more commonly linked to telomere erosion (replicative senescence), oncogene-induced replication stress or exogenous DNA damage[58]. Self-inflicted senescence adds to other well-documented risks[31] associated with HoRReR, which include the introduction of mutations and chromosome rearrangements by ectopic recombination. We suggest that DNA2 has a key role in mitigating these dangers by largely suppressing HoRReR.

Our results indicate that high concentrations of DNA2 expression in cancer cells[8–11] might reflect an adaptation to intrinsic DNA replication stress[59], necessitating stricter control over potentially toxic levels of HoRReR at stalled RFs. Checkpoint override in DNA2-deprived cells resulted in the formation of micronuclei, indicating progression into mitosis with unresolved replication and recombination intermediates. This supports the notion that checkpoint deficiency may identify cancer cells, in which emerging DNA2 inhibitors[60–62] could be used to cause catastrophic mitotic DNA damage. In checkpoint-proficient cancerous cells, elevated replication stress may synergize with DNA2 inhibitors to create a window of opportunity for triggering senescence at doses tolerated by normal cells.

Our experimental downregulation of DNA2 and expression of the Seckel syndrome patient-derived DNA2 T655A hypomorph indicate that a critical level of DNA2 activity exists at which cells can either continue to proliferate without overt DNA damage or enter HoRReR-induced senescence. This dichotomy in cell fate provides a compelling model for why primordial dwarfism-associated *DNA2* mutations[5–7] allow organismal survival while severely limiting the number of proliferative cells, thus causing intrauterine and postnatal growth restriction. A growing number of genes with important roles in DNA replication and RF stability are being linked to primordial dwarfism[63–65]. In the future, it will be important to investigate whether the paradigm of HoRReR-induced senescence extends beyond *DNA2*-linked disease and contributes to the hypocellularity in patients affected by other DNA replication disorders.

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

# Methods

## Yeast methods

*S. pombe* strains (Supplementary Table 1) were grown and genetically manipulated according to standard protocols[66]. The *dna2-2* allele was generated by Cre recombinase-mediated cassette exchange[67]. Bright-field microscopy images of logarithmically growing cells were taken using a Nikon Eclipse E400 microscope with ×10 magnification to determine the cell length using Fiji (ImageJ; v.1.53t)[68]. Direct repeat recombination at *RTS1* was measured in reporter strains harbouring direct repeat *ade6*− heteroalleles[26], as described[69], starting with *his*+ *ade*− cells from colonies grown on minimal medium containing low adenine and no histidine. For plasmid-based expression of wild-type Dna2, the *dna2*+ gene was cloned into pREP42 (Addgene; plasmid no. 52691). HoRReR-mediated DNA synthesis at *RTS1* was monitored by polymerase usage sequencing (Pu-Seq), and Pu-Seq traces were analysed using a custom R script, as described[70]. To analyse the effect of truncating Cdc27, the *cdc27-D1* allele was introduced into strains[71] BAY123–126 harbouring an active (*RTS1*-ON) or inactive (*RTS1*-OFF; disabled by *rtf1*+ deletion) RF barrier and DNA polymerase ε (*cdc20*–M630F) and δ (*cdc6*–L591M) mutations that allow for polymerase usage tracking to give strains BAY127–130.

## Human cell line construction and cell culture

The coding sequence for the mAID/SMASh double-degron tag on *DNA2* was inserted using CRISPR–Cas9-mediated homology-directed repair in human telomerase reverse transcriptase immortalized RPE-1 *OsTIR1* cells, as described[42]. The *DNA2* targeting construct contained a left arm corresponding to 239 bp immediately upstream of the *DNA2* STOP codon and a right arm corresponding to 500 bp immediately downstream of the *DNA2* STOP codon. This construct was generated using HiFi Assembly (NEB; E5520S). The left arm contained a silent mutation targeting the protospacer adjacent motif site to prevent recutting by Cas9 after double-degron integration (5′-TTTGAAAGTCACCCAATATGTGa; mutation shown in lower case). Guide RNA design was aided by CHOPCHOP[72], and the guide 5′-TTTGAAAGTCACCCAATATG was inserted into the vector pSpCas9(BB)–2A–GFP (Addgene; plasmid no. 48138), as described[73]. To generate the RPE-1 *OsTIR1 DNA2*dd cell line, 0.5 μg of the guide RNA–Cas9 expression plasmid was co-transfected with 1.5 μg of the *DNA2* targeting construct using Lipofectamine LTX Reagent with PLUS Reagent (Invitrogen) into RPE-1 *OsTIR1* cells[42]. After serial dilution into 96-well plates, clones were incubated for 3 weeks in the presence of 400 μg ml−1 G418 (Corning; 30-234-CR) before preparing gDNA from surviving single clones to screen for double-degron integration by diagnostic polymerase chain reaction (PCR) using three different primer pairs (PCR1: 5′-TTGTGGTCCTGTAAGTCAAATTAC + 5′-AGCCCCAGAAGAATCTGTTACCCC; PCR2: 5′-GAGATCTAGCCTGGT GGCAGAGC + 5′-GAGGCCTTATACATCCTCAGGTCG; PCR3: 5′-GCGAC TTGCGGAAGTCCAGGACTCTTCC + 5′-CTGCTGCAGAGGTGTGCTCA GAAGTC) with Phusion Polymerase Master Mix (NEB; M0531) according to the manufacturer's instructions. Cells were routinely grown at 37 °C with 5% $CO_2$ in DMEM/F12 (Sigma-Aldrich; D8437) medium containing 10% Tet-free fetal bovine serum and penicillin/streptomycin, with Asv (3 μM; GENERON; A3195), IAA (65 μM or as indicated; Sigma-Aldrich; I5148), doxycycline hyclate (1 μg ml−1; Thermo Fisher Scientific), ATMi KU55933 (Selleckchem), ATRi VE-821 (Selleckchem), RAD51i B02 (Merck; 553525), APH (Abcam; Ab142400), etoposide (Sigma-Aldrich; E1383), FEN1i[43], poly(ADP-ribose) glycohydrolase inhibitor (PARGi) PDD 0017273 (Tocris/Sigma-Aldrich; 5952/SML1781), as indicated. Cells were regularly tested to exclude *Mycoplasma* infection.

## Sleeping Beauty transposon-based complementation of RPE-1 *OsTIR1 DNA2*dd cells

The Tet-on Sleeping Beauty plasmid pSB-tet-BP[74] (Addgene; no. 60496) with constitutive expression of blue fluorescent protein was used to integrate DNA2 cDNA or a luciferase control into RPE-1 *OsTIR1 DNA2*dd cells. DNA2 cDNA was inserted into pSB-tet-BP after digestion with and NcoI and ClaI. Then, 1.9 μg of pSB DNA and 100 ng of pCMV(CAT) T7-SB100 DNA (Addgene; no. 34879) for transposase expression were transfected into RPE-1 *OsTIR1 DNA2*dd cells using Lipofectamine LTX with PLUS Reagent (Invitrogen; A12621). Forty-eight hours after transfection, cells were FACS sorted into a 96-well plate for blue fluorescent protein expression (excitation at 456 nm) using a FACSMelody Cell Sorter (BD Biosciences). Clonal cell cultures were then analysed for DNA2 expression by immunoblotting after doxycycline addition. For complementation with mutant versions of DNA2 and corresponding wild-type controls, cells were transfected, as detailed above, and cell pools were selected with puromycin (1 μg ml−1) for 8–10 days before analysis.

## Immunoblotting

For immunodetection of proteins on western blots, $2 \times 10^5$ cells were plated and drug-treated as appropriate no sooner than 18 h later. The cells were then lysed in 60 μl lysis buffer (20 mM HEPES (pH 7.4), 0.5% NP-40, 40 mM NaCl, 2 mM $MgCl_2$, 1× EDTA-free protease inhibitor cocktail (Roche; 4693132001), 1× PhosSTOP phosphatase inhibitor cocktail (Roche; 4906845001) and 250 U ml−1 of benzonase (Merck; E1014)). Lysates were incubated for 30 min on ice and cleared by centrifugation (16,000*g*; 10 min). After determination of total protein concentration using bicinchoninic acid assay (Pierce; 23227) and total protein normalization, the samples were boiled with Laemmli protein sample buffer for SDS–PAGE (Bio-Rad; 1610747). Immunoblotting was performed using standard techniques with the following antibodies and dilutions: anti-CHK1 G4 (Santa Cruz Biotechnology; sc-8408; 1:1,000), anti-phospho-CHK1 (S345) 133D3 (Cell Signaling Technology; 2346; 1:1,000), anti-KU80 EPR3468 (Abcam; ab80592; 1:2,500), anti-p21 EA10 (Merck; OP64; 1:1,000), anti-DNA2 antibody raised against an N-terminal immunogen (Proteintech; 21599-1-AP; 1:500), anti-DNA2 antibody raised against a C-terminal immunogen (Abcam; Ab96488; 1:500), anti-POLD3 (Abnova; H00010714-M0; 1:1,000), anti-FBH1 (Abcam; 2353C1a; 1:100), anti-SMARCAL1 (Santa Cruz Biotechnology; SC-376377; 1:100), anti-HLTF (Proteintech; 14286-1-AP; 1:300) and anti-ZRANB3 (Proteintech; 23111-1-AP; 1:500). Double-degron addition to the DNA2 C terminus precludes detection with Ab96488. For detection of DNA2dd with 21599-1-AP, $3 \times 10^6$ RPE-1 *OsTIR1 DNA2*dd cells were plated and DIA-treated 24 h later, as indicated. The cells were lysed in 400-μl lysis buffer and normalized for total protein content (as described above), and NaCl concentration was adjusted to 150 mM. Then, 1-μl anti-DNA2 antibody (21599-1-AP) was added per 100-μl lysate volume, and the mixture was rotated at 4 °C for 1 h. Following centrifugation at 16,000*g* for 10 min, the supernatant was mixed with Dynabeads Protein G for immunoprecipitation (Invitrogen; 10003D) and incubated with rolling at 4 °C for 2 h. The beads were washed three times with wash buffer (20 mM HEPES (pH 7.4) and 150 mM NaCl) and resuspended in Laemmli protein sample buffer for analysis by immunoblotting. All antibodies were diluted in 5% non-fat milk powder (Marvel) in Tris-buffered saline + 0.1% Tween-20, except phospho-specific antibodies, which were diluted in 5% BSA in Tris-buffered saline + 0.1% Tween-20. Secondary antibodies were HRP-linked anti-rabbit or anti-mouse IgG (Cell Signaling Technology; 7074 and 7076) used at a dilution of 1:5,000. Visualization was done using Hyperfilm (Fisher; 10627265) or ImageQuant LAS400 system with ImageQuant (Cytiva) western blot imaging software (v.12).

## Cell proliferation, nuclear size and senescence measurements

A total of 1,500 cells per well were plated in a black-walled 96-well plate (Corning; 3603). Up to 24 h later, the cells were treated, as indicated. For proliferation and nuclear size measurements, after 3 days of incubation, the cells were fixed by the addition of paraformaldehyde to the medium to a final concentration of 4% for 20 min. For senescence

analysis, the CellEvent Senescence Green Detection Kit (Invitrogen; C10850) was used according to the manufacturer's instructions. For all assays, cell fixation was followed by a PBS wash, and nuclei were stained using DAPI (0.2 µg ml$^{-1}$ in PBS; 10 min), washed in PBS and stored in PBS for imaging. Nuclei were imaged using a PerkinElmer Operetta CLS High-Content Analysis System with a ×10/0.3 objective and an Andor Zyla 5.5. camera. The number of nuclei, nuclear area, number of micro-nuclei and fluorescence intensity in the area surrounding the nuclei were determined using the Harmony imaging platform. For colony formation assays, 100 cells were plated in six-well plates and treated 24 h later, as indicated. After at least 9 days, colonies were stained by the addition of 0.1% methylene blue to the medium for counting.

## Analysis of cell viability

Cells ($5 \times 10^4$) were plated in six-well plates and treated 24–30 h later. After 60 h, the cells were trypsinized and resuspended in PBS before adding trypan blue vital dye to a final concentration of 0.2% (Thermo Fisher Scientific; 15250061). Within 5 min, the cells were counted and scored for exclusion or uptake of the blue dye.

## Flow cytometry

Cells ($8 \times 10^5$) were plated in a 6-cm dish and treated 24 h later. At appropriate time points after treatment, the cells were fixed in 70% ethanol at 4 °C overnight and stained with FxCycle PI/RNase Staining Solution (Invitrogen; F10797) according to the manufacturer's instructions. Cell-cycle profiles were obtained from data acquired on an Accuri C6 Flow Cytometer (BD Biosciences) and analysed using BD Accuri C6 (v.1) software. The gating strategy is described in Supplementary Fig. 2.

## Live-cell imaging

Cells were seeded in a four-well coverslip chamber (Sarstedt; 94.6140.402). Five hours before the addition of DIA, nuclei were stained using the SiR-DNA Kit (Spirochrome; SC007). After the addition of DIA, live-cell videos were recorded using the ZEISS Axio Observer Z1 system with a Plan-Apochromat ×20/0.8 M27 objective. Tiling scanning of an area of 1,270 µm$^2$ was performed every 20 min for 60 h. ZEN blue software (v.2.6) was used for image and video acquisition and image tile stitching. Cells that underwent cell division within the first 12 h of the recording were tracked manually to observe subsequent mitoses.

## Immunofluorescence microscopy

Cells ($2 \times 10^5$) were seeded on 24-mm coverslips in six-well plates; 5,000 cells per well were seeded in PhenoPlate 96-well plates (Revvity; 6055302). The cells were treated with DIA on the following day in the presence or absence of 10 µM EdU or BrdU (Merck; B5002) for the indicated times, washed with ice-cold PBS and incubated on ice with cytoskeletal pre-extraction buffer (10 mM PIPES (pH 7.0), 100 mM NaCl, 300 mM sucrose, 3 mM MgCl$_2$, 1 mM EGTA and 0.5% Triton X-100) for 5 min. Following fixation with 4% paraformaldehyde in PBS for 10 min and two washes in PBS, the cells were permeabilized in 0.2% Triton-X 100 in PBS for 5 min. To image cyclin B1, pre-extraction and fixation were replaced with a single step of incubation in fixative solution (250 mM HEPES (pH 7.4), 0.1% Triton X-100 and 4% paraformaldehyde in PBS) for 20 min on ice, followed by washing and permeabilization, as described above. Mono(ADP-ribose) (MAR)/PAR imaging was performed, as described[43]. Briefly, the cells were washed in pre-extraction buffer (25 mM HEPES (pH 7.4), 50 mM NaCl, 1 mM EDTA, 3 mM MgCl$_2$, 0.3 M sucrose and 0.5% Triton X-100) supplemented with PARGi for 5 min on ice and fixed with cold 4% formaldehyde for 15 min. The cells were then permeabilized using ice-cold methanol/acetone solution (1:1) for 5 min and PBS containing 0.5% Triton X-100. In all cases, the cells were blocked with 5% BSA in PBS for 30 min before EdU labelling with the Click-iT EdU Alexa Fluor 488 or 647 Imaging Kit (Invitrogen; C10337) according to the manufacturer's instructions. The cells were incubated for 1 h with the indicated primary antibodies (in 5% BSA/PBS),

followed by incubation with secondary antibodies (in 5% BSA/PBS) for 1 h. After each incubation, the cells were washed three times for 5 min in PBS. DNA was counterstained with 0.2 µg ml$^{-1}$ of DAPI, and cover-slips were mounted on slides using VECTASHIELD Antifade Mounting Medium (Vector Laboratories; H-1000-10). The slides were imaged using a ZEISS Axio Observer Z1 epifluorescence microscopy system equipped with ×40/1.3 and ×100/1.4 oil Plan-APOCHROMAT objectives and a Hamamatsu ORCA-Flash4.0 LT Plus camera. Z-stacks were acquired at 200-nm intervals, and image deconvolution was performed, as described[47]. Fiji (ImageJ; v.1.53t) was used to generate representative images and to analyse foci brightness from slides, together with ZEN blue software (v.2.6). Multi-well plates were imaged with a Perkin-Elmer Operetta CLS High-Content Analysis System using a ×20/0.4 LW objective and an Andor Zyla 5.5. camera. Images were analysed using Harmony software (v.4.9). Cell-cycle phase determination was done by pulsing cells with EdU before fixation and measuring DAPI content per cell using ZEN blue software (v.2.6). To identify DNA2-deprived cells at different times after traversing the late S phase, we treated non-synchronous *OsTIR1 DNA2$^{dd}$* cells with DIA for 14 h before adding EdU to the culture medium to continuously label replicating DNA. The cells were fixed 2 h, 4 h and 8 h later, and only cells with a typical patchy and peripheral late S phase EdU incorporation pattern[47] (as seen in cells marked with arrowheads in Fig. 3d) were selected (that is, cells that had been in late S phase 2 h, 4 h and 8 h earlier). The overall γH2AX intensity was measured using ZEN blue software (v.2.6). MAR/PAR images were acquired using an Olympus IX-81 microscope equipped with scanR screening system using a ×40 objective at a single autofocus-directed z-position under non-saturating settings, as described previously[43]. Olympus scanR image analysis software (v.3.2.0) was used to analyse and quantify the fluorescence intensity of individual cells. Nuclei were identified by DAPI signal using an integrated intensity-based object detection module. G1, S and G2 phase cells were gated on the basis of EdU and DAPI intensity. The following antibodies and dilutions were used: anti-RPA32 9H8 (GeneTex; GTX22175; 1:500), anti-RPA70 (Abcam; ab97338; 1:1,000), anti-phospho-RPA32 (S4/8) (Bethyl Laboratories; A300-245A; 1:1,000), anti-RAD51 Ab-1 (EMD Millipore; PC130; 1:500), anti-FANCD2 (Novus Biologicals; NB100-182; 1:500), anti-γH2AX (S139) JBW301 (Sigma-Aldrich; 05-636; 1:500), anti-cyclin B1 (BD Biosciences; 610220; 1:100), anti-BrdU (BD Biosciences; 555627; 1:250) and anti-MAR/PAR (Cell Signaling Technology; 83732; 1:500). The following secondary antibodies (1:500) were used: donkey anti-mouse IgG Alexa Fluor 488, donkey anti-rabbit IgG Alexa Fluor 555, donkey anti-rabbit IgG Alexa Fluor 488 and goat anti-mouse IgG Alexa Fluor 647 (Thermo Fisher Scientific A-21202; A-21235; A-31572).

## Transfection with siRNA

To deplete p21 or POLD3 before DNA2$^{dd}$ degradation, cells were reverse transfected with p21 SMARTpool siRNA (Dharmacon; L003471-00; 20 nM final concentration), non-targeting control siRNA (Dharmacon; D-001810-10-05; 20 nM final concentration) or POLD3 siRNAs (Dharmacon; J026692-11 and J026692-12; 10 nM final concentration each) using Lipofectamine RNAiMAX Transfection Reagent (Invitrogen; 13778030), according to the manufacturer's instructions. After 30 h, the medium was changed and DIA was added, as indicated. Western blot samples were taken 48 h post-transfection. To deplete fork remodellers FBH1, SMARCAL1, HLTF and ZRANB3, Dharmacon siRNA pools (L-026692-01, L-013058-00, L-006448-00 and L-010025-01; 20 nM final concentration) were used. Transfections were carried out, as described above, but with an initial reverse transfection at the time of plating, followed by a second transfection at 24 h. The medium was then changed after 54 h and DIA was added. Western blot samples were taken at 72 h.

## Statistical analysis

Statistical tests were performed using Prism v.10.4 (GraphPad software).

## Reporting summary

Further information on research design is available in the Nature Portfolio Reporting Summary linked to this article.

## Data availability

All data supporting the findings of this study are available within the paper. Supplementary Information is available for this study. For gel source data, see Supplementary Fig. 1. Source data are provided with this paper.

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

**Acknowledgements** This study was supported by the Department for Science, Innovation and Technology through The Academy of Medical Sciences Professorship Scheme award AMSPR1\1018 to U.R. and by UKRI through BBSRC research grants BB/W008505/1 and BB/Y01166X/1 (J.J.R.H., R.A., K.D. and U.R.). K.-L.C. is supported by the Wellcome Trust Career Development Award 225348/Z/22/Z. A.M.C. acknowledges the Welcome Trust award 110047/Z/15/Z and KWC Cancer Research UK Award DRCRPG-Nov24/100005. We thank S. Forsburg, University of Southern California, for sharing the *pfh1-mt** cells, and H. Hochegger, University of Sussex, for RPE-1 *OsTIR1* cells and discussions on human cell line construction. We are grateful for the support provided by the Genome Damage and Stability Centre's cell culture and live-cell sorting facilities and help from Y. Gu of the Wolfson Centre for Biological Imaging, University of Sussex.

**Author contributions** J.J.R.H., R.A. and U.R. designed this study. J.J.R.H. engineered and analysed the human cell lines with help from K.D. R.A. generated yeast strains and analysed recombination rates at *RTS1* with help from D.J. A.M.B. generated the yeast strains and performed Pu-Seq experiments. A.M.C. supervised A.M.B. and analysed the Pu-Seq data. K.W.C. and A.V. performed the MAR/PAR analyses in human cells. K.-L.C. advised on microscopy and set up live-cell imaging. U.R. wrote the paper with contributions from all the authors.

**Competing interests** The authors declare no competing interests.

## Additional information

**Correspondence and requests for materials** should be addressed to Ulrich Rass.

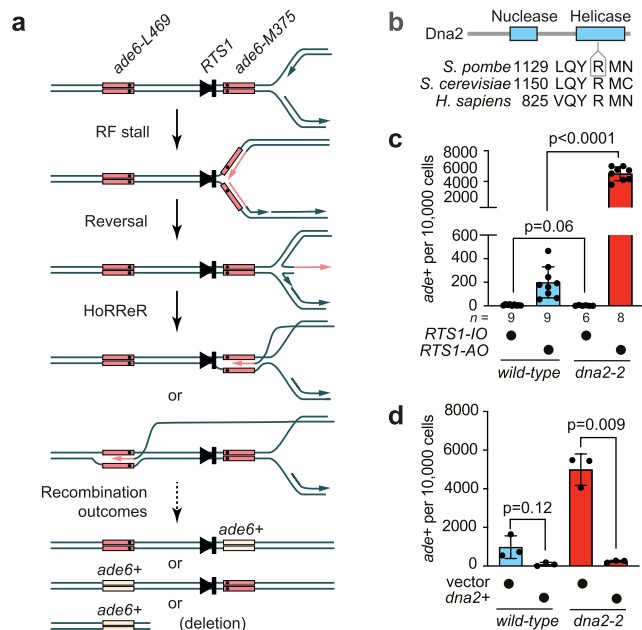

**Extended Data Fig. 1 | Dna2 controls HoRReR at *RTS1*-stalled RFs. a**, Schematic of the genetic reporter used to measure HoRReR levels upon fork arrest at the *RTS1* replication barrier in *S. pombe*. HoRReR-associated template switching allows restoration of the *ade*+ marker (shown in white) with recombination outcomes including gene conversion between *ade6-L469* and *ade6-M375* (shown in magenta with point mutations indicated by black circles) or partial deletion of the heteroallele locus. **b**, Cartoon view of Dna2 with the conserved amino acid residue mutated in helicase-defective allele *dna2-2* (R1132) indicated. **c**, Measurements of *ade*+ recombinants in wild-type and *dna2-2* cells in strains with the *RTS1* replication barrier inserted in inactive (*RTS1-IO*) or active (*RTS1-AO*) orientation over the indicated numbers (*n*) of independent experiments. **d**, Effect of plasmid-based *dna2*+ expression on the formation of HoRReR-induced *ade*+ recombinants at *RTS1-AO* (*n* = 3 independent experiments). Data is presented as mean ± s.d. and significance was determined using the two-sided t-test with Welch's correction. A.u., arbitrary units.

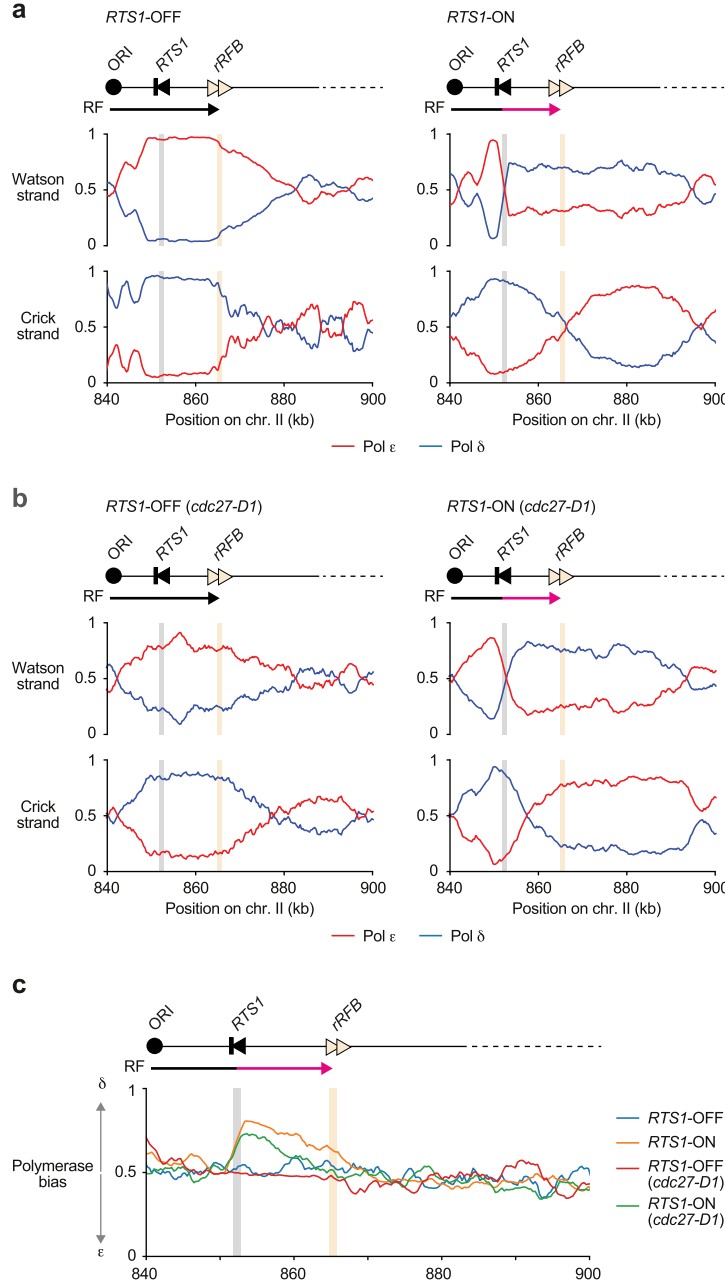

**Extended Data Fig. 2 | The Cdc27 C-terminal domain is required for HoRReR at *RTS1*. a**, Pu-Seq (polymerase usage sequencing) traces (representative of two independent experiments) at *RTS1* on chromosome II when the replication barrier is ON (right panel) and OFF (left panel). Ratios for Pol ε (red) and Pol δ (blue) usage on the Watson and the Crick strands are shown. Note the switch from Pol ε to δ on the Watson strand at the active *RTS1* indicative of HoRReR-associated δ/δ synthesis. The direction of travel for the RF from the origin of replication (ORI) towards *RTS1* is indicated. A red arrow denotes the region of δ/δ synthesis downstream of *RTS1* and upstream of a second replication barrier (rRFB) which was inserted to delay convergence with the leftward moving RF. **b**, Pu-Seq analysis as in panel **a**, but in a *cdc27-D1* background. **c**, Polymerase bias graph calculated using the ratio of polymerase usage on both DNA strands as shown in panels **a** and **b**. Note the strong Pol δ bias indicating HoRReR-associated δ/δ synthesis when *RTS1* is ON. This bias is reduced in the *cdc27-D1* background. The reduction in both the height and width of the Pol δ bias peak suggests that a reduction in restart and D-loop DNA synthesis upon truncation of Cdc27.

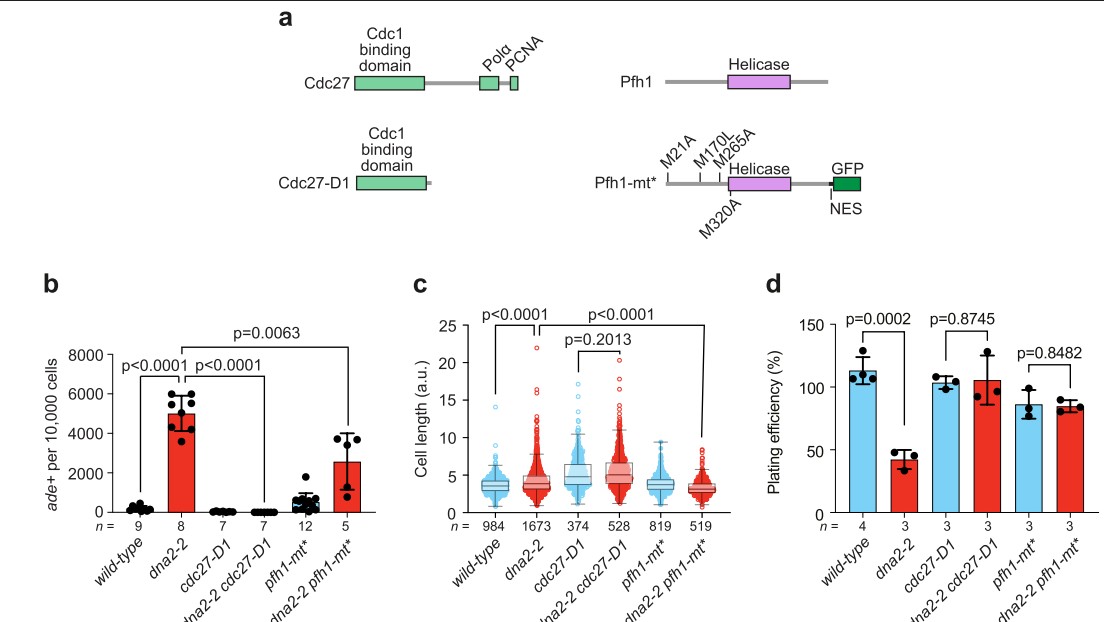

**Extended Data Fig. 3 | Suppressing HoRReR alleviates checkpoint activation and lethality in Dna2-defective cells. a**, Cartoon view of Cdc27 and Pfh1. Cdc27-D1 retains the binding domain for Cdc1, but loses those for Pol α, and PCNA. Pfh1-mt* harbours the indicated mutations which eliminate nuclear protein isoforms, a nuclear export signal (NES) from human immunodeficiency virus Rev protein, and a GFP tag. **b**, Effect of the recombination-dependent DNA synthesis-defective mutants Cdc27-D1 and Pfh1-mt* on *ade+* recombinants at *RTS1-AO* (see Extended Data Fig. 1a) measured over the indicated numbers (*n*) of independent experiments. Data is presented as mean ± s.d. Significance was determined using the two-sided t-test with Welch's correction (Bonferroni correction for multiple pairwise comparisons adjusts significance threshold to p = 0.017). **c**, Quantification of cell length as a measure of checkpoint activity in the indicated strains and numbers (*n*) of cells over two independent experiments. Significance was determined using the two-sided Mann-Whitney *U* test (Bonferroni correction for multiple pairwise comparisons adjusts significance threshold to p = 0.025). Box centres are defined by the median, bounds by first to third quartile, and whiskers mark minima and maxima within 1.5 x interquartile range from the bounds of the box. A.u., arbitrary units. **d**, Cell viability of the indicated strains as measured by plating efficiency in the indicated numbers (*n*) of independent experiments. Data is presented as mean ± s.d. Significance was determined using the two-sided t-test with Welch's correction.

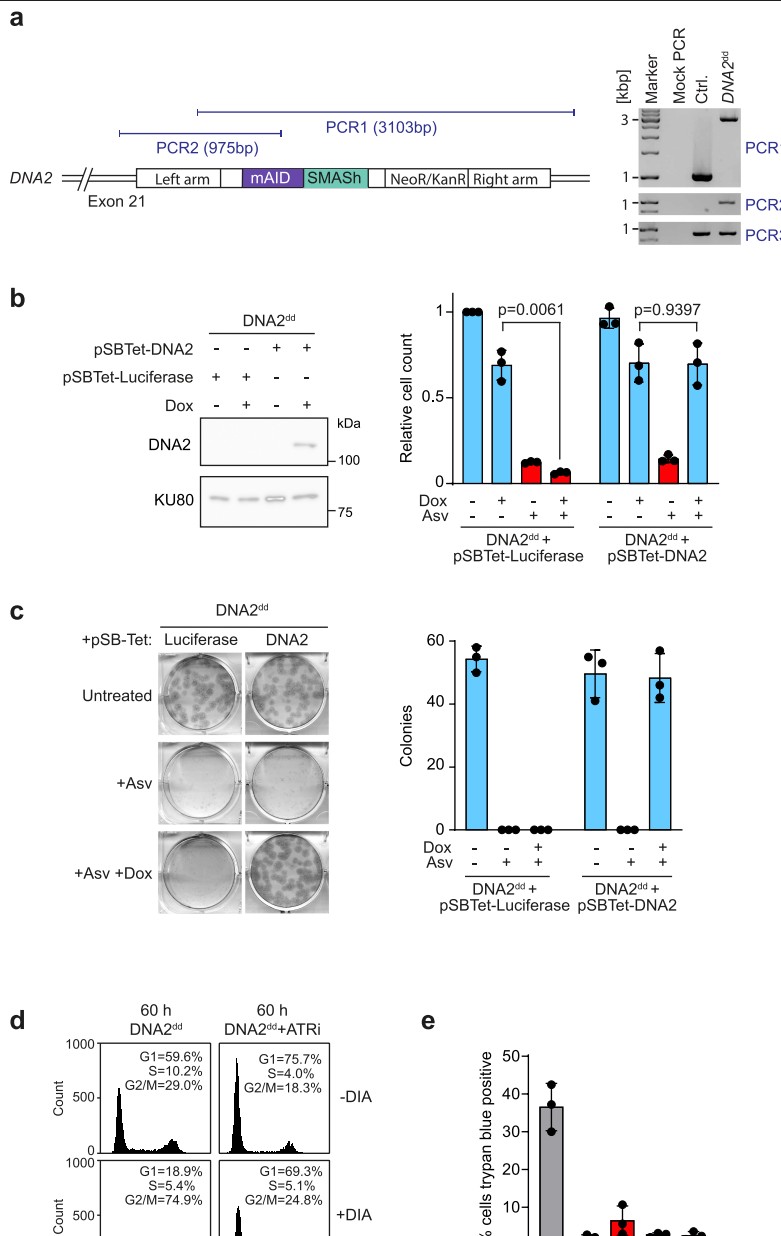

**Extended Data Fig. 4 | Construction and complementation of DNA2<sup>dd</sup> cells.**
**a**, Schematic of the double-degron construct knock-in at endogenous *DNA2*
with diagnostic PCR (representative of two independent experiments) on
genomic DNA from RPE-1 *OsTIR1* (Ctrl.) cells and RPE-1 *OsTIR1 DNA2<sup>dd</sup>* cells
confirming correct integration. PCR3 amplifies an unrelated locus. Right arm
and Left arm indicate the homologous regions used for gene targeting.
**b**, Complementation of proliferation defects caused by DNA2<sup>dd</sup> degradation.
Sleeping Beauty plasmids (pSB) allowing doxycycline (Dox)-inducible expression
of either luciferase or DNA2 (as shown by a western blot representative of three
independent experiments) were integrated into RPE-1 *OsTIR1 DNA2<sup>dd</sup>* cells.
500 cells were plated and grown in the presence of Dox (1 μg/ml) and Asv (3 μM)

for 6 days prior to fixation, staining, and counting nuclei. Quantification of the
results shows the mean ± s.d. of *n* = 3 independent experiments. Data normalised
to untreated conditions. Statistical analysis by two-sided t-test with Welch's
correction. **c**, Complementation of colony forming defects caused by DNA2<sup>dd</sup>
degradation. Cell lines and drug treatment as in panel **b**. Representative image
and quantification showing mean ± s.d. of *n* = 3 independent experiments.
**d**, Flow cytometric analysis of RPE-1 *OsTIR1 DNA2<sup>dd</sup>* following 60 h treatment ±
ATRi (VE-821; 1 μM) ± DIA (*n* = 1). **e**, Cell viability of RPE-1 *OsTIR1 DNA2<sup>dd</sup>* cells
treated as in panel **d**, as measured by trypan blue exclusion. Treatment with
etoposide (50 μM) for 60 h to induce cell death serves as control. Quantification
of the results shows mean ± s.d. of *n* = 3 independent experiments.

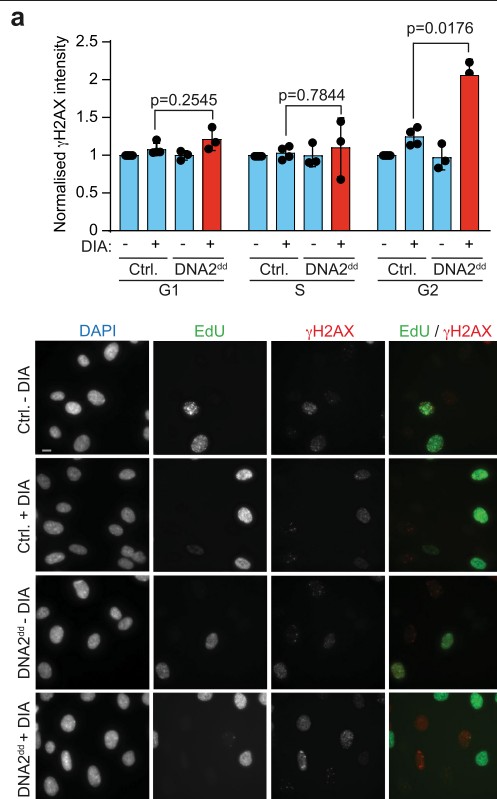

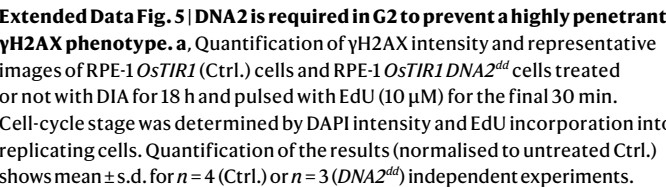

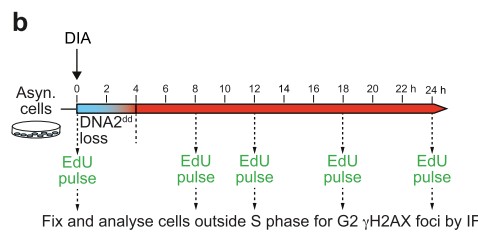

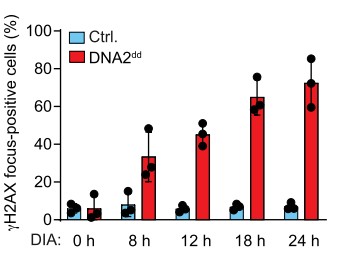

**Extended Data Fig. 5 | DNA2 is required in G2 to prevent a highly penetrant γH2AX phenotype. a**, Quantification of γH2AX intensity and representative images of RPE-1 *OsTIR1* (Ctrl.) cells and RPE-1 *OsTIR1 DNA2^dd* cells treated or not with DIA for 18 h and pulsed with EdU (10 μM) for the final 30 min. Cell-cycle stage was determined by DAPI intensity and EdU incorporation into replicating cells. Quantification of the results (normalised to untreated Ctrl.) shows mean ± s.d. for *n* = 4 (Ctrl.) or *n* = 3 (*DNA2^dd*) independent experiments.

Significance determined by two-sided t-test with Welch's correction. **b**, Experimental set-up and immunofluorescence-based analysis of γH2AX foci in RPE-1 *OsTIR1* (Ctrl.) cells and RPE-1 *OsTIR1 DNA2^dd* cells over 24 h following DIA-induced DNA2^dd degradation. A 30-min EdU pulse at each indicated timepoint was used to identify non-S phase, γH2AX focus-positive cells. Data shows mean ± s.d. for *n* = 3 independent experiments.

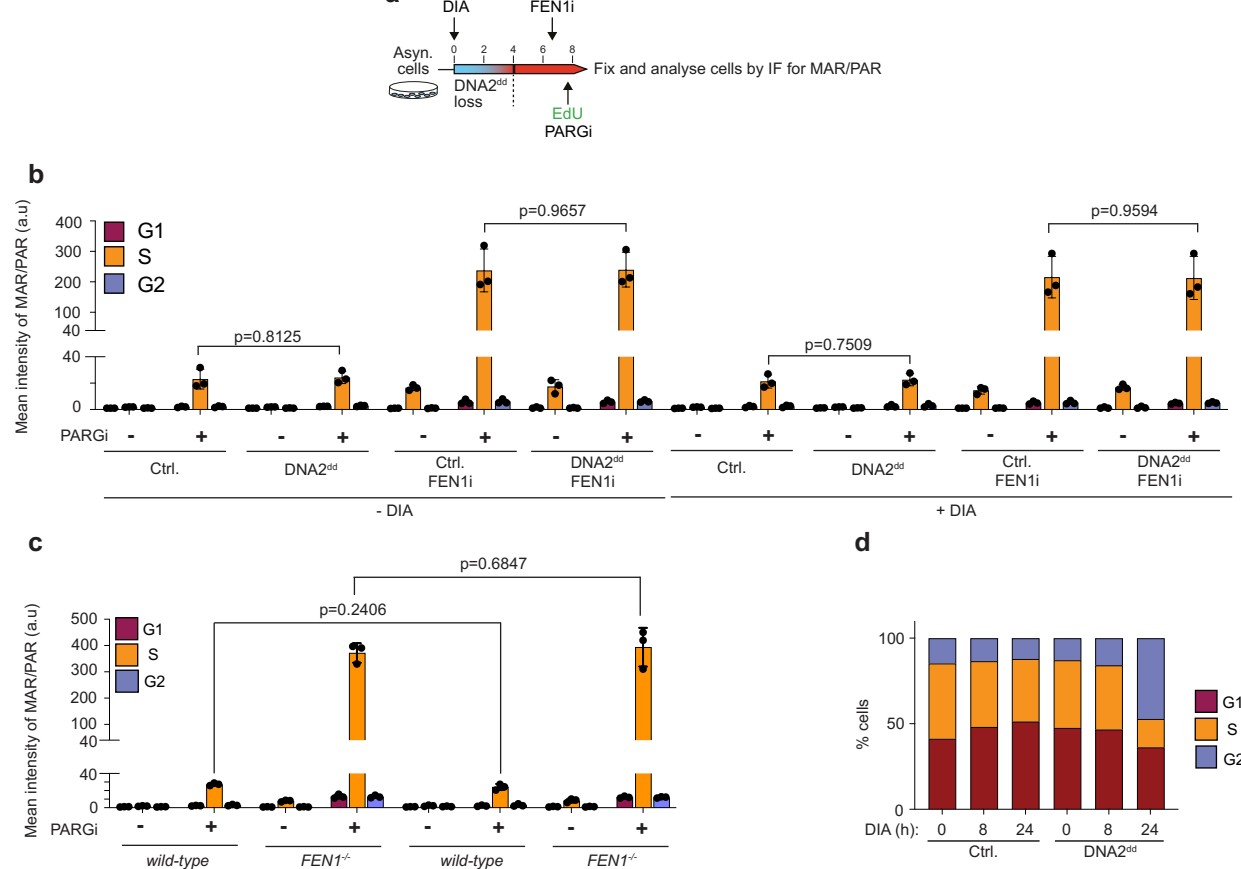

**Extended Data Fig. 6 | Loss of DNA2 does not result in perturbed canonical Okazaki fragment processing. a,** Experimental set-up for the assessment of ADP-ribose (MAR/PAR) in RPE-1 *OsTIR1 DNA2^dd^* cells treated with FEN1i (20 μM; added 90 min before IF sample preparation) and PARGi (10 μM; added 30 min before IF sample preparation) as indicated. **b,** RPE-1 *OsTIR1* control (Ctrl.) cells and RPE-1 *OsTIR1 DNA2^dd^* cells were treated as shown in panel **a** and assessed for MAR/PAR. The data is represented as the mean ± s.d. for *n* = 3 independent experiments. Statistical analysis by two-sided t-test with Welch's correction.

**c,** Wild-type and *FEN1^-/-^* RPE-1 cells analysed as in panel **a** showing that DIA has no effect on the MAR/PAR signals (*n* = 3 independent experiments). **d,** Cell-cycle distribution determined by EdU staining and DNA (DAPI) content of RPE-1 *OsTIR1* control (Ctrl.) cells and RPE-1 *OsTIR1 DNA2^dd^* cells treated for 0, 8, and 24 h with DIA. FEN1i and EdU + PARGi were present for the final 90 and 30 min, respectively (representative of two independent experiments). A.u., arbitrary units.

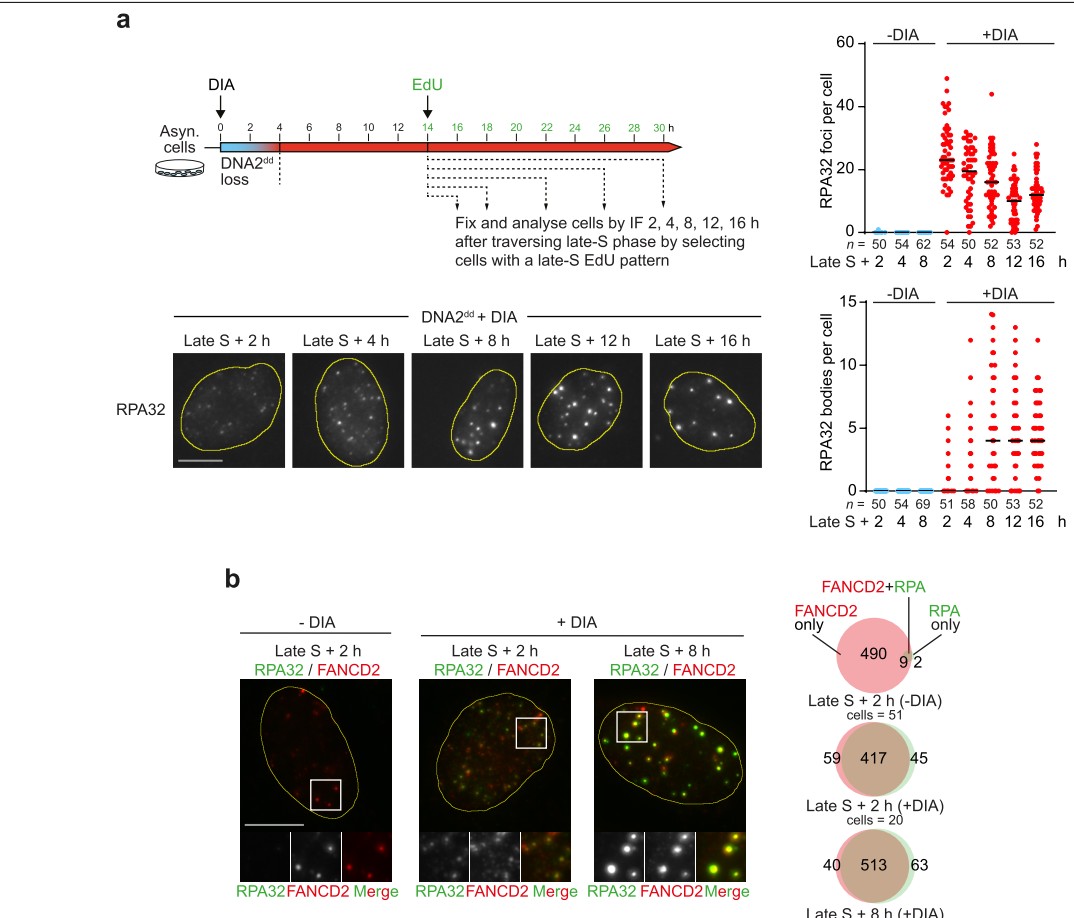

**Extended Data Fig. 7 | Persistent RPA bodies in DNA2-deprived cells and RPA colocalization with FANCD2. a**, Experimental set-up and quantification of RPA foci and bodies throughout G2 in the indicated numbers (*n*) of RPE-1 *OsTIR1 DNA2dd* cells treated or not with DIA over two independent experiments, with representative images. Bars mark the median. **b**, Images (representative of three independent experiments) of RPA and FANCD2 co-localisation and quantification of foci across the indicated numbers of RPE-1 *OsTIR1 DNA2dd* cells treated as in Fig. 3, panel **a**. Scale bars, 10 μm.

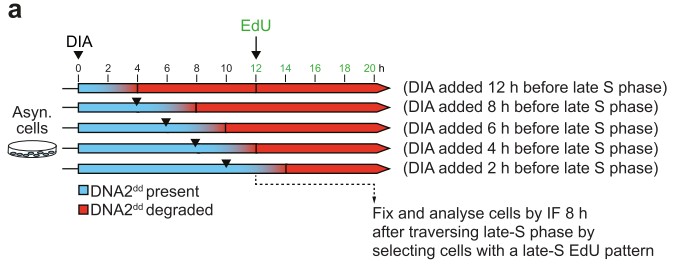

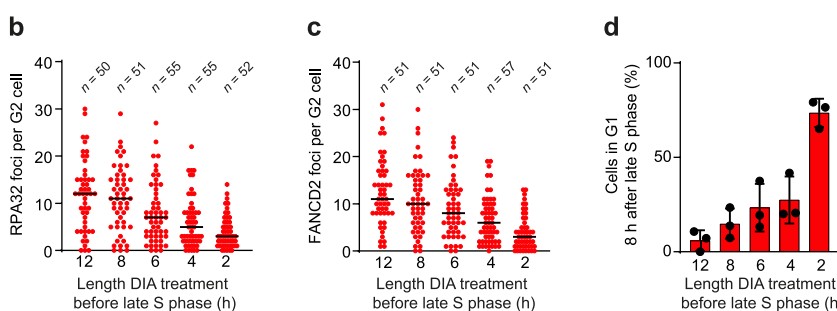

**Extended Data Fig. 8 | DNA2 is required throughout S/G2 phase.**
**a**, Experimental set-up for the detection of RPA in RPE-1 *OsTIR1 DNA2^dd* cells when DNA2^dd is degraded for different lengths of time in S phase. **b**, Number of RPA foci in the indicated numbers (*n*) of G2 RPE-1 *OsTIR1 DNA2^dd* cells over two independent experiments. Cells were treated as in panel **a** and analysed 8 h after traversing late S phase. Bars mark the median. **c**, As panel **b**, but cells analysed for FANCD2 foci. **d**, Proportion of RPE-1 *OsTIR1 DNA2^dd* cells that have progressed through mitosis into the next G1 phase 8 h after traversing late S phase following the indicated lengths of DIA treatment before late S phase. The data is represented as the mean ± s.d. for *n* = 3 independent experiments.

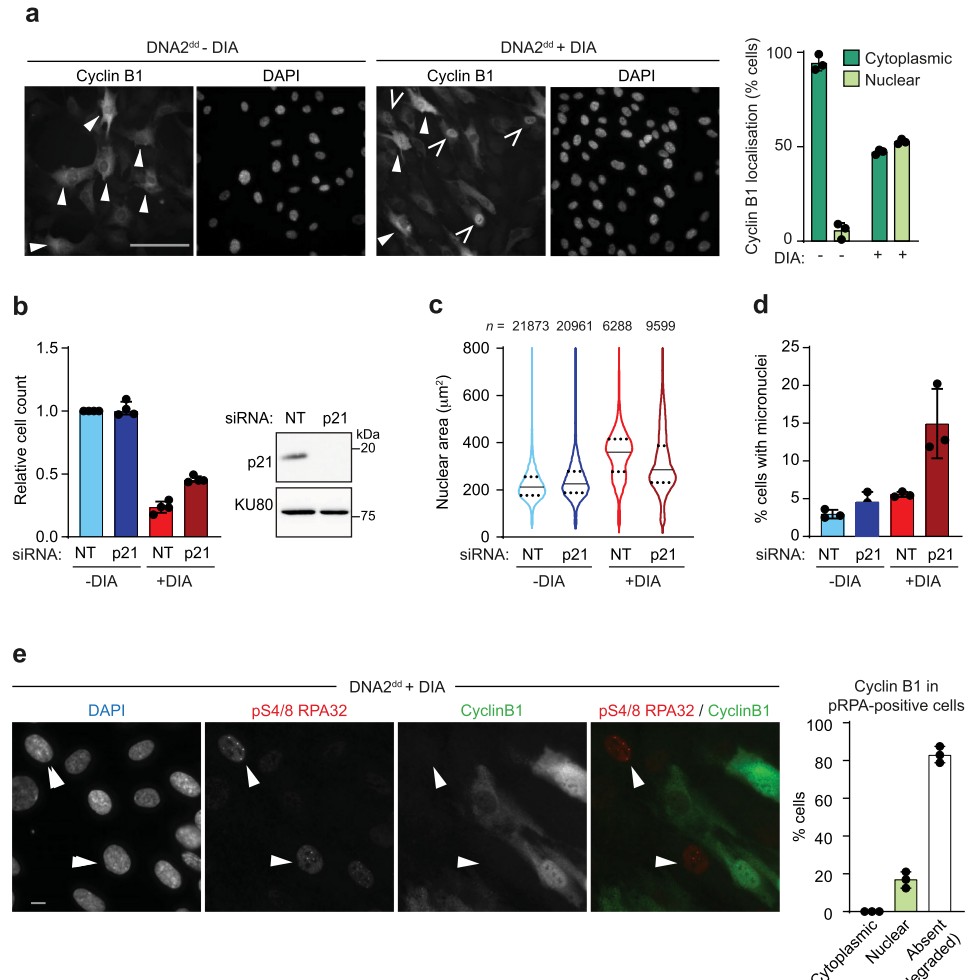

**Extended Data Fig. 9 | Commitment to senescence in DNA2-deprived cells is mediated by p21 and independent of DNA double-strand breaks.**
**a**, Cytoplasmic (closed arrowheads) and nuclear (open arrowheads) cyclin B1 localisation in RPE-1 *OsTIR1 DNA2[dd]* cells treated or not with DIA (18 h). Mean localisation frequency ± s.d. was determined for *n* = 3 independent experiments (−DIA, 255 cells; +DIA, 246 cells). Scale bar, 100 µm. **b**, Effect of p21 siRNA on cell proliferation following DNA2[dd] degradation in RPE-1 *OsTIR1 DNA2[dd]* cells (*n* = 3 independent experiments) and western blot analysis (*n* = 1) of p21 depletion. p21 siRNA was applied for 24 h before addition of DIA. After three days, nuclei were stained and counted. 3500 cells plated and data normalised to cells treated with non-targeting (NT) control siRNA only. Data presented as mean values ± s.d. **c**, Effect of p21 siRNA on the increase in nuclear area following DNA2[dd] degradation. Experimental set-up as in panel **a**, but nuclear area

measured for the indicated numbers (*n*) of cells. **d**, Cell cycle exit override in DIA-treated RPE-1 *OsTIR1 DNA2[dd]* cells induces micronuclei formation. Experiment carried out as in panel **b**, but micronuclei-positive cells identified and scored. Data presented as mean values ± s.d. for *n* = 3 independent experiments. **e**, Determination of the stage of DIA-induced cell-cycle exit in RPE-1 *OsTIR1 DNA2[dd]* cells at which phosphorylation of RPA (pS4/8 on RPA32; a proxy measure for DNA breakage) is detected. Cells were treated 18 h with DIA and stained for cyclin B1. pRPA-positive cells (indicated by white arrowheads) were assessed for cyclin B1. Approximately 5% of cells were positive for pRPA, and cyclin B1 was mostly absent (degraded) in these cells, indicating that any DNA double-strand break formation occurs after commitment to senescence (*n* = 3 independent experiments). Data presented as mean values ± s.d. Scale bar, 10 µm.

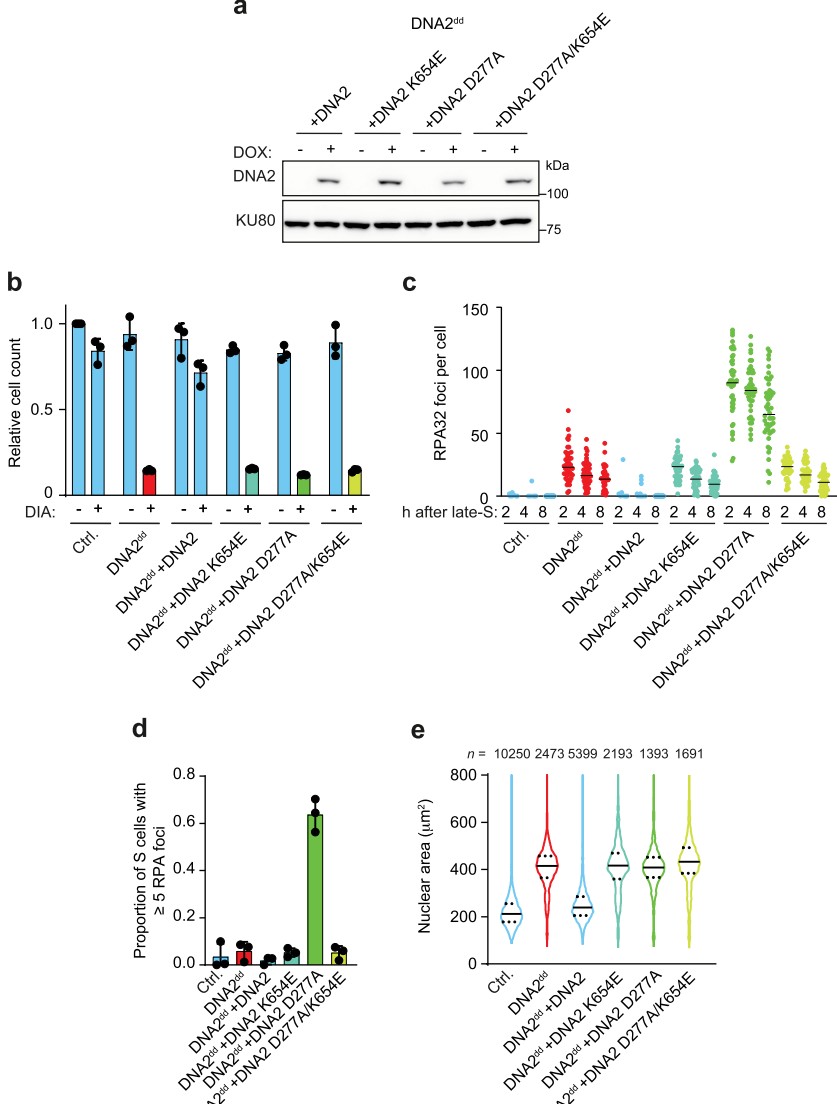

**Extended Data Fig. 10 | The nuclease and helicase activities of DNA2 jointly safeguard cell proliferation and unrestrained helicase activity of nuclease-dead DNA2 D277A causes genotoxic stress in S and G2 phase. a**, Western blot analysis (*n* = 1) on lysates from RPE-1 *OsTIR1 DNA2 dd* cells upon doxycycline-induced (1 μg/ml) expression of the indicated wild-type and mutant versions of DNA2 for 8 h. KU80, loading control. **b**, Proliferation of RPE-1 *OsTIR1 DNA2 dd* cells upon expression of wild type or mutant DNA2 as indicated in the presence or absence of DIA over 3 days. 1500 cells plated. Data normalised to untreated RPE-1 *OsTIR1* control (Ctrl.) cells and presented as mean values ± s.d. (*n* = 3 independent experiments). **c**, Quantification of RPA foci 2, 4, and 8 h after late S phase upon treatment of the indicated cells as shown in Fig. 3a (*n* = 50 cells over two independent experiments at each timepoint). Note the induction of

supernumerary RPA foci associated with the expression of nuclease-dead DNA2 D277A. Bars indicate the median. **d**, Analysis of RPA foci in RPE-1 *OsTIR1 DNA2 dd* cells complemented or not with doxycycline-induced DNA2 versions as indicated after treatment with DIA for 16 h and pulsed with EdU (10 μM) for the final 60 min. Cell-cycle stage was determined by DAPI intensity and EdU incorporation into replicating cells. Data is presented as mean values ± s.d. (*n* = 3 independent experiments). **e**, Nuclear area measurements for the indicated numbers (*n*) of RPE-1 *OsTIR1* control (Ctrl.) and RPE-1 *OsTIR1 DNA2 dd* cells (representative of three independent experiments). Cells were complemented as indicated and treated with DIA for three days. Median and quartiles marked.

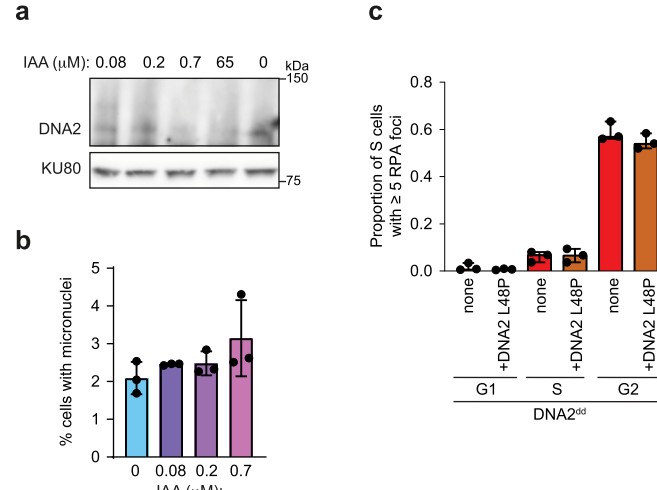

**a**

IAA (μM): 0.08  0.2  0.7  65  0    kDa

DNA2                              ―150

KU80                              ―75

**b**

% cells with micronuclei

5
4
3
2
1
0

IAA (μM):  0  0.08  0.2  0.7

**c**

Proportion of S cells
with ≥ 5 RPA foci

0.8
0.6
0.4
0.2
0.0

none  +DNA2 L48P | none  +DNA2 L48P | none  +DNA2 L48P

G1 | S | G2

DNA2$^{dd}$

**Extended Data Fig. 11 | Analysis of partial degradation of DNA2dd and expression of patient-derived DNA2 mutant L48P. a**, DNA2 immunoprecipitation and western blot (representative of two independent experiments) on lysates from RPE-1 *OsTIR1 DNA2$^{dd}$* cells treated with doxycycline (1 μg/ml) and the indicated concentrations of IAA for 24 h. KU80, input control. **b**, RPE-1 *OsTIR1 DNA2$^{dd}$* cells treated as in panel **a** and analysed for micronuclei after three days (*n* = 3 technical replicates). Data presented as mean values ± s.d. **c**, Analysis of RPA foci in RPE-1 *OsTIR1 DNA2$^{dd}$* cells complemented or not with doxycycline-induced DNA2 L48P as indicated after treatment with DIA for 16 h and pulsed with EdU (10 μM) for the final 60 min. Cell-cycle stage was determined by DAPI intensity and EdU incorporation into replicating cells. Data is presented as mean values ± s.d. (*n* = 3 independent experiments).

# Reporting Summary

## Statistics

For all statistical analyses, confirm that the following items are present in the figure legend, table legend, main text, or Methods section.

| n/a | Confirmed | |
|---|---|---|
| ☐ | ☒ | The exact sample size (*n*) for each experimental group/condition, given as a discrete number and unit of measurement |
| ☐ | ☒ | A statement on whether measurements were taken from distinct samples or whether the same sample was measured repeatedly |
| ☐ | ☒ | The statistical test(s) used AND whether they are one- or two-sided<br>*Only common tests should be described solely by name; describe more complex techniques in the Methods section.* |
| ☒ | ☐ | A description of all covariates tested |
| ☐ | ☒ | A description of any assumptions or corrections, such as tests of normality and adjustment for multiple comparisons |
| ☐ | ☒ | A full description of the statistical parameters including central tendency (e.g. means) or other basic estimates (e.g. regression coefficient) AND variation (e.g. standard deviation) or associated estimates of uncertainty (e.g. confidence intervals) |
| ☐ | ☒ | For null hypothesis testing, the test statistic (e.g. *F*, *t*, *r*) with confidence intervals, effect sizes, degrees of freedom and *P* value noted<br>*Give P values as exact values whenever suitable.* |
| ☒ | ☐ | For Bayesian analysis, information on the choice of priors and Markov chain Monte Carlo settings |
| ☒ | ☐ | For hierarchical and complex designs, identification of the appropriate level for tests and full reporting of outcomes |
| ☒ | ☐ | Estimates of effect sizes (e.g. Cohen's *d*, Pearson's *r*), indicating how they were calculated |

*Our web collection on statistics for biologists contains articles on many of the points above.*

## Software and code

Policy information about availability of computer code

| Data collection | ImageQuant (Cytiva) western blot imaging system software (v12)<br>Zen Blue (2.6)<br>Harmony<br>BD Accuri C6 (v1)<br>Olympus ScanR (3.2.0) |
|---|---|
| Data analysis | Prism (v 10.4, GraphPad software)<br>Zen Blue (2.6)<br>Harmony (v4.9)<br>Fiji (Image J v.1.53t)<br>BD Accuri C6 (v1)<br>Olympus ScanR (3.2.0) |

For manuscripts utilizing custom algorithms or software that are central to the research but not yet described in published literature, software must be made available to editors and reviewers. We strongly encourage code deposition in a community repository (e.g. GitHub). See the Nature Portfolio guidelines for submitting code & software for further information.

## Data

Policy information about availability of data

All manuscripts must include a data availability statement. This statement should provide the following information, where applicable:

- Accession codes, unique identifiers, or web links for publicly available datasets
- A description of any restrictions on data availability
- For clinical datasets or third party data, please ensure that the statement adheres to our policy

> All data supporting the findings of this study are available within the paper and accompanying Source Data. For gel source data, see Supplementary Figure 1.

## Research involving human participants, their data, or biological material

Policy information about studies with human participants or human data. See also policy information about sex, gender (identity/presentation), and sexual orientation and race, ethnicity and racism.

| | |
|---|---|
| Reporting on sex and gender | N/A |
| Reporting on race, ethnicity, or other socially relevant groupings | N/A |
| Population characteristics | N/A |
| Recruitment | N/A |
| Ethics oversight | N/A |

Note that full information on the approval of the study protocol must also be provided in the manuscript.

# Field-specific reporting

Please select the one below that is the best fit for your research. If you are not sure, read the appropriate sections before making your selection.

☒ Life sciences          ☐ Behavioural & social sciences          ☐ Ecological, evolutionary & environmental sciences

For a reference copy of the document with all sections, see nature.com/documents/nr-reporting-summary-flat.pdf

# Life sciences study design

All studies must disclose on these points even when the disclosure is negative.

| | |
|---|---|
| Sample size | No statistics were carried out to determine sample size. Sample sizes based on common practice and experimental experience. All sample sizes indicated in legends and Source Data. |
| Data exclusions | No data points excluded. |
| Replication | All replication attempts provided results as those described. Experimental replicates described in legends. |
| Randomization | We use genetically defined cell lines cultured together and without assignment into groups. |
| Blinding | Several analyses are automated so do not require blinding. Manual analyses were duplictaed using automated analysis where possible. Investiagtor blinding was not perfromed as differences in staining made sample identification possible without labelling, making blinding ineffective. |

# Reporting for specific materials, systems and methods

We require information from authors about some types of materials, experimental systems and methods used in many studies. Here, indicate whether each material, system or method listed is relevant to your study. If you are not sure if a list item applies to your research, read the appropriate section before selecting a response.

## Materials & experimental systems

| n/a | Involved in the study |
|---|---|
| ☐ | ☒ Antibodies |
| ☐ | ☒ Eukaryotic cell lines |
| ☒ | ☐ Palaeontology and archaeology |
| ☒ | ☐ Animals and other organisms |
| ☒ | ☐ Clinical data |
| ☒ | ☐ Dual use research of concern |
| ☒ | ☐ Plants |

## Methods

| n/a | Involved in the study |
|---|---|
| ☒ | ☐ ChIP-seq |
| ☐ | ☒ Flow cytometry |
| ☒ | ☐ MRI-based neuroimaging |

# Antibodies

| Antibodies used | Anti-CHK1 G4 (Santa Cruz, sc-8408) \| Anti-phospho-CHK1 (S345) 133D3 (Cell Signalling Technology, 2346)<br>Anti-KU80 EPR3468 (Abcam, ab80592) \| Anti-p21 EA10 (Merck, OP64)<br>Anti-DNA2 antibody raised against an N-terminal immunogen (Proteintech, 21599-1-AP)<br>Anti-DNA2 antibody raised against a C-terminal immunogen (Abcam, Ab96488)<br>Anti-POLD3 (Abnova, H00010714-M0) \| Anti-FBH1 (Abcam, 2353C1a)<br>Anti-SMARCAL1 (Santa Cruz Biotechnology, SC-376377) \| Anti-HLTF (Proteintech, 14286-1-AP)<br>Anti-ZRANB3 (Proteintech, 23111-1-AP) \| Anti-RPA32 9H8 (Genetex, GTX22175)<br>Anti-RPA70 (Abcam, ab97338) \| Anti-phospho-RPA32 (S4/8) (Bethyl Laboratories, A300-245A)<br>Anti-RAD51 Ab-1 (EMD Millipore, PC130) \| Anti-FANCD2 (Novus Biologicals, NB100-182)<br>Anti-(gamma)H2AX (S139) JBW301 (Sigma-Aldrich, 05-636) \| Anti-cyclin B1 (BD Biosciences, 610220)<br>Anti-BrdU (BD Biosciences, 555627) \| Anti-MAR/PAR (Cell Signaling Technology, 83732)<br>Donkey anti-mouse IgG AlexaFluor-488 (Thermo Fisher Scientific A-21202)<br>Donkey anti-rabbit IgG AlexaFluor-555 Thermo Fisher Scientific A-31572)<br>Donkey anti-rabbit IgG Alexa Fluor 488 (Thermo Fisher Scientific A-21206)<br>Goat anti-mouse IgG AlexaFluor-647 (Thermo Fisher Scientific A-21235)<br>HRP-linked anti rabbit or anti-mouse IgG (Cell Signaling Technology, 7074 and 7076) |
|---|---|
| Validation | Anti-CHK-1  is validated by siRNA by the manufacturer.<br>Anti-phospho-CHK1 (S345) was experimentally validated using camptothecin treatment, and UV treatment by the manufacturer.<br>Anti-(gamma)H2AX (S139) JBW301 validated experimentally using etoposide and by the manufacturer by the same method.<br>Anti-phospho-RPA32 validated experimentally using camptothecin, and by the manufacturer using etoposide.<br>Anti-DNA2 antibody raised against a C-terminal immunogen, Anti-p21, Anti-POLD3, Anti-FBH1, Anti-SMARCAL1, Anti-HLTF, Anti-ZRANB3, all validated experimentally by disappearance of band on western blot after siRNA .<br>Anti-KU80 validated by immunoprecipitation by the manufacturer.<br>Anti-cyclin B1 validated by manufacturer and by PMID: 18195732.<br>Anti-RAD51 validated in by signal inhibition by Rad51 inhibitors, PMID: 28076755.<br>Anti-FANCD2 validated by siRNA/western blot by the manufacturer.<br>Anti-MAR/PAR validated using relevant genotype/inhibitor controls, PMID: 35332322.<br>Anti-DNA2 antibody raised against an N-terminal immunogen validated by the manufacturer and by us through depletion of mAID-immunoprecipitated DNA2dd.<br>Anti-RPA70 cross-referenced with anti-RPA32 for presence of RPA heterotrimer and vice versa; and tested experimentally using camptothecin.<br>Anti-BrdU was tested experimentally for specificity with and without the addition of BrdU. |

# Eukaryotic cell lines

Policy information about cell lines and Sex and Gender in Research

| Cell line source(s) | All yeast cell lines were derived from standard laboratory strains of S. pombe and their genotypes are fully described in Supplementary Table S1. Human cell lines were derived from RPE-1 as detailed in the methods. RPE-1 cells were aquired from ATCC (cat. CRL-4000). |
|---|---|
| Authentication | Genotypes were verified using genetic markers and sequencing of all loci modified in this study. Genetic modifications in human cell lines verified by PCR. |
| Mycoplasma contamination | Cell lines were confirmed mycoplasma negative every 3 months. |
| Commonly misidentified lines<br>(See ICLAC register) | No commonly misidentified lines used. |

# Plants

| | |
|---|---|
| Seed stocks | *Report on the source of all seed stocks or other plant material used. If applicable, state the seed stock centre and catalogue number. If plant specimens were collected from the field, describe the collection location, date and sampling procedures.* |
| Novel plant genotypes | *Describe the methods by which all novel plant genotypes were produced. This includes those generated by transgenic approaches, gene editing, chemical/radiation-based mutagenesis and hybridization. For transgenic lines, describe the transformation method, the number of independent lines analyzed and the generation upon which experiments were performed. For gene-edited lines, describe the editor used, the endogenous sequence targeted for editing, the targeting guide RNA sequence (if applicable) and how the editor was applied.* |
| Authentication | *Describe any authentication procedures for each seed stock used or novel genotype generated. Describe any experiments used to assess the effect of a mutation and, where applicable, how potential secondary effects (e.g. second site T-DNA insertions, mosiacism, off-target gene editing) were examined.* |

# Flow Cytometry

## Plots

Confirm that:

☒ The axis labels state the marker and fluorochrome used (e.g. CD4-FITC).

☒ The axis scales are clearly visible. Include numbers along axes only for bottom left plot of group (a 'group' is an analysis of identical markers).

☒ All plots are contour plots with outliers or pseudocolor plots.

☒ A numerical value for number of cells or percentage (with statistics) is provided.

## Methodology

| | |
|---|---|
| Sample preparation | RPE-1 were fixed in 70% ethanol at 4 °C overnight and stained with FxCycle PI/RNase Staining Solution (Invitrogen, F10797) according to the manufacturer's instructions |
| Instrument | Accuri C6 Flow cytometer (BD Biosciences) |
| Software | BD Accuri C6 software |
| Cell population abundance | For each sample, 10000 events were collected. Examples of populations in each gate are shown in Supplemtary Fig. 2. |
| Gating strategy | Gating was applied to exclude (1) cell debris and (2) doublets; subsequently uniform gating was used to distinguish G1, S, and G2 cells based on PI intensity. Gating strategy described in Supplementary Fig. 2. |

☒ Tick this box to confirm that a figure exemplifying the gating strategy is provided in the Supplementary Information.

