## [Peer Review File · Nature]

DNA2 enables growth by restricting recombination-restarted replication

Corresponding Author: Professor Ulrich Rass

Version 0:

Reviewer comments:

Referee #1

(Remarks to the Author)

Hudson and colleagues investigated the possible involvement of DNA2 in the processing of replication intermediates and the consequences of DNA2 loss on cell fitness, particularly in human cells. They utilized a degron-based model that allows precise control of protein depletion at specific moments of the cell cycle. The ultimate goal of the authors was to provide a mechanism explaining the recent findings of DNA2 dysfunctions in genetic diseases.

The authors began their study by presenting findings from the yeast model system. They then transitioned to using the degron-based model in human cells, focusing their experiments on the consequences of DNA2 loss on DNA damage, checkpoint activation, and cell fitness. They exploited the degron system to deplete DNA2 at specific moments of the cell cycle and identified that DNA2 loss is crucial for processing unknown replication intermediates in the G2 phase, preventing recombination from completing replication. While some findings presented in this study have been previously reported in yeast (see Rossi et al., 2018), the experiments using the human degron-based model are intriguing.

Given that DNA2 is involved in various transactions at perturbed replication forks, and its roles are still not fully understood, particularly in human cell models, the study by Hudson and colleagues is novel. The potential link between altered processing/resolution of replication intermediates, DNA2 loss, and genetic diseases makes this manuscript of interest to Nature's audience.

However, in its current form, the study is interesting but still underdeveloped and requires additional experiments to further improve its novelty, especially over studies performed in yeast, and to get a more mechanistic and detailed view of the role of DNA2 in G2, as the authors claim.

In my opinion, the most important findings requiring further studies are:

- The identity of the replication intermediate DNA2 is working on. This is particularly important since DNA2 has been reported to act at reversed forks and behind the fork to remove flaps, and the authors currently provide no experiments addressing this.
- The involvement of DNA2 in G2 cells or cells that are only very delayed in S-phase. The manuscript indirectly addresses this, but the involvement of DNA2 in G2 is a novel and main claim of the work. In the current form, most experimental data are inferred from phenotypes but no direct assessment of DNA2 loading in G2 has been performed. Additionally, the degron model could allow the authors to perform on-off experiments to determine if DNA2 has distinct functions in S-phase and G2-phase, associated with distinct phenotypes, also differentiating between the "physiological" role during unperturbed replication and that carried out in response to replication arrest.
- The correlation between DNA2 loss, persisting and alternative processing of intermediates, and loss of cell fitness/senescence. This part of the work is the most novel and outstanding but it is just minimally investigated.

While I acknowledge the challenges in fully developing the third point, developing the first two is feasible for the authors' lab and is critical to providing the full mechanism of action and not just the consequences of DNA2 loss, which in my opinion is critical for Nature's audience.

Below, I provide the authors with specific comments and suggestions:

Fig. 1: In light of the additional work the authors may want to add, I suggest eliminating this part of the study. It is merely incremental over what we know from yeast studies and not so much useful, in my opinion, as the bulk of the study is performed in human cells.

Fig. 2: Overall, findings shown in human cells are reminiscent of those reported by Rossi and colleagues in yeast (Nat. Commun., 2018). It would be important to determine which of DNA2's two potential enzymatic activities is more critical. Previous work on DNA2's role at perturbed replication forks implicated its nuclease activity, while Fig. 1 suggests a role for the helicase.

In Fig. 2d the authors showed the consequences of DNA2 degradation at 24h of DIA treatment. It would be interesting to check what happens at 60h to match the live-cell imaging data of Fig. 2f.

It would be interesting to back up long-term depletion of DNA2 with other methods. Given the long DIA treatment, I think that good siRNA or inhibitors might be an orthogonal method to check if the phenotype is somehow recapitulated.

What happens to cells treated with DIA and ATRi? I would expect they die. It is important to follow this phenotype as 60h should be enough to induce cell death. The absence of cell death would represent, per se, a nice result.

Fig. 3: The authors claim that DNA2 processes an unknown intermediate. The identity of this intermediate has not been investigated but it should be as it also helps assess why ATR gets activated. Is the DNA2dd phenotype dependent on fork reversal? Which proteins known to participate in fork reversal work upstream of DNA2? Previous work identified that DNA2 works after FBH1, at least under replication perturbation. What happens after replication fork arrest? Is DNA2's involvement different under normal or perturbed replication? The authors implicated DNA2 in G2 for processing persisting intermediates. This claim is based on experiments depleting DNA2 at specific moments of the cell cycle. However, the protocol used has shortcomings. For instance, how do the authors exclude that DNA2 is working on cells in very late S or very delayed in S-phase? Why not also use shorter DIA treatment given that DNA2 goes down completely after 4h? Performing parallel experiments with some enrichment strategy might be better. If I am not wrong, RPE1 cells can be synchronized using serum starvation. Alternatively, QIBC could be performed to determine the cell cycle stage in which cells experience DNA damage, RPA foci, and so on. This would help a lot in each of the IF experiments shown in this study.

Another critical point is whether DNA2 is recruited differently at the fork, behind the work in S-phase, or G2 cells. Orthogonal approaches can be used to determine when DNA2's function is needed.

Fig. 4: The authors claim that RAD51-dependent recombination is required to replicate intermediates accumulating in the absence of DNA2 in G2. This is interesting but is similar to findings we might expect from studies in yeast if it remains underdeveloped. For instance, the authors showed that EdU labelling can be observed at RPA foci and somehow suggest that this incorporation occurs downstream recombination-based replication. However, while loss of EdU labelling when RAD51 nucleofilament formation is inhibited can be consistent with the hypothesis, I did not understand why RPA foci should be reduced as well. I would expect that ssDNA is covered by RAD51. Perhaps labelling of parental DNA and evaluation of the presence of ssDNA directly might help reinforce this data. It is also unclear if what the authors identified differs from MiDaS, for example. Experiments to differentiate between the two should be evaluated, or at least this should be discussed.

Fig. 5: It would be useful to add an explanation here for the use of the double degron to evaluate whether the simpler mAID system cannot be used or cannot mimic hypomorphic mutations as they occur in genetic diseases. The rationale for the experiment shown in Fig. 5d is unclear. It seems expected that RPA foci are observed in S/G2 cells and in cells progressing into mitosis, given the other results shown in Fig. 5 and 3. Perhaps this experiment could be moved to the supplementary figures. If the main message of this figure is that cells depleted of DNA2 in G2 do not recover and are marked for senescence, more compelling experiments analyzing other readouts of senescence should be added. If the authors' hypothesis is that senescence (or quiescence) directly derives from the cell cycle arrest shown in Fig. 2f, then the study should contain something performed with ATRi or other genetic strategy to affect checkpoint signalling from the still unknown intermediate left unprocessed by DNA2 depletion.

In general, the authors should add more compelling data on this critical point. In my opinion, this might represent the key finding of the story and the most intriguing one perhaps.

As the cell model used here is not a primary one, the authors might want to exploit other strategies to interfere with DNA2 function (inhibitors?) or, as an alternative strategy, they might evaluate to investigate if DNA2 loss affects senescence induced by DNA-damaging agents.

Overall, the data are well presented with adequate reference to statistical methods and data analyses. The manuscript is well written and key bibliography is properly mentioned.

Referee #2

(Remarks to the Author)

DNA2 is a cell-essential nuclease/helicase that is important for genome stability. DNA2 deficiency causes undergrowth syndromes and DNA2 is commonly expressed at higher levels in cancer, so understanding its cellular role is not only of fundamental importance, but also relevant to human disease. DNA2 has a number of proposed biological functions, but in spite of decades of research the function that is essential for continued proliferation has remained elusive. Here, Hudson, Rass and colleagues build on previous work from the Rass lab in *S. cerevisiae*, and propose that the most important function of DNA2 is to suppress HoRRer and checkpoint activation at stalled replication forks in both *S. pombe* and human cells.

Experiments in *S. pombe* make use of a helicase mutant allele that is viable but has increased checkpoint activation and decreased survival, as well as increased recombination at an inducible replication barrier. Effects on recombination and survival, but not checkpoint activation, are rescued by a *Cdc27* allele with a C-terminal truncation (*cdc27-D1*) that reduces D-loop DNA synthesis required for HoRRer. Pu-Seq is used to show increased Pol delta DNA synthesis (which implies increased HoRRer), which is partially suppressed by *cdc27-D1*.

Experiments in human cells make use of rapid degradation of endogenously degron-tagged DNA2 paired with EdU labelling and measuring DAPI intensity to identify cells at different stages of the cell cycle, followed by a range of fluorescence microscopy-based analyses to determine the consequences. DNA2 depletion causes cells to accumulate in G2/M, leading to increased gH2AX, FANCD2 and RPA foci. Reduced RPA foci intensity and reduced DNA synthesis at RPA foci upon RAD51 inhibition suggests that this is due to excessive HoRRer at stalled replication forks upon loss of DNA2. Finally, partial depletion of DNA2 (a situation that mimics the genetic cause of human undergrowth syndromes) is shown to cause increased senescence as measured by CyclinB1 nuclear translocation and increased nuclear area, with the latter reduced upon p21 knockdown at the cost of increased formation of micronuclei. This provides a plausible model for how partial loss of DNA2 function in humans causes reduced growth.

Altogether this is an interesting and important piece of work that establishes the essential function of DNA2, relevant to human disease. However, I would recommend strengthening the paper by addressing the following points.

1. Clarify EdU labelling experiments used to identify cells at time points after late S phase

Can the authors please clarify the experimental setup for Figs. 3b-e and 4a,b and ED Fig. 4? Firstly, it is not clearly indicated whether a pulse of EdU was administered or if EdU was present from $t=0$. Secondly, I understand that patterns of EdU foci can help identify late S-phase cells. However, because this requires fixation, I do not understand how cells that were in late S at the time of labelling can then be followed after that as the text seems to imply (lines 156-160 and 180-182). In contrast to what the main text implies, in the figure diagrams it is indicated that late S-phase cells are identified at the different time points, which obviously can be done, but it changes the meaning of the results. I assume that the authors indeed mean that the cells that WERE in late S phase at the time of labelling are identified at the different time points. For reviewing the paper (in spite of this confusion), I made the assumption that it is indeed possible to identify pulse-labelled cells that were in late S at the time and that those cells were then identified at different time points after that for further staining and analysis. Textual changes and changes to the diagrams should ensure other readers do not get confused and know how to interpret the experiments shown.

2. Perform additional experiments to provide more detail for the molecular mechanism by which DNA2 suppresses HoRRer, prevents checkpoint activation and promotes RF restart

- *Cdc27-D1* rescues reduced viability of *dna2-2* (Fig. 1g), but does not rescue checkpoint activation (as measured by cell length increase; Fig. 1f): relative to WT there is increased cell length for *dna2-2*, *cdc27-D1* as well as the double mutant. This seems to be somewhat contradictory. Is checkpoint activation for *cdc27-D1* due to non-HoRRer functions of *Cdc27*, or does it say something more about the details of the molecular mechanism of checkpoint activation vs cell death? Are there other mutants (e.g. *Pfh1-m21/mt**, PMID: 30667359) that suppress HoRRer that can give further insight?

- All *S. pombe* experiments make use of the *dna2-2* helicase mutant, but nuclease and not helicase activity of DNA2 has previously been shown to be necessary to prevent RF reversal (ref 2) and to promote restart (ref 28). Do the Fig. 1 experiments using nuclease-defective instead of helicase-defective DNA2 give similar or different results?

- How does DNA2 suppress HoRRer in human cells? Is the helicase or nuclease function (or both) essential for DNA2 suppressing HoRRer, and how does this relate to findings from ref 28? The setup as presented in ED Fig. 2 should allow complementation with helicase- and nuclease-defective DNA2 to address this. Beyond establishing more detail for the molecular mechanism, this is important for two reasons. Firstly, in addition to a strong reduction in DNA2 levels in most cases of human undergrowth syndromes associated with DNA2 mutations, a likely helicase-defective mutant has been identified in primordial dwarfism (ref 6) and a point mutation that may affect the nuclease domain has been identified in an RTS-like syndrome (ref 7). The latter includes poikiloderma, with DNA repair defects a common genetic cause of this skin condition, which may be relevant. Secondly, it could help establish if DNA2 inhibitors to treat cancer should target helicase and/or nuclease activities.

3. Strengthen the link between PD-related DNA2 deficiency and senescence

The authors provide good evidence for increased senescence in cells depleted for DNA2: increased p21 levels, increased Cyclin-B1 nuclear translocation/degradation and increased nuclear size, with nuclear size increase rescued by ATRi and p21 depletion. A few clarifications and some additional work would help further strengthen the relevance of this to PD-related DNA2 deficiency.

- I suspect this may be difficult (and perhaps impossible) due to antibody sensitivity, but is there any way in which IAA concentrations can be linked to approximate remaining levels of DNA2? Or at least determine a minimum level of DNA2

required for normal proliferation?

- Is increased senescence detected in cells depleted for degron-tagged DNA2 and complemented with putative helicase- or nuclease-defective disease mutations?
- Fig. 5d, representative of how many experiments? Fig. 5f, same as for 5e (i.e. 1,000 cells for 3 independent experiments)? (Or is this 1,000 cells for one experiment or 3x 1,000 cells?) Fig. 5h, is this a single experiment?
- Note that in support of increased senescence with DNA2 deficiency, this has been shown using primary fibroblasts from one PD patient with DNA2 mutations (ref 5).

4. Provide more detail for statistical analyses

- ED Fig. 1, were these experiments performed once? Can it be shown that the partial suppression of increased Pol delta synthesis in the *dna2-2 cdc27-D1* strains is statistically significant?
- Show exact P-values for each comparison.
- Indicate statistical test performed in legends, as well as relevant n per sample and what these represent (be clear whether n is per experiment or total).
- I may be wrong, but for Fig. 1f, is a t-test really the appropriate test?

5. Other

- Re: ED Fig. 3b (line 170-174), can it really be concluded that “cells had fully completed bulk DNA replication prior to the induction of DNA2 degradation”?
- In Fig. 5i and ED Fig. 5c, micronuclei per cell are plotted. However, increased number of MN per cell can indicate a small number of cells with many MN, or more cells with (one or a few) MN. It is therefore more common to quantify and plot % cells with MN, and probably better to do so here as well.
- It would be helpful to add to the legend of ED Fig. 1 that all strains were *dna2-2*.

Referee #3

(Remarks to the Author)

In the manuscript, “DNA2 enables proliferation by restricting homologous recombination-restarted replication”, Ulrich Rass and colleagues investigate the essential function of the nuclease-helicase DNA2. DNA2 is set apart from most DNA repair proteins by its essential nature suggesting the protein will take care of a problem that is fundamental and encountered during every cell cycle. In this manuscript, the essential function of DNA2 is investigated in fission yeast and human cell lines. Using the fission yeast RTS1 replication barrier systems the authors implicate DNA2 in recombination-dependent replication restart of stalled replication forks (HoRRerR). They link this function to the essential function of DNA2, as DNA2 phenotypes were found to be suppressed by mutation of the CDC27 subunit of polymerase delta, which is critical for recombination-dependent replication restart. They then move on to experiments with non-cancerous human cell-lines, where they show that targeted degradation of DNA2 leads (i) to accumulation of RAD51-dependent replication intermediates with (ii) exposed single-stranded DNA (ssDNA) which in turn leads to (iii) ATR-dependent checkpoint hyperactivation and (iv) cell cycle exit and senescence. Lastly, they explore whether reduced levels of DNA2, as seen in genetic disease such as DNA2-linked microcephalic primordial dwarfism, triggers a similar response and find that it sporadically causes checkpoint hyperactivation and cell cycle exit, potentially explaining patient phenotypes.

This is a well-designed study with very interesting results, which will have strong impact on the field and as such would profit greatly from being published in a journal such as Nature. Currently, there are still a number of points which need support by additional experiments and also the manuscript needs improvement in some places.

Major Points:

- #1 – Fig. 1 focuses on fission yeast and shows an increase in recombination-dependent replication using the RTS1-system. While this system is elegant, this part of the paper should be better connected to the rest of the paper. A major result of Fig. 1 is that a mutation in the polymerase delta subunit *cdc27* suppresses the *dna2-2* mutant phenotype. I therefore suggest testing for suppression of the DNA2-degron phenotype by POLD2 knockdown in the human cell culture experiments.
- #2 – Fig. 2: DNA2 degradation needs better characterization, including quantification and a more detailed time-course.
- #3 – Fig. 2e shows only normalized cell proliferation. Please also show non-normalized cell proliferation to allow better judgement of effect sizes.
- #4 – Fig. 4e shows that RAD51 inhibition partially suppresses the ssDNA-accumulation phenotype observed upon DNA2 depletion. Given that the effect size is moderate, I would suggest to also test genetic knock-down of RAD51 in this context.
- #5 – In general, the manuscript would profit from a comparative analysis of the two rescue conditions (HR-inhibition vs checkpoint inhibition). Moreover, the authors had also previously shown *dna2* rescue by checkpoint inhibition in budding yeast, but these results are hardly discussed in the current paper. I think the paper would overall profit from a more detailed discussion of similarities and differences of DNA2 phenotypes in different organisms.
- #6 - Fig. 5e shows a dose-response curve for proliferation over IAA concentrations. The authors need to show information on DNA2 levels from the same experiment and show whether there is correlation of DNA2 levels and phenotype.

#7 – In the same vein, the authors claim that their DNA2-degron model mimics patient scenarios, but they do not show in the paper that DNA2 protein levels are actually similar.

#8 – The paper would profit from a discussion of similarities and differences between the recombination-dependent replication described here and MIDAS.

Minor Points:

#1 - The authors use graded shading of “DNA2 degradation” in several figure legends, but it is unclear whether (i) this corresponds to actual levels and (ii) it is not adjusted to the changing time scales in the experiment.

#2 – Abstract, “Three decades after DNA2 was first described...”: I would argue that the historical perspective is not sufficiently important for the paper to be featured this prominently in the abstract.

#3 – Sentence line 96 ff.: This sentence seems to have unnecessarily complicated grammar.

Version 1:

Reviewer comments:

Referee #1

(Remarks to the Author)

I read the revised version of the manuscript by Hudson, Appanah and colleagues, and I am very pleased by the additional work done to arrange comments and suggestions from the two other reviewers and myself. In my view, the additional experiments using DNA2 mutants, depletion of fork reversal enzymes and careful analysis of phenotypes improved the manuscript and made the authors' assumptions more solid. While I feel that an attempt to directly evaluate the relevance of fork collapse post-fork reversal in the initiation of HR would have helped authors' conclusions, I am very convinced by the claims and found the revised manuscript improved.

The revised version of the manuscript retained its logical writing and accuracy in reporting methods and statistical details. I noticed also that authors improved presentation of the statistical details in figures vis-a-vis specific comments from another reviewer.

I have no additional comments but I would suggest authors to include in the abstract some mention to the results from DNA2 disease mutants and also to mitigate the statement about the growth failure in DNA2-linked dwarfism as, even if of high relevance, authors' findings still do not provide a solid molecular foundation to this clinical phenotype.

Similarly, I would suggest changing the sentence in the discussion that refers to the same link in a way that delivers a potential explanation rather than an established connection.

Referee #2

(Remarks to the Author)

My compliments to the authors for thoroughly addressing all reviewers' comments. This is a very nice piece of work that will be of interest to Nature readers.

There is one relatively small thing I feel would be helpful to add. Although I'm absolutely convinced that it is indeed possible to identify late S-phase cells using EdU incorporation patterns, it is still not entirely clear to me from ref. 54 and the rebuttal what the exact pattern is supposed to be. Is it simply those cells that display EdU foci (and not pan-nuclear EdU staining, or absence of staining), or is the number of EdU foci also important? And to what extent is DAPI content taken into account?

Therefore, can the authors please: 1) clearly define (in their methods section) how they decided on which cells were late S-phase and which ones not; and potentially 2) provide example images of cells with late S-phase EdU incorporation patterns (the latter may not be necessary if the description is clear enough and it is simply the presence of any number of EdU foci in cells that have 4N DNA content).

Referee #3

(Remarks to the Author)

I congratulate Rass and colleagues to the revised version of their manuscript “DNA2 enables proliferation by restricting homologous recombination-restarted replication”, which has addressed all points raised in my previous review. I think it is suitable for publication in Nature.

Rebuttal Letter Nature MS 2024-10-21223

Hudson et al.: “DNA2 enables proliferation by restricting homologous recombination-restarted replication”

We thank all three referees for their assessment and enthusiastic support for our MS. **Referee 1** describes our study as “of interest for Nature’s audience”, **referee 2** highlights an “important piece of work that establishes the essential function of DNA2”, and **referee 3** commends our “interesting results, which will have strong impact on the field”. The constructive comments and suggestions we have received have been immensely helpful in guiding our experimental MS revision. Below, please find a point-by-point response to questions raised, a description of additional experimental data included in the revised version of the MS, and details of changes made to the MS text (highlighted in magenta in the MS file) and figures.

Referee #1:

Hudson and colleagues investigated the possible involvement of DNA2 in the processing of replication intermediates and the consequences of DNA2 loss on cell fitness, particularly in human cells. They utilized a degron-based model that allows precise control of protein depletion at specific moments of the cell cycle. The ultimate goal of the authors was to provide a mechanism explaining the recent findings of DNA2 dysfunctions in genetic diseases.

The authors began their study by presenting findings from the yeast model system. They then transitioned to using the degron-based model in human cells, focusing their experiments on the consequences of DNA2 loss on DNA damage, checkpoint activation, and cell fitness. They exploited the degron system to deplete DNA2 at specific moments of the cell cycle and identified that DNA2 loss is crucial for processing unknown replication intermediates in the G2 phase, preventing recombination from completing replication. While some findings presented in this study have been previously reported in yeast (see Rossi et al., 2018), the experiments using the human degron-based model are intriguing.

Given that DNA2 is involved in various transactions at perturbed replication forks, and its roles are still not fully understood, particularly in human cell models, the study by Hudson and colleagues is novel. The potential link between altered processing/resolution of replication intermediates, DNA2 loss, and genetic diseases makes this manuscript of interest to Nature's audience.

However, in its current form, the study is interesting but still underdeveloped and requires additional experiments to further improve its novelty, especially over studies performed in yeast, and to get a more mechanistic and detailed view of the role of DNA2 in G2, as the authors claim.

In my opinion, the most important findings requiring further studies are:

- The identity of the replication intermediate DNA2 is working on. This is particularly

important since DNA2 has been reported to act at reversed forks and behind the fork to remove flaps, and the authors currently provide no experiments addressing this.

Authors' response: We agree that the identity of the replication intermediates targeted by DNA2 and giving rise to RPA-ssDNA accumulations in G2 upon escaping the attention of DNA2 is a key question. Upon deprivation of DNA2, cells accumulate reversed forks (Hu et al. (2012) Cell 149:1221-32. doi: 10.1016/j.cell.2012.04.030; Thangavel S et al (2015) J Cell Biol 208:545-62. doi: 10.1083/jcb.201406100; Rossi et al (2018) Nat Commun 9:4830. doi: 10.1038/s41467-018-07378-5), and our results are consistent with a model where stalled DNA replication forks that are not processed by DNA2 become susceptible to HoRRer, thus leading to the generation of excessive RPA-ssDNA, checkpoint activation, and cellular senescence. However, Rossi and co-workers have observed ssDNA in budding yeast cells upon degradation of DNA2 and proposed “that Dna2 plays a key role in chromosome replication to remove long ssDNA flaps generated by Pol δ - and Pif1-dependent strand displacement of Okazaki fragments”. In the revised version of the MS, we have performed additional experiments to address these different possibilities directly.

First, we addressed a potential Okazaki fragment processing defect associated with DNA2 loss in collaboration with Prof. Caldecott. We now show that, in contrast to FEN1 loss or inhibition, DNA2 loss (alone or in combination with FEN1 inhibition) does not cause a perturbed Okazaki fragment maturation phenotype detected by elevated S-phase poly(ADP-ribose). Moreover, we provide new evidence that RPA-ssDNA accumulations in DNA2-deprived cells contain newly synthesized ssDNA, while DNA that is made single-stranded by strand-displacement synthesis on the lagging strand would consist of ‘old’ parental DNA.

We further followed the suggestion of **referee 1** to address the nature of the DNA intermediates that remain unresolved in the absence of DNA2 further by analysing the dependence of RPA-ssDNA accumulation in DNA2-deprived cells on fork remodeling factors. We find that depletion of the FBH1 helicase suppresses DNA2 loss phenotypes, indicating that dead-end recombination in DNA2-deprived cells occurs downstream of fork reversal, consistent with HoRRer.

These results are described in detail in the response to **specific comment (3) and (4)** below and presented in **new Extended Data Fig. 5** and **new panels c, d, g** in **Fig. 4** in the revised version of the MS. Our additional experimental findings strongly support our model for the essential role of DNA2 based on safeguarding cells from HoRRer-dependent accumulations of RPA-ssDNA at stalled replication intermediates.

- The involvement of DNA2 in G2 cells or cells that are only very delayed in S-phase. The manuscript indirectly addresses this, but the involvement of DNA2 in G2 is a novel and main claim of the work. In the current form, most experimental data are inferred from phenotypes but no direct assessment of DNA2 loading in G2 has been performed. Additionally, the degron model could allow the authors to perform on-off experiments to determine if DNA2 has distinct functions in S-phase and G2-phase, associated with distinct phenotypes, also differentiating between the “physiological” role during unperturbed replication and that carried out in response to replication arrest.

Authors' response: Additional experimental data included in the revised version of the MS strongly support that DNA2 is not associated with distinct S-phase and G2-phase phenotypes relating to the essential role of DNA2. Firstly, we do not detect evidence

supporting an S-phase defect in Okazaki fragment processing upon loss of DNA2 (please see **specific point (3e)** below). Secondly, we were able to solidify that dead-end recombination leading to RPA-ssDNA originates at stalled replication intermediates that are targeted by FBH1 (please see **specific point (3a)** below). Thirdly, we now performed experiments in the presence of exogenous replication stress (please see **specific point (3b)** below). Mild replication stress resulted in additional RPA-ssDNA accumulations in DNA2-deprived cells, consistent with an increase in HoRRer events on account of additional stalled DNA replication forks. These findings show that DNA2 critically protects cells from HoRRer-induced senescence both under physiological and replication stress conditions.

Regarding the timing of fork resolution by DNA2, our data show that if DNA2 degradation is induced when cells have already replicated the bulk of their genome, persistent replication intermediates accumulate in some cells. This indicates that the presence of DNA2 remains critical in very late-S/G2 phase. However, this is not to say that DNA2 is only required late in the cell-cycle and given that DNA2 promotes the restart of stalled DNA replication forks (Thangavel S et al. (2015) J Cell Biol 208:545-62. doi: 10.1083/jcb.2014061000), the most likely scenario is that DNA2 attends to stalled forks throughout S phase, although a critical role late might be implied considering that alternative fork recovery, for example by fork convergence, may be less effective in the last stages of S phase. Following the suggestion of **referee 1**, we have now analysed cells in which DNA2^{dd} was degraded for increasing periods of S phase and find that the levels of persistent replication intermediates and RPA-ssDNA accumulations rise with increasing lengths of S phase completed in the absence of DNA2. Accordingly, the actions of DNA2 throughout S phase limit the burden of unresolved replication intermediates cells face at the end of S phase, thus reducing the risk of dead-end HoRRer events in G2. The new data is presented in the revised version of the MS in **new Extended Data Fig. 7** (please see **specific comment (3c)**).

- The correlation between DNA2 loss, persisting and alternative processing of intermediates, and loss of cell fitness/senescence. This part of the work is the most novel and outstanding but it is just minimally investigated.

Authors' response: We have added several experiments to develop this point. In the revised version of the MS, we have further integrated the findings in the yeast and human systems by analysing the effect of Cdc27 homolog POLD3 on RPA-ssDNA accumulations in DNA2-deprived cells (please see reply to **specific comment (1)** below). We find that POLD3 depletion strongly reduces G2 DNA synthesis and RPA-ssDNA foci associated with loss of DNA2, consistent with a conserved essential function of DNA2 in preventing HoRRer-mediated checkpoint-activation and a block to mitosis from yeast to human. In the human system, checkpoint activation in DNA2-deprived cells results in immediate cell-cycle exit from G2 and we now provide additional evidence that cells enter a senescent state (**new panel d** in **Fig. 5**). Please see **response to specific comment (5)** for details of clarifications and new experimental evidence for cellular senescence upon DNA2 loss added in the revised version of the MS. Finally, upstream intervention in the HoRRer pathway by depletion of fork remodeler FBH1 brings down the incidence of RPA-ssDNA accumulations substantially and allows cells to circumvent cell-cycle exit and progress through

mitosis. Please refer to response to **specific comment (3)** and **new panels c and d** in **Fig. 4** in the revised version of the MS.

While I acknowledge the challenges in fully developing the third point, developing the first two is feasible for the authors' lab and is critical to providing the full mechanism of action and not just the consequences of DNA2 loss, which in my opinion is critical for Nature's audience.

Below, I provide the authors with specific comments and suggestions:

(1) **Fig. 1**: In light of the additional work the authors may want to add, I suggest eliminating this part of the study. It is merely incremental over what we know from yeast studies and not so much useful, in my opinion, as the bulk of the study is performed in human cells.

Authors' response: The data presented in **Fig. 1** is an integral part of our study because it represents the first demonstration that Dna2 dysfunction results in increased HoRReR at stalled DNA replication forks. To date, the prevailing model for the essential function of Dna2 in yeast posits an indispensable role of Dna2 during Okazaki fragment processing (recently reviewed in Zheng L et al. (2020) *Nucleic Acids Res.* 48:16-35. doi: 10.1093/nar/gkz1101). According to this model, Dna2-mutant cells suffer from rising checkpoint activation due to accumulating DNA flaps as they pass through S phase and eventually arrest before mitosis (Burgers P M (2011) *Cell Cycle* 10:2417-8. doi: 10.4161/cc.10.15.16201). We have previously suggested an alternative model based on evidence suggesting that cell cycle arrest-inducing checkpoint signalling in Dna2-mutant cells arises downstream of DNA replication fork stalling (Ölmezer G et al. (2016) *Nat Commun.* 7:13157. doi: 10.1038/ncomms13157). Subsequently, we showed that this checkpoint response in Dna2-defective cells can be suppressed by mutations compromising homologous recombination-dependent DNA synthesis (Falquet B et al., *Nucleic Acids Res.* (2020) 48:7265-7278. doi: 10.1093/nar/gkaa524). This allowed us to articulate the hypothesis that Dna2 is essential for guarding cells against the lethal consequences of excessive HoRReR at stalled DNA replication forks in yeast (Appanah R et al., (2020) *Curr Genet.* 66:1085-1092. doi: 10.1007/s00294-020-01106-7), but this has never been directly tested. Using the *RTS1* system in fission yeast, we now demonstrate for the first time that HoRReR levels at stalled forks are dramatically elevated in Dna2-defective cells. Furthermore, we show that a mutation of Pol δ subunit Cdc27, which specifically compromises recombination-dependent DNA synthesis, mitigates excessive HoRReR at *RTS1*, *dna2-2*-associated checkpoint activation, and the cessation of proliferation in Dna2-defective cells. To connect the yeast data better with the data obtained in human cells, following a suggestion by **referee 3**, we investigated the Cdc27 homolog POLD3 in human cells. We find that depletion of POLD3 strongly reduces the prolonged DNA synthesis at stalled DNA replication forks in human cells deprived of DNA2 (please see **new panel h** in **revised Fig. 4**). This further integrates the yeast and human data in the revised version of the MS, underscoring the conserved nature of the protective role of DNA2 in controlling HoRReR at stalled DNA replication forks (please see response to **referee 3**, **major point 1**). Overall, we regard the data pertaining to the fission yeast *RTS1* system (**Fig. 1**, **Extended Data Fig. 1**, and **new Extended Data Fig. 2** with an analysis of the relationship of *dna2-2* and *pfh1* in fission

corroborating the conclusions drawn from *dna2-2 cdc27-D1* cells; please see **referee 2, point 2**) as fundamental to the main thrust and logic of the MS for two main reasons: Firstly, they provide crucial evidence that recombination and checkpoint activation arises at stalled forks in Dna2-defective cells, demonstrating that Dna2 is needed to suppress HoRRer and linking this role to the essential requirement for Dna2 in yeast. This is a significant advance in the yeast system and important demarcation from the canonical view that Dna2 is indispensable because of an essential function in Okazaki fragment processing. Secondly, the demonstration of elevated HoRRer in Dna2-defective yeast cells provides the conceptual framework for our subsequent investigation of DNA2 in the human system, where evidence for an involvement of DNA2 in Okazaki fragment processing has not transpired (Duxin J P et al. (2012) J Biol Chem. 287:21980-91. doi: 10.1074/jbc.M112.359018), and where we go on to detect what is a conserved function of DNA2 in suppressing HoRRer at stalled forks, which is essential for cell proliferation.

(2a) **Fig. 2: Overall, findings shown in human cells are reminiscent of those reported by Rossi and colleagues in yeast (Nat. Commun., 2018). It would be important to determine which of DNA2's two potential enzymatic activities is more critical. Previous work on DNA2's role at perturbed replication forks implicated its nuclease activity, while Fig. 1 suggests a role for the helicase.**

Authors' response: We agree that there is a pleasing congruence between the rapid cessation of cell proliferation upon loss of Dna2 in fission yeast (Hu J et al. (2012) Cell 149:1221-32. doi: 10.1016/j.cell.2012.04.030. PMID: 22682245), budding yeast (Rossi S E et al. (2018) Nat Commun. 9:4830. doi: 10.1038/s41467-018-07378-5. PMID: 30446656) and the human system, as shown in our current MS. To address the contribution of the nuclease and the helicase activities of human DNA2 to this phenotype, we have now analysed cells in which endogenous DNA2^{dd} is degraded while inducing exogenous expression of wild-type or nuclease- and helicase-dead versions of DNA2. We find that, in contrast to the wild-type DNA2, neither the DNA2 D277A nuclease-deficient nor the DNA2 K654E helicase-deficient version can suppress the cessation of proliferation (please see **new Extended Data Fig. 9, panels a and b**). Thus, in contrast to yeast, where inactivation of the helicase activity of Dna2 is compatible with cell proliferation, this is not the case in human cells which succumb to both, inactivation of the helicase or nuclease activity of DNA2. Like cells deprived of DNA2, cells expressing nuclease- or helicase-deficient versions of DNA2 exhibit accumulations of RPA-ssDNA and exit the cell cycle (**Extended Data Fig. 9, panels c and e**), indicating that the coordinated action of the DNA2 nuclease and helicase domains is required at stalled DNA replication forks to prevent dead-end HoRRer events in human cells. Interestingly, upon expression of nuclease-deficient DNA2 D277A, G2 RPA-ssDNA foci per cell were markedly elevated, and this was preceded by a much higher number of cells showing multiple RPA foci (>5) in S phase as compared to wild-type DNA2 expressing cells (**Extended Data Fig. 9d**). This observation suggests that the helicase activity of DNA2, if uncoupled from the nuclease activity, generates replication stress in S phase. This detrimental gain-of-function effect exerted by the nuclease-deficient version of DNA2 further highlights the requirement for coordinated actions of the DNA2 nuclease and helicase activities. The additional data is compiled in

new Extended Data Fig. 9 and described on **page 13, line 320ff.** in the revised version of the MS:

*“As expected, expression of wild-type DNA2 in DIA-treated RPE-1 OsTIR1 DNA2^{dd} cells prevented the formation of RPA foci and rescued cell-cycle exit into senescence. In contrast, expressing nuclease-deficient (D277A), helicase-deficient (K654E), and double-deficient (D277A/K645E) versions of DNA2 did not restore proliferation, suppress RPA foci formation in G2, or cell-cycle exit in DNA2-deprived cells (**Extended Data Fig. 9**). Expression of DNA2 D277A caused strongly elevated RPA foci in S phase (**Extended Data Fig. 9d**), and an increase in G2 RPA foci compared to cells deprived of DNA2, likely reflecting uncontrolled DNA unwinding⁶⁹ in the absence of DNA2 nuclease activity. Thus, only the coordinated actions of the DNA2 nuclease and helicase activities can suppress an accumulation of stalled replication intermediates, HoRReR, and cell-cycle exit, providing a new explanation for why DNA2 is essential for cell proliferation.”*

(2b) In Fig. 2d the authors showed the consequences of DNA2 degradation at 24h of DIA treatment. It would be interesting to check what happens at 60h to match the live-cell imaging data of Fig. 2f.

Authors' response: We have extended the FACS analysis as shown in **Fig. 2d** to cover the 60-hour period used for live-cell imaging in **Fig. 2f** for RPE-1 OsTIR1 DNA2^{dd} cells in the presence or absence of DIA and treated or not with ATR inhibitor. Consistent with the invariable, ATR-dependent block to mitosis experienced by DIA-treated RPE-1 OsTIR1 DNA2^{dd} cells, and the accumulation of cells with a 4N DNA content after 48 hours (**Fig. 2d**), the 60-hour FACS analysis shows a further accumulation of cells with a 4 N DNA content at the 60 h timepoint, which is alleviated by inhibition of ATR. This data is now added as **new panel d** in **Extended Data Fig. 3** (previously **Extended Data Fig. 2**) and referred to in the **text on p. 7 (line 155)** in the revised version of the MS.

(2c) It would be interesting to back up long-term depletion of DNA2 with other methods. Given the long DIA treatment, I think that good siRNA or inhibitors might be an orthogonal method to check if the phenotype is somehow recapitulated.

Authors' response: Ligand-induced degrons such as the mAID and SMASH tags employed to degrade DNA2 in our study are a game-changer in the analysis of loss-of-function phenotypes in the space of a single cell cycle. Importantly, the use of DNA2^{dd} degradation-activating DIA has no intrinsic capacity to induce the phenotypes we ascribe to loss of DNA2 as there is no evidence of any significant interference with cell proliferation (**Fig. 2b-d**) or of the induction of γ H2AX foci in G2 phase (**Fig. 3a** and **Extended Data Fig. 4**) in DIA-treated control cells not harbouring the double-degron tag on DNA2. In the revised version of the MS we have included additional data which formally rule out the possibility that DIA treatment could potentially affect RPA-ssDNA accumulations in G2. The observations presented in **new Extended Data Figure 9** show that (1) DIA-treated control cells not containing the degron tag on DNA2 do not exhibit RPA-ssDNA accumulations in G2, while (2) DIA-treated RPE-1 OsTIR1 DNA2^{dd} cells do exhibit persistent RPA foci in G2 and exit the cell-cycle as expected upon DNA2 degradation; importantly – and in the presence of DIA – these phenotypes are fully suppressed by expression of wild-type DNA2, but not by the expression of an inactive (*i.e.*, combined nuclease/helicase-deficient) version of DNA2 as control. This

demonstrates conclusively that DIA treatment in our experimental set-up causes the phenotypes we ascribe to loss of DNA2 by activating the degron tag on DNA2^{dd}, thus leading to DNA2^{dd} degradation, as intended.

Traditional siRNA-mediated protein depletion requires cell transfection, is relatively slow-acting, and does not remove existing protein; the inherent caveat associated with siRNA is therefore an accumulation of secondary effects which may obscure primary defects linked to the loss of the protein of interest. Since DNA2-loss phenotypes take effect with full penetrance in the course of a single S/G2 phase upon rapid and efficient protein degradation, slower siRNA-mediated depletion of DNA2 is not suitable to adequately study DNA2-loss phenotypes.

Redacted.

(2d) What happens to cells treated with DIA and ATRi? I would expect they die. It is important to follow this phenotype as 60h should be enough to induce cell death. The absence of cell death would represent, per se, a nice result.

Authors' response: ATRi and DIA-treated RPE-1 *OsTIR1 DNA2^{dd}* cells continue to exclude trypan blue at the 60 h timepoint indicating that progression of DNA2-deprived through mitosis and into the next G1 phase does not result in acute cell death. The data

is included in the revised version of the MS as new **Extended Data Fig. 3e** and referred to in the text on **page 7, line 151ff.:**

“We observed that new-born daughter cells in which DNA2^{dd} was degraded prior to S phase failed to undergo even one subsequent mitosis; the invariable block to cell division could be suppressed by addition of ATRi, allowing cells to progress through mitosis and form live daughter cells (Fig. 2f and Extended Data Fig. 3d, e).”

(3a) Fig. 3: The authors claim that DNA2 processes an unknown intermediate. The identity of this intermediate has not been investigated but it should be as it also helps assess why ATR gets activated. Is the DNA2^{dd} phenotype dependent on fork reversal? Which proteins known to participate in fork reversal work upstream of DNA2? Previous work identified that DNA2 works after FBH1, at least under replication perturbation. What happens after replication fork arrest?

Authors' response: ATR is activated in the presence of substantial RPA-ssDNA which we detect in DNA2-deprived cells. We find that these RPA-ssDNA accumulations are generated by prolonged recombination-dependent DNA synthesis at stalled replication intermediates marked by FANCD2, which persist into G2 phase of the cell cycle. The consensus view currently is that recombination-dependent fork restart requires fork regression, which serves to generate a recombinogenic substrate with homology to the upstream template. In fission yeast, there is good evidence that HoRRer initiates at the regressed arm formed by fork reversal (Teixeira-Silva A et al. (2017) Nat Commun 8:1982. doi: 10.1038/s41467-017-02144-5). Published work by us and others showing that reversed fork intermediates accumulate upon deprivation of cells of DNA2 across organisms (Hu et al. (2012) Cell 149:1221-32. doi: 10.1016/j.cell.2012.04.030; Thangavel S et al (2015) J Cell Biol 208:545-62. doi: 10.1083/jcb.201406100; Rossi et al (2018) Nat Commun 9:4830. doi: 10.1038/s41467-018-07378-5), further support the notion that intermediates giving rise to HoRRer are four-way DNA junctions produced by fork reversal. We therefore concur with the expectation of **referee 1** that interference with fork reversal pathways might be expected to bring down HoRRer-dependent RPA-ssDNA accumulations in DNA2-deprived cells. To address this question directly, we have now depleted the factors defining two major fork remodeling pathways (Liu W et al. (2020) Sci Adv 6:eabc3598. doi: 10.1126/sciadv.abc3598), which are defined by the SNF2 translocases SMARCAL1, ZRANB3, and HLTF, and the UvrD helicase FBH1, respectively. Strikingly, depletion of FBH1, but not depletion of the SNF2 translocases, significantly reduced the formation of RPA foci in DIA-treated DNA2^{dd} cells (**new panels c and d in Fig. 4**). A subset of FBH1-depleted cells traversing S phase in absence of DNA2 even proceeded into the next G1 phase of the cell cycle, indicating that RPA-ssDNA and checkpoint signalling was reduced sufficiently to allow cells to progress through mitosis. Thus, FBH1-dependent fork reversal is required to initiate the dead-end HoRRer events causing cell-cycle exit upon DNA2 loss. These data further corroborate our model for the molecular pathology in DNA2-deprived cells by indicating that after replication fork arrest the actions of DNA2 are indispensable for balancing FBH1-dependent fork reversal and HoRRer at stalled forks. We refer to these new findings in the revised version of the MS on **page 10, line 239ff.:**

“Reversed RF intermediates have been shown to accumulate in DNA2-deprived cells^{2,28,30}, and fork reversal provides a substrate for HoRReR³⁶. Two major fork reversal pathways in human cells are defined by SNF2 translocases including SMARCAL1, HLF1, and ZRANB3, and by the UvrD helicase FBH1, respectively.²² Depletion of FBH1, but not depletion of the SNF2 translocases, significantly reduced the formation of RPA foci in DIA-treated RPE-1 OsTIR1 DNA2^{dd} cells (Fig. 4c). Strikingly, DNA2-deprived cells in which FBH1 depletion had fully suppressed persistent RPA-ssDNA accumulations were able to undergo mitosis and could be identified as G1 cells 8 h after traversing late S phase (Fig. 4d). These data are consistent with the notion that the complete block to mitosis that is associated with loss of DNA2 arises from RPA-ssDNA accumulations generated by HoRReR at unresolved RFs that have been remodeled in an FBH1-dependent manner.”

(3b) Is DNA2’s involvement different under normal or perturbed replication?

Authors’ response: To test the effect of perturbed replication in DNA2-deprived cells, we exposed cells to mild replication stress by adding low-dose (0.4 μM) aphidicolin (APH) to the culture medium. We have shown previously that completion of DNA replication is delayed under these conditions and cells continue DNA synthesis from S-phase through to early mitosis, thus minimizing genome under-replication and disengaging a transient ATR-imposed delay to mitotic entry (Mocanu C et al. (2022) Cell Rep 39:110701, doi:10.1016/j.celrep.2022.110701). The transitory nature of this stress response can be readily distinguished from the formation of persistent RPA-ssDNA accumulations leading to cell-cycle exit in DNA2-deprived cells. Accordingly, APH treatment resulted in RPA-focus formation in very few cells when DNA2 is present; in contrast, in DNA2-deprived cells, APH treatment resulted in a significant upshift of the median number of RPA-ssDNA accumulations across the entire cell population. This provides additional evidence supporting the critical role of DNA2 in processing stalled replication intermediates and indicates that DNA2 prevents dead-end HoRReR events upon endogenous as well as exogenous replication fork stalling. The data is presented in **new panel b of revised Fig. 4** and referred to in the revised version of the MS on **page 9, line 220ff.:**

“Next, we treated RPE-1 OsTIR1 DNA2^{dd} cells transiently with a low dose (0.4 μM) of replicative DNA polymerase inhibitor aphidicolin (APH) to induce mild replication stress⁵⁴ for increased RF stalling. In the absence of DIA, APH treatment resulted in a slight increase in cells with RPA foci 8 h after traversing late S phase (with the median of RPA foci per cell remaining zero). In contrast, co-treatment with DIA and APH strongly induced additional RPA foci (median of 22 foci per cell as compared to 14 without APH treatment) (Fig. 4b). Thus, DNA2 exposes RFs to RPA-ssDNA formation upon endogenous and exogenous fork stalling.”

(3c) The authors implicated DNA2 in G2 for processing persisting intermediates. This claim is based on experiments depleting DNA2 at specific moments of the cell cycle. However, the protocol used has shortcomings. For instance, how do the authors exclude that DNA2 is working on cells in very late S or very delayed in S-phase?

Authors’ response: In the original version of the MS, we noted that when DIA and EdU were administered to RPE-1 OsTIR1 DNA2^{dd} cells simultaneously, we were able to identify cells with a very late-S replication pattern of EdU incorporation exhibiting DIA-

induced γ H2AX foci (**Extended Data Fig. 3b** in the original version of the MS). In these cells DNA2 degradation was initiated only after bulk DNA replication, and thus there is a requirement for DNA2 after bulk DNA synthesis and into G2 for safeguarding cells from an accumulation of γ H2AX-marked intermediate. This does, however, not address the question of the extent to which the actions of DNA2 during S phase contribute to preventing the accumulation of persistent replication intermediates which otherwise go on to undergo prolonged HoRReR-mediated DNA synthesis and give rise to RPA bodies in G2. To address this point, we have now quantitatively analysed RPE-1 *OsTIR1 DNA2^{dd}* cells to which DIA was added at different timepoints before late-S phase for RPA focus formation in G2 and their ability to progress through mitosis and into the next G1 phase (please see **new Extended Data Fig. 7**). We find that the number of RPA-ssDNA accumulations post S-phase correlates with the proportion of S phase cells traverse after degradation of DNA2^{dd} (for example, if DNA2 degradation is induced prior to S phase, the number of RPA-ssDNA accumulations in G2 is much higher than if DNA2^{dd} degradation is induced in the middle of S phase). If DNA2 was present through late S phase, the number of RPA-ssDNA foci in G2 was greatly diminished and a large proportion of cells progressed to the next G1 phase, indicating that checkpoint activation was sufficiently reduced in a subset of cells to allow mitosis. These data are consistent with DNA2 promoting the recovery of stalled forks through to very late S/G2 phase, thus limiting the persistence of potential substrates for HoRReR and an accumulation of RPA-ssDNA by prolonged recombination-dependent DNA synthesis in G2. That DNA2 is active throughout S phase is in line with its role in promoting fork restart (Thangavel S et al (2015) J Cell Biol 208:545-62. doi: 10.1083/jcb.201406100) and our finding of the gain-of-function effect associated with the nuclease-deficient version of DNA2 which manifests itself during S phase (please see **new Extended Data Fig. 9d**). In the revised version of the MS, our analysis of the relationship between DNA2 availability in S phase and the extent of RPA-ssDNA formed in G2 is described in the text on **page 10, line 227ff.**:

*“When we treated RPE-1 *OsTIR1 DNA2^{dd}* cells with DIA at varying timepoints before late S-phase, we observed that the number of FANCD2 and RPA foci persisting 8 h after late S phase correlated with the proportion of S phase that cells had traversed after the induction of DNA2^{dd} degradation. Thus, if DNA2 degradation was induced prior to S phase, cells exhibited the expected level of RPA-ssDNA accumulations (median 12 if DIA was added 12 h before late S phase). When DNA2^{dd} was degraded in the first half of S phase, RPA-ssDNA accumulations were moderately reduced (median 11 if DIA was added 8 h before late S phase) and steadily declined as cells traversed less of S phase without DNA2^{dd}. If DIA was added 2 h before late S phase, the median of RPA-ssDNA foci per G2 cell was reduced to 3 and over 70% of cells progressed to the next G1 phase (**Extended Data Fig. 7**). These data show that DNA2 is required throughout S phase to avoid a build-up of stalled replication intermediates accumulating RPA-ssDNA in G2 phase of the cell cycle.”*

(3d) Why not also use shorter DIA treatment given that DNA2 goes down completely after 4h. Performing parallel experiments with some enrichment strategy might be better. If I am not wrong, RPE1 cells can be synchronized using serum starvation. Alternatively, QIBC could be performed to determine the cell cycle stage in which cells

experience DNA damage, RPA foci, and so on. This would help a lot in each of the IF experiments shown in this study.

Authors' response: The mAID-SMASH degron tag allows removal of DNA2 from cells within 4 hours of DIA addition; subsequently, DIA is retained in the culture medium, which is necessary for the continued posttranslational removal of DNA2^{dd} while HoRRer-dependent RPA-ssDNA accumulations emerge at persistent replication intermediates after bulk replication by prolonged recombination-dependent DNA synthesis in G2 phase of the cell cycle. Thus, the time of DIA exposure in our experiments is not determined by the time it takes to degrade DNA2^{dd}, but by the time required for DNA2-loss phenotypes to present themselves post-S phase. In the revised version of the MS, we provide new experiments with shorter DIA exposures (please see **response to point 3c** above) and additional data to show that the DIA treatment we employ causes the observed phenotypes exclusively by way of DNA2 degradation (please see **response to point 2c** and **new Extended Data Fig. 9**). The need for cell-cycle synchronisation or additional quantitative image-based cytometry is obviated by our ability to determine cell-cycle phase on a per-cell basis (by the combined use of EdU labelling/EdU incorporation pattern, measurements of DAPI content, and IF of cell-cycle markers). Importantly, our method also provides a precise and cell-specific measure of time elapsed since completion of S phase, which is critical for our analyses. We therefore feel there is no strong impetus to change our IF workflow and favour avoiding the introduction of non-essential experimental treatments such as for example serum starvation.

(3e) **Another critical point is whether DNA2 is recruited differently at the fork, behind the work in S-phase, or G2 cells. Orthogonal approaches can be used to determine when DNA2's function is needed.**

Authors' response: Our new analysis (as detailed in response to **specific comment 3c**) presented in **Extended Data Fig. 7** shows that DNA2 is required throughout S phase to limit the amount of unresolved replication intermediates in G2 and the risk of dead-end HoRRer events occurring at these intermediates in G2. Previously, DNA2 has been proposed to promote Okazaki fragment maturation in S phase, thereby avoiding the accumulation of DNA flaps behind replication forks (Rossi S E (2018) Nat Commun. 9:4830. doi: 10.1038/s41467-018-07378-5. PMID: 30446656). This is consistent with the two-nuclease model where Okazaki fragment maturation mainly proceeds by generating short-lived one- or two-nucleotide flaps that are quickly removed by Rad27 (FEN1), and Dna2 is required to trim down longer flaps bound by RPA to make them accessible for subsequent processing by Rad27 (Burgers P M (2011) Cell Cycle 10:2417-8. doi: 10.4161/cc.10.15.16201). According to this model, Okazaki fragments remain unligated and checkpoint-mediated cell-cycle arrest occurs on account of RPA-bound flaps generated by displacement DNA synthesis on the lagging strand. As we point out in the Discussion section of our MS, strand-displacement synthesis on the lagging strand has been shown to be restricted to no more than 150-200 nucleotides by the deposition of nucleosomes on the nascent sister chromatids. This is not compatible with the extensive accumulation of RPA-ssDNA depending on prolonged recombination-dependent DNA synthesis in G2 phase of the cell cycle in DNA2-deprived cells, but we fully agree with **referee 1** that it is of interest to address a potential defect in Okazaki fragment processing in DNA2-deprived cells directly. We

have therefore collaborated with Prof. Caldecott whose lab has recently shown that PARP activation in S phase is a sensitive measure of unligated Okazaki fragments (Hanzlikova H et al. (2018) Mol Cell 71:319-331.e3. doi: 10.1016/j.molcel.2018.06.004). Accordingly, we find that S phase poly(ADP-ribose) (PAR) is strongly induced by inhibition of FEN1. In contrast, loss of DNA2 did not result in increased PAR signals in DIA-treated DNA2^{dd} cells, nor did loss of DNA2 synergize with FEN1 inhibition to further increase the elevated PAR signal in FEN1i-treated cells. These data are consistent with previous reports of proficient Okazaki fragment processing in DNA2-depleted human cells (Duxin J P et al. (2012) J Biol Chem. 287:21980-91. doi: 10.1074/jbc.M112.359018) and have been included in the revised version of the MS (presented in **new Extended Data Fig. 5**) and referred to in the text on **page 8, line 182ff.** and **page 17, line 420ff.:**

*“It has been suggested that Dna2 dysfunction in yeast results in DNA damage checkpoint activation at unligated Okazaki fragments bearing DNA flaps.^{13,18,30} In human cells, poly(ADP-ribose) polymerase PARP1 acts as a sensor of unligated Okazaki fragments and S-phase poly(ADP-ribose) (PAR) is strongly elevated upon inhibition of the canonical Okazaki fragment processing nuclease FEN1.⁵⁵ To address whether γ H2AX foci in DNA2-depleted cells might be linked to perturbed Okazaki fragment maturation, we analysed PAR levels in DIA-treated RPE-1 OsTIR1 DNA2^{dd} cells. DIA addition did not result in increased PAR signals, nor did it synergize with FEN1 inhibitor treatment to elevate PAR levels above those detected upon inhibition of FEN1 (**Extended Data Fig. 5**). This is in line with previous reports of proficient Okazaki fragment processing in DNA2-depleted human cells.¹⁹”*

“The long-standing Okazaki fragment processing model¹³ for the essential role of DNA2 is not easily compatible with our data, and – in contrast to conditions of FEN1 deficiency – we find no evidence of perturbed Okazaki fragment maturation upon DNA2 deprivation.”

We thank referee 1 for raising this point as addressing it has significantly improved our ability to make the distinction between our new model for the essential role of DNA2 in safeguarding cells from HoRRerR-dependent accumulations of RPA-ssDNA and the canonical Okazaki-fragment-processing-defect model.

(4) Fig. 4: The authors claim that RAD51-dependent recombination is required to replicate intermediates accumulating in the absence of DNA2 in G2. This is interesting but is similar to findings we might expect from studies in yeast if it remains underdeveloped. For instance, the authors showed that EdU labelling can be observed at RPA foci and somehow suggest that this incorporation occurs downstream recombination-based replication. However, while loss of EdU labelling when RAD51 nucleofilament formation is inhibited can be consistent with the hypothesis, I did not understand why RPA foci should be reduced as well. I would expect that ssDNA is covered by RAD51. Perhaps labelling of parental DNA and evaluation of the presence of ssDNA directly might help reinforce this data. It is also unclear if what the authors identified differs from MiDaS, for example. Experiments to differentiate between the two should be evaluated, or at least this should be discussed.

Authors' response: Our data show that RAD51 strand exchange activity is required for the prolonged G2 DNA synthesis which we observe at γ H2AX/FANCD2/RPA-marked intermediates in DNA2-deprived cells. We agree with the referee that this is consistent with our model of DNA2 being an essential gatekeeper to HoRRer at stalled DNA replication intermediates. In the revised version of the MS we now add additional supporting data by showing that depletion of POLD3, which is required for recombination-dependent DNA synthesis across organisms (Lydeard J R et al. (2007) Nature 448:820-823, doi:10.1038/nature06047; Costantino L. et al. (2014) Science 343:88-91, doi:10.1126/science.1243211; Xu Y et al. (2025) Mol Cell 85:91-106 e105, doi:10.1016/j.molcel.2024.10.026) also reduces EdU incorporation at stalled replication intermediates in human cells deprived of DNA2, similarly to RAD51 inhibition (**new panel h in revised Fig. 4**). In both cases, RAD51 inhibition and POLD3 depletion, RPA foci intensity reduces along with reduced EdU incorporation at these sites. This is consistent with a reduction in nascent DNA being spooled out as ssDNA and bound by RPA behind the moving D-loop during recombination-dependent DNA synthesis. To validate this interpretation further, we followed the suggestion of **referee 1** and probed directly for ssDNA, showing by native detection of BrdU incorporation the generation of nascent ssDNA in a RAD51-dependent manner following loss of DNA2.

Finally, the MiDAS (mitotic DNA synthesis) model (reviewed in: Bhowmick R et al. (2023) Mol Cell 83:3596-3607. doi: 10.1016/j.molcel.2023.08.023) postulates a persistent replication structure with which cells cannot deal in G2 phase. Thus, cells enter mitosis, and upon chromosome condensation this inaccessible intermediate is converted into a DNA double-strand break which gives rise to break-induced replication (BIR). Thus, by definition, MiDAS occurs in, and is triggered by, entry of cells into mitosis. Upon loss of DNA2, cells invariably fail to enter mitosis and HoRRer-dependent RPA-ssDNA accumulations in G2 arise independently of DNA double-strand break formation. In the revised version of the MS we now highlight these differences by introducing MiDAS as a mitosis-specific BIR-type reaction that represents a different recombination-dependent DNA synthesis sub-pathway compared to the HoRRer events in DNA2-deprived cells occurring in a different phase of the cell cycle and independently of DNA double-strand break formation. The text on **page 13 (line 308ff.)** of the revised version of the MS has been amended accordingly, now reading:

*“Because RPA-ssDNA accumulation in a RAD51/POLD3-dependent manner (see Fig. 4) could be compatible with either HoRRer at stalled RFs or BIR, which mediates recombination-dependent DNA synthesis after fork breakage and DNA double-strand break formation in interphase or in mitosis (known as mitotic DNA synthesis, or MiDAS)⁶⁶, we probed for DNA double-strand break-specific phosphorylation of RPA32 on S4 and S8 (pRPA)^{67,68}. Only a small subset of cells (approx. 5% of RPE-1 OsTIR1 DNA2^{dd} cells 18 h after DIA addition) exhibited pRPA foci, and these were only detected in cells in which cyclin B1 had translocated into the nucleus or been degraded (**Extended Data Fig. 8d**). This shows that RAD51/POLD3-dependent RPA-ssDNA formation and the commitment to cell-cycle exit is initiated at stalled replication intermediates in a manner that is independent of DNA double-strand break formation and consistent with HoRRer.”*

(5a) **Fig. 5:** It would be useful to add an explanation here for the use of the double

degron to evaluate whether the simpler mAID system cannot be used or cannot mimic hypomorphic mutations as they occur in genetic diseases.

Authors' response: The double-degron tag combining mAID and SMASh tag has been shown to work highly effectively for rapid protein degradation (Lemmens B et al. (2018) Mol Cell 71:117-128.e3. doi: 10.1016/j.molcel.2018.05.026) and by default we activate both tags to rapidly degrade DNA2^{dd} in this study. However, as expected, activation of only the SMASh or mAID tag alone recapitulates DIA-induced DNA2 loss (see **Extended Data Fig. 3b, c** and **Fig. 6**, respectively). In experiments designed to mimic the effects of reduced DNA2 levels as associated with the hypomorphic DNA2 mutations in primordial dwarfism syndromes, we only target the mAID tag. Here, the aim is a well-controlled dose-response curve (**Fig. 6a**) (and not necessarily maximal protein degradation speed), which is more suitably done with by titrating a single (IAA) rather than two drugs (*i.e.*, IAA and asunaprevir). The effects of different IAA concentrations on DNA2^{dd} levels are shown in **new Extended Data Fig. 10a** in the revised version of the MS. Moreover, we have extended our analysis to the expression of patient-derived DNA2 variants T655A (Tarnauskaitė Ž et al (2019) Hum Mutat 40:1063-1070. doi: 10.1002/humu.23776) and DNA2 L48P (Di Lazzaro Filho R et al (2023) J Med Genet 60:1127-1132, doi:10.1136/jmg-2022-109119) in DNA2-deprived cells. While DNA2 L48P phenocopies DNA2-loss, DNA2 T655A provides a mild rescue of cell proliferation and cell-cycle exit into senescence. Importantly, this validates our proposed mechanism where cells that reach a critical level of residual DNA2 activity either continue to proliferate or trigger a DNA2-loss phenotype and enter senescence, thus no longer contributing to growth, with a hypomorphic DNA2 mutation identified in an individual with primordial dwarfism. The new data is described on **page 15 (line 357ff.)** of the revised version of the MS:

*“While most of the reported primordial dwarfism-linked DNA2 mutations are intronic and reduce DNA2 levels through the expression of aberrant and mis-spliced DNA2 transcripts, one Seckel syndrome patient and one Rothmund-Thomson-related patient were found to be compound heterozygous for intronic mutations in trans with missense mutations resulting in variants DNA2 T655A and DNA2 L48P, respectively^{6,7}. T655 is located proximal to the ATP-binding site within the helicase-domain of DNA2, L48 within an oligonucleotide/oligosaccharide binding (OB) domain⁷⁰ which precedes the nuclease domain. We expressed the patient-derived DNA2 variants in DIA-treated RPE-1 OsTIR1 DNA2^{dd} cells (**Fig. 6e**). DNA2 T655A expression provided a mild suppression of the cell proliferation defect in DNA2-deprived cells (**Fig. 6f**) and recapitulated a partial cell-cycle exit phenotype similar to that observed after partial degradation of DNA2^{dd} (**Fig. 6g**). Concomitantly, DNA2 T655A-expressing cells exhibited a reduced number of G2 γH2AX/RPA foci compared to DNA2-deprived cells 8 h after S phase, and cells that had transitioned to the next G1 phase were largely devoid of γH2AX/RPA foci (**Fig. 6h**). In contrast, expression of DNA2 L48P did not rescue cell proliferation, cell-cycle exit, or γH2AX/RPA foci formation in DNA2-deprived cells (**Fig. 6f-h**); while the OB domain of DNA2 folds against the nuclease domain, we did not find evidence of elevated RPA foci in S phase^{7,70}, indicating that the DNA2 L48P mutant does not behave like a nuclease-deficient/helicase-proficient DNA2 D277A mutant (**Extended Data Fig. 10c**). These data validate the pathogenicity of the DNA2 T655A and L48P patient mutations and indicate they result in partial and complete loss of DNA2 function, respectively.”*

And referred to in the Discussion on **page 19, line 468f.:**

“Similarly, expression of patient-derived variant DNA2 T655A resulted in stochastic cell-cycle exit.”

(5b) The rationale for the experiment shown in Fig. 5d is unclear. It seems expected that RPA foci are observed in S/G2 cells and in cells progressing into mitosis, given the other results shown in Fig. 5 and 3. Perhaps this experiment could be moved to the supplementary figures. If the main message of this figure is that cells depleted of DNA2 in G2 do not recover and are marked for senescence, more compelling experiments analyzing other readouts of senescence should be added. If the authors' hypothesis is that senescence (or quiescence) directly derives from the cell cycle arrest shown in Fig. 2f, then the study should contain something performed with ATRi or other genetic strategy to affect checkpoint signalling from the still unknown intermediate left unprocessed by DNA2 depletion

In general, the authors should add more compelling data on this critical point. In my opinion, this might represent the key finding of the story and the most intriguing one perhaps.

Authors' response: **Fig. 5d** in the original MS (now **Fig. 5e**) does not show mitotic cells, but the transition of cells from G2 (cyclin B1 accumulation in the cytoplasm) to commitment to cell-cycle exit and mitosis (cyclin B1 nuclear sequestration and eventual degradation marking the point of no return for cell-cycle exit from G2) which ensues upon loss of DNA2. Specifically, the experiment shown in **Fig. 5e** establishes the temporal relationship between cyclin B1 nuclear sequestration and the accumulation of RPA-ssDNA, which we find precedes cyclin B1 sequestration/commitment to cell-cycle exit. To add additional evidence demonstrating the fate of DNA2-deprived cells is cell-cycle exit into senescence from G2 (without progression into mitosis), we have now measured upregulated β -galactosidase activity as a biomarker for senescent cells. As shown in **new panel d of revised Fig. 5** (and referred to in the text of the revised of the MS on **page 12, line 294ff.:** *“Moreover, DIA-treated RPE-1 OsTIR1 DNA2^{dd} cells monitored over a period of 14 days stained for increasing levels of senescence-associated β -galactosidase and showed an enlarged cell morphology, confirming that DNA2-deprived cells exiting the cell cycle from G2 enter a senescent state (Fig. 5d).”*), DNA2-deprived cells adopt the enlarged morphology of senescent cells and show ever-increasing β -galactosidase activity from day eight after induction of DNA2^{dd} degradation. This is fully consistent with our findings of cyclin B sequestration and the striking increase in nuclear area upon loss upon DNA2 loss. To affect checkpoint signalling in DNA2-deprived cells, we deplete p21 and show that this prevents the invariable cell-cycle exit phenotype exhibited after DNA2 loss and suppresses entry into senescence (**Extended Data Fig. 8a-c** in the revised MS).

(5c) As the cell model used here is not a primary one, the authors might want to exploit other strategies to interfere with DNA2 function (inhibitors?) or, as an alternative strategy, they might evaluate to investigate if DNA2 loss affects senescence induced by DNA-damaging agents.

Authors' response: We note that senescence has been observed in cells directly derived from a primordial dwarfism patient with *DNA2* mutations (Shaheen R et al. (2014) *Genome Res* 24:291-9. doi: 10.1101/gr.160572.113). While this was a qualitative rather than quantitative observation and the underpinning mechanism remained elusive, the data are in good agreement with our findings and we now refer to this in the text on **page 14, line 332ff.:**

“Case reports indicate low-level expression of wild-type DNA2 in DNA2-linked microcephalic primordial dwarfism, Seckel syndrome, and Rothmund-Thomson syndrome, and senescent cells have been observed among cultured patient fibroblasts.”⁵⁻⁷

Redacted. With respect to a potential synergy between DNA2 loss and DNA damage-induced senescence, we note that DNA2-loss alone already results in a fully penetrant cell-cycle exit and senescence phenotype. However, our extended complementation experiments with wild-type and catalytically inactive DNA2 (**new Extended Data Fig. 9**) in the revised version of the MS demonstrate unequivocally that HoRRer-dependent RPA-ssDNA and cell-cycle exit after DNA2^{dd} degradation are the direct consequences of DNA2 loss. We also provide β -galactosidase assays in the revised version of the MS to validate that cells exit the cell cycle into senescence upon DNA2 loss (**new panel d** in revised **Fig. 5**).

(5d) Overall, the data are well presented with adequate reference to statistical methods and data analyses. The manuscript is well written and key bibliography is properly mentioned.

Authors' response: We thank **referee 1** for the positive assessment of our MS and the insightful suggestions for our experimental revision.

Referee #2:

DNA2 is a cell-essential nuclease/helicase that is important for genome stability. DNA2 deficiency causes undergrowth syndromes and DNA2 is commonly expressed at higher levels in cancer, so understanding its cellular role is not only of fundamental importance, but also relevant to human disease. DNA2 has a number of proposed biological functions, but in spite of decades of research the function that is essential for continued proliferation has remained elusive. Here, Hudson, Rass and colleagues build on previous work from the Rass lab in *S. cerevisiae*, and propose that the most important function of DNA2 is to suppress HoRRer and checkpoint activation at stalled replication forks in both *S. pombe* and human cells.

Experiments in *S. pombe* make use of a helicase mutant allele that is viable but has increased checkpoint activation and decreased survival, as well as increased recombination at an inducible replication barrier. Effects on recombination and survival, but not checkpoint activation, are rescued by a Cdc27 allele with a C-terminal truncation (cd27-D1) that reduces D-loop DNA synthesis required for HoRRer. Pu-Seq is used to show increased Pol delta DNA synthesis (which implies increased HoRRer), which is partially suppressed by cdc27-D1.

Experiments in human cells make use of rapid degradation of endogenously degran-tagged DNA2 paired with EdU labelling and measuring DAPI intensity to identify cells at different stages of the cell cycle, followed by a range of fluorescence microscopy-based analyses to determine the consequences. DNA2 depletion causes cells to accumulate in G2/M, leading to increased gH2AX, FANCD2 and RPA foci. Reduced RPA foci intensity and reduced DNA synthesis at RPA foci upon RAD51 inhibition suggests that this is due to excessive HoRRer at stalled replication forks upon loss of DNA2. Finally, partial depletion of DNA2 (a situation that mimics the genetic cause of human undergrowth syndromes) is shown to cause increased senescence as measured by CyclinB1 nuclear translocation and increased nuclear area, with the latter reduced upon p21 knockdown at the cost of increased formation of micronuclei. This provides a plausible model for how partial loss of DNA2 function in humans causes reduced growth.

Altogether this is an interesting and important piece of work that establishes the essential function of DNA2, relevant to human disease. However, I would recommend strengthening the paper by addressing the following points.

(1) Clarify EdU labelling experiments used to identify cells at time points after late S phase

Can the authors please clarify the experimental setup for Figs. 3b-e and 4a,b and ED Fig. 4? Firstly, it is not clearly indicated whether a pulse of EdU was administered or if EdU was present from t=0. Secondly, I understand that patterns of EdU foci can help identify late S-phase cells. However, because this requires fixation, I do not understand how cells that were in late S at the time of labelling can then be followed after that as the text seems to imply (lines 156-160 and 180-182).

In contrast to what the main text implies, in the figure diagrams it is indicated that late S-phase cells are identified at the different time points, which obviously can be done, but it changes the meaning of the results. I assume that the authors indeed mean that the cells that at the time of labelling are identified at the different time points. For reviewing the paper (in spite of this confusion), I made the assumption that it is indeed possible to identify pulse-labelled cells that were in late S at the time and that those cells were then identified at different time points after that for further staining and analysis. Textual changes and changes to the diagrams should ensure other readers do not get confused and know how to interpret the experiments shown.

Authors' response: **Referee 2** is correct in their interpretation that in the workflow described in **Fig. 3b** and deployed to generate the data presented in **Figs. 3c,d** and **4a, b**, cells are identified which were in late S phase (as identified by EdU incorporation pattern) at the time of EdU addition. In these experiments, EdU is added and kept on (it is not a pulse-label) in the culture medium for the remainder of the experiments. Cells traversing S phase in the period between EdU addition and fixation will therefore show a pan-nuclear EdU signal, while cells that had already reached late-S at the time of EdU addition will exhibit a late-S EdU incorporation pattern (since for the remaining incubation period in the presence of EdU, these cells will be in G2). Thus, fixation 2, 4, or 8 hours after EdU addition allows the selection and analysis of cells which had been in late S phase 2, 4, or 8 hrs earlier. We apologize for the confusion and have extended the description for the work-flow diagram in **Fig. 3b** which previously read “*To quantify γ H2AX foci at different times after cells had entered G2, we analysed RPE-1 *OstTIR1 DNA2^{dd}* cells 2, 4, and 8 h after they had traversed late S-phase (as identified by EdU incorporation pattern⁵⁰) (Fig. 3b)*” to a revised version on **page 7, line 164ff.:**

*“Next, we treated non-synchronous *OstTIR1 DNA2^{dd}* cells with DIA for 14 h before adding EdU to the culture medium to label replicating DNA. Upon fixation 2, 4, and 8 h later, cells with a late S-phase EdU incorporation pattern⁵⁴ (i.e., cells which had been in late S phase 2, 4, and 8 h earlier) were identified, allowing us to quantify γ H2AX foci at different times after cells had entered G2 following their traversal of S phase in the absence of DNA2 (Fig. 3b).”*

Given that pulse-labelling with EdU is used elsewhere in the MS, we have also redrawn all experimental workflow diagrams to clearly indicate where EdU exposure is continuous after the time of addition and improved the workflow description in the relevant figure legends.

(2) Perform additional experiments to provide more detail for the molecular mechanism by which DNA2 suppresses HoRRer, prevents checkpoint activation and promotes RF restart

- Cdc27-D1 rescues reduced viability of *dna2-2* (Fig. 1g), but does not rescue checkpoint activation (as measured by cell length increase; Fig. 1f): relative to WT there is increased cell length for *dna2-2*, *cdc27-D1* as well as the double. This seems to be somewhat contradictory. Is checkpoint activation for *cdc27-D1* due to non-HoRRer

functions of Cdc27, or does it say something more about the details of the molecular mechanism of checkpoint activation vs cell death? Are there other mutants (e.g. Pfh1-m21/mt*, PMID: 30667359) that suppress HoRReR that can give further insight?

Authors' response: The *cdc27-D1* allele, like *dna2-2*, is associated with chronic checkpoint activation (**Fig. 1f**). In contrast to budding yeast, where a Pol31 polymorphism (providing enhanced stability to the DNA polymerase δ complex) renders the Cdc27 homolog Pol32 non-essential (Loeillet S et al. (2023) DNA Repair 127:103514. doi: 10.1016/j.dnarep.2023.103514), in fission yeast, Pol δ cannot support cell proliferation without Cdc27. This heightened dependence on Cdc27 for canonical replication provides a HoRReR-independent explanation for the chronic checkpoint activation in cells expressing the C-terminally truncated Cdc27-*D1* hypomorph. Importantly, while *dna2-2* significantly increases the cell length compared to wild-type, in the *cdc27-D1* background, the additional introduction of *dna2-2* does not result in a significant increase in cell length over *cdc27-D1* (**Fig. 1f**). This lack of an additive checkpoint activation phenotype in *cdc27-D1 dna2-2* double-mutant cells indicates that under conditions where elevated HoRReR at stalled DNA replication forks linked to Dna2 dysfunction is suppressed by *cdc27-D1*, *dna2-2* itself no longer causes checkpoint activation. To be more explicit on this point, we changed the text on **page 5 (line 103ff.)** of the revised version of the MS to read:

“Moreover, while dna2-2 and cdc27-D1 are each associated with chronic checkpoint activation leading to cell elongation compared to wild-type, introduction of the dna2-2 allele into the cdc27-D1 background did not result in a further increase in cell length beyond the baseline provided by cdc27-D1 cells (Fig. 1f). Finally, cdc27-D1 restored the diminished cell viability of dna2-2 cells (Fig. 1g).”

In addition, we have now constructed the relevant reporter strains harbouring *pfh1-mt** (Pinter S F, et al. (2008) Mol Cell Biol 28:6594-608. doi: 10.1128/MCB.00191-08) and *dna2-2 pfh1-mt** and performed an analysis of HoRReR at *RTS1* as suggested. In fission yeast, *pfh1-mt** suppresses recombination-dependent DNA synthesis less effectively compared to *cdc27-D1* (Xu Y et al. (2025) Mol Cell 85:91-106.e5. doi: 10.1016/j.molcel.2024.10.026) and *ade+* recombinants in the reporter assay we use have been shown to be slightly elevated *pfh1-mt** compared to wild-type, likely due to fission yeast-specific roles of Pfh1 in suppressing template switching (Jalan M et al. (2019) Elife 8:e41697. doi: 10.7554/eLife.41697). Nonetheless, we find that the high levels of *ade+* recombinants at *RTS1* in *dna2-2* cells are strongly suppressed in the presence of *pfh1-mt**. Concomitantly, *pfh1-mt** suppresses cell elongation in cells harbouring *dna2-2* and rescues the poor plating efficiency of *dna2-2* cells. These data fully corroborate the conclusions drawn from **Fig. 1** and are presented in **new Extended Data Fig. 2** in the revised version of the MS. We refer to these new data on **page 5 (line 107ff.)**:

“Similar results were obtained for interactions between Dna2 and Pif1 helicase homolog Pfh1, another factor required for recombination-dependent DNA synthesis⁴⁵. Thus, pfh1-mt, encoding a Pfh1 version that is excluded from the nucleus⁴⁸, suppressed excessive recombination at RTS1, checkpoint activation, and poor viability in dna2-2 cells (Extended Data Fig. 2).”*

- All *S. pombe* experiments make use of the *dna2-2* helicase mutant, but nuclease and not helicase activity of DNA2 has previously been shown to be necessary to prevent RF reversal (ref 2) and to promote restart (ref 28). Do the Fig. 1 experiments using nuclease-defective instead of helicase-defective DNA2 give similar or different results?

Authors' response: Nuclease-defective versions of Dna2 do not support cell proliferation in budding or fission yeast (Budd M E et al. (2000) *J Biol Chem* 275:16518-29. doi: 10.1074/jbc.M909511199; Hu J et al. (2012) *Cell* 149:1221-32. doi: 10.1016/j.cell.2012.04.030). Consequently, experiments such as those presented in **Fig. 1** cannot be performed with a nuclease-dead version of Dna2. However, we have addressed the contributions of the nuclease and helicase activities of DNA2 to suppressing HoRReR and senescence in human cells in the revised version of the MS, as detailed in the response to the subsequent question below. Briefly, we now demonstrate that expression of a helicase-defective version of DNA in human cells cannot protect from HoRReR-dependent RPA-ssDNA accumulations, which is consistent with the elevated levels of HoRReR we detect at *RTS1* in fission yeast cells expressing Dna2 R1132Q encoded by the *dna2-2* allele. Furthermore, we uncover additional replication stress and an aggravated HoRReR phenotype upon expression of a DNA2 nuclease-deficient version in human cells that is not seen upon expression of the helicase-defective version; this additional detrimental effect might explain the lethality associated with Dna2 nuclease-deficient alleles in yeast.

- How does DNA2 suppress HoRReR in human cells? Is the helicase or nuclease function (or both) essential for DNA2 suppressing HoRReR, and how does this relate to findings from ref 28? The setup as presented in ED Fig. 2 should allow complementation with helicase- and nuclease-defective DNA2 to address this. Beyond establishing more detail for the molecular mechanism, this is important for two reasons. Firstly, in addition to a strong reduction in DNA2 levels in most cases of human undergrowth syndromes associated with DNA2 mutations, a likely helicase-defective mutant has been identified in primordial dwarfism (ref 6) and a point mutation that may affect the nuclease domain has been identified in an RTS-like syndrome (ref 7). The latter includes poikiloderma, with DNA repair defects a common genetic cause of this skin condition, which may be relevant. Secondly, it could help establish if DNA2 inhibitors to treat cancer should target helicase and/or nuclease activities.

Authors' response: Ref. 28 in the first version of the MS (Thangavel S et al. (2015) *J Cell Biol* 208:545-62. doi: 10.1083/jcb.2014061000) measures replication fork degradation, which can be observed upon prolonged exposure of human cells to hydroxyurea. In the presence of siRECQ1, fork degradation was in part dependent on DNA2 and could not be complemented with a nuclease-deficient version of DNA2. A helicase-deficient but nuclease-proficient version of DNA2 did support apparent fork degradation, but whether this mutant might promote fork restart under these conditions was not addressed. Hyper-resection at HU-arrested replication forks constitutes a pathological phenotype; thus, the ability of a helicase-deficient version of DNA2 to attack forks in this experimental set-up provides evidence of its ability to act on stalled replication forks, which seems consistent with the hypomorphic (and therefore viable) Dna2 helicase mutants in budding yeast (e.g., Ölmezer et al. (2016) *Nat Commun* 7:13157. doi: 10.1038/ncomms13157) and fission yeast (**Fig. 1** in our current MS). However,

whether this might complement the essential role of DNA2 in cell proliferation was not addressed for helicase-deficient or nuclease-deficient versions of DNA2 in the study by Thangavel and co-workers. To address these points, we followed the suggestions of **referee 2** and have now conducted a comprehensive complementation analysis by expressing nuclease-deficient (D277A), helicase-deficient (K654E), and double-deficient (D277A/K645E) versions of DNA2 in cells upon degradation of endogenous DNA2^{dd}. We find that cells expressing the nuclease-, helicase-, and double-deficient DNA2 mutants exhibit persistent replication intermediates after S phase which are marked by FANCD2 and give rise to RPA bodies in G2. Thus, individually, the nuclease or helicase activity of DNA2 fail to suppress dead-end HoRRer events in human cells, indicating close cooperation of the DNA2 nuclease and helicase activities in the proper processing of stalled DNA replication forks. Consistently, and in contrast to re-expression of wild-type DNA2, the nuclease-, helicase-, and double-deficient mutants fail to suppress the immediate exit from the cell cycle and cessation of proliferation caused by loss of DNA2. This demonstrates that any residual activity associated with either the helicase-deficient or the nuclease-deficient version of DNA2 is insufficient to support cell proliferation in human cells (**new Extended Data Fig. 9a-c**, and **e**).

In addition, we uncover an unexpected and detrimental gain-of-function linked to the expression of the nuclease-deficient version of DNA2. Thus, the nuclease-deficient DNA2 version causes strongly elevated RPA focus formation during S phase and an increase in RPA bodies in G2 compared to cells deprived of DNA2 (**new Extended Data Fig. 9c, d**). This suggests that the unleashed helicase activity of DNA2, which the Cejka lab have demonstrated is hyperactive in nuclease-dead versions of the protein *in vitro* (Levikova M, et al. (2013) Proc Natl Acad Sci U S A 110:E1992-2001. doi: 10.1073/pnas.1300390110; Pinto C et al. (2016) eLife 5:e18574. doi: 10.7554/eLife.18574), causes replication stress and ultimately gives rise to an even greater number of unresolved replication intermediates and dead-end HoRRer events as compared to those that occur when DNA2 is absent.

In the revised version of the MS, the data showing that both the nuclease and helicase activities of DNA2 are required to safeguard cells from dead-end HoRRer events are included in **new Extended Data Fig. 9** and described on **page 13, line 320ff.:**

*“As expected, expression of wild-type DNA2 in DIA-treated RPE-1 OsTIR1 DNA2^{dd} cells prevented the formation of RPA foci and rescued cell-cycle exit into senescence. In contrast, expressing nuclease-deficient (D277A), helicase-deficient (K654E), and double-deficient (D277A/K645E) versions of DNA2 did not restore proliferation, suppress RPA foci formation in G2, or cell-cycle exit in DNA2-deprived cells (**Extended Data Fig. 9**). Expression of DNA2 D277A caused strongly elevated RPA foci in S phase (**Extended Data Fig. 9d**), and an increase in G2 RPA foci compared to cells deprived of DNA2, likely reflecting uncontrolled DNA unwinding⁶⁹ in the absence of DNA2 nuclease activity. Thus, only the coordinated actions of the DNA2 nuclease and helicase activities can suppress an accumulation of stalled replication intermediates, HoRRer, and cell-cycle exit, providing a new explanation for why DNA2 is essential for cell proliferation.”*

Regarding opportunities for DNA2 inhibitors, our new complementation experiments show that expression of a combined nuclease- and helicase-deficient version of DNA2 is phenotypically equivalent to DNA2 loss (please see **new Extended**

Data Fig. 9). This is important, because it demonstrates that inactivated DNA2 – regardless of its physical presence – results in potentially exploitable DNA-loss phenotypes, supporting the rationale for DNA2 small-molecule inhibitor development. Our data further show that targeting either the DNA nuclease- or the helicase activity with small-molecule inhibitors will lead to DNA2-loss phenotypes. The finding that a nuclease-dead version of DNA2 is associated with a dominant-negative effect causing additional stress when compared to simple loss of DNA2 activity should be considered when deciding on an inhibitor strategy.

Please also see **reply to point 3** below for additional experiments addressing specific disease-linked DNA2 mutants and the implications for our molecular model for primordial dwarfism.

3. Strengthen the link between PD-related DNA2 deficiency and senescence

The authors provide good evidence for increased senescence in cells depleted for DNA2: increased p21 levels, increased Cyclin-B1 nuclear translocation/degradation and increased nuclear size, with nuclear size increase rescued by ATRi and p21 depletion. A few clarifications and some additional work would help further strengthen the relevance of this to PD-related DNA2 deficiency.

- I suspect this may be difficult (and perhaps impossible) due to antibody sensitivity, but is there any way in which IAA concentrations can be linked to approximate remaining levels of DNA2? Or at least determine a minimum level of DNA2 required for normal proliferation?

Authors' response: The consensus view that clearly emerges from the available case reports of *DNA2*-linked primordial dwarfism patients is that the levels of functional DNA2 are severely reduced; however, beyond this, there is currently no meaningful measure of what constitutes a critical DNA2 concentration that may determine normal or stunted growth in a human. Where DNA2 levels have been analysed in fibroblasts derived from patients with *DNA2*-linked primordial dwarfism, residual protein has been found reduced by half (compared to the levels found in cells from healthy individuals) (Shaheen R et al. (2014) *Genome Res* 24:291-9. doi: 10.1101/gr.160572.113) or undetectable (Di Lazzaro Filho R et al (2023) *J Med Genet* 60:1127-1132, doi:10.1136/jmg-2022-109119) by western blotting. This may reflect differences in the cell lines and/or detection, limitations which equally apply to our experimental set-up. Moreover, measurements in fibroblasts may not reflect critical DNA2 expression levels during development. We therefore believe it prudent to regard our partial degradation of DNA2 as a proof-of-principle experiment that shows that ever-diminishing levels of DNA2 result in ever increasing stochastic cell-cycle exit, and we do not attempt to determine a precisely quantified critical level of DNA2 in cells to avoid any impression this could be directly extrapolated in absolute quantitative terms to a whole-body patient situation. Nonetheless, we agree it is important to demonstrate partial depletion of DNA2 in RPE-1 *OsTIR1 DNA2^{dd}* cells titrated with IAA and have included a western blot analysis accordingly in the revised version of the MS (**new Extended Data Fig. 10a**). As expected, DNA2^{dd} diminishes with increasing IAA concentration. Satisfyingly, a fully penetrant senescence phenotype correlates with DNA2 dipping below the level of immunodetection. As described below, expression of

DNA2 patient variant T655A recapitulates stochastic cell-cycle exit, providing an example where cells expressing a pathophysiologically-relevant DNA2 hypomorph behave like partially DNA2-deprived cells. These findings strongly corroborate the notion of a critical level of DNA2 activity that allows sufficient residual proliferative potential for human development in DNA2-linked primordial dwarfism (please see response to subsequent point for more details).

- Is increased senescence detected in cells depleted for degron-tagged DNA2 and complemented with putative helicase- or nuclease-defective disease mutations?

Authors' response: We thank the referee for suggesting this experiment as a possibility to test our proposed model for the molecular pathology underpinning DNA2-linked primordial dwarfism with pathophysiologically-relevant DNA2 missense mutations identified in patients. While most primordial dwarfism-associated mutations in DNA2 result in diminished expression of wild-type DNA2 (as modelled by our partial depletion of DNA2), two disease-linked DNA2 variants with amino-acid substitutions have also been found: DNA2 K654E (Tarnauskaitė Ž et al (2019) Hum Mutat 40:1063-1070. doi: 10.1002/humu.23776) and DNA2 L48P (Di Lazzaro Filho R et al (2023) J Med Genet 60:1127-1132, doi:10.1136/jmg-2022-109119), described as putative helicase- and nuclease-defective variants. We have now conducted complementation experiments by expressing DNA2 T655A and DNA2 L48P in DNA2-deprived cells.

The L48P mutation was identified as a heterozygous missense variant of likely pathogenicity (c.143T>C) in combination with an intronic mutation shared by multiple patients with Rothmund-Thomson syndrome-like characteristics (Di Lazzaro Filho R et al (2023) J Med Genet 60:1127-1132, doi:10.1136/jmg-2022-109119). L48 is situated in proximity of the DNA2 nuclease domain; specifically in the OB fold preceding the nuclease active site affected by the D277A mutation described above in the reply to point 2). Based on modelling, the authors suggested potential steric clashes caused by L48P might affect the nuclease activity or overall stability of DNA2. We find no evidence for DNA2 L48P instability (please see **panel e** in **new Fig. 6** in the revised version of the MS). Expression of DNA2 L48P, however, cannot protect DIA-treated DNA2^{dd} cells from RPA-ssDNA accumulations and cell-cycle exit into senescence, thus phenocopying DNA2-loss (**new Fig. 6**, panels **f-h**). Moreover, in contrast to expression of nuclease-deficient DNA2 D277A (please see **new Extended Data Fig. 9**), DNA2 L48P expression does not result in elevated levels of RPA foci in S (**new Extended Data Fig. 10c**) or G2 phase (**new Fig. 6h**) as compared to DNA2 loss. The absence of this dominant-negative effect associated with nuclease-deficient DNA2 shows that DNA2 L48P is not a nuclease-deficient/helicase-proficient variant. The L48P mutation may affect function though impacting protein structure (as suggested by Di Lazzaro and co-workers) and we would suggest that there is little, if any, residual activity associated with DNA2 L48P. These data corroborate the pathogenicity of the c.143T>C allele and suggest that loss-of-function mutations can be found among the primordial dwarfism-associated DNA2 variants.

The DNA2 T655A mutation is adjacent to the helicase-inactivating K654E mutation described above. The respective patient mutation c.1963A>G was observed in combination with a splice-site mutation in the second DNA2 allele (intron 1; c.74+4A>C) from which an undefined level of residual wild-type DNA2 protein can be expressed (Tarnauskaitė Ž et al (2019) Hum Mutat 40:1063-1070. doi:

10.1002/humu.23776). Based on modelling, Tarnauskaitė and co-workers suggested that threonine 655 makes direct physical contact with ATP and that mutation T655A can be expected to negatively affect cellular DNA2 enzyme function. Upon expression of DNA2 T655A in DNA2-deprived cells, we now find a mild rescue of cell proliferation, supporting residual activity for this disease-linked mutant (please see **new panels e-h in new Fig. 6**). Consistently, we observe partial – rather than complete – cell-cycle exit into senescence in cell populations expressing DNA2 T655A. Strikingly, this recapitulates stochastic cell-cycle exit upon partial depletion of DNA2 (**see Fig. 6b**) with important biomedical implications: (1) The data provides important experimental support for the pathogenicity of c.1963A>G (DNA2 T655A) identified by Tarnauskaitė and co-workers; (2) The data recapitulates our model for DNA2-linked hypocellularity and primordial dwarfism using a DNA2 mutation found in an individual with primordial dwarfism. This provides important pathophysiologically-relevant evidence for our proposed mechanism where cells that reach a critical level of residual DNA2 activity either continue to proliferate or trigger a DNA2-loss phenotype and enter senescence, thus no longer contributing to growth. The new data is described on **page 15 (line 357ff.)** and **page 19 (line 468f.)** of the revised version of the MS:

*“While most of the reported primordial dwarfism-linked DNA2 mutations are intronic and reduce DNA2 levels through the expression of aberrant and mis-spliced DNA2 transcripts, one Seckel syndrome patient and one Rothmund-Thomson-related patient were found to be compound heterozygous for intronic mutations in trans with missense mutations resulting in variants DNA2 T655A and DNA2 L48P, respectively^{6,7}. T655 is located proximal to the ATP-binding site within the helicase-domain of DNA2, L48 within an oligonucleotide/oligosaccharide binding (OB) domain⁷⁰ which precedes the nuclease domain. We expressed the patient-derived DNA2 variants in DIA-treated RPE-1 OsTIR1 DNA2^{dd} cells (**Fig. 6e**). DNA2 T655A expression provided a mild suppression of the cell proliferation defect in DNA2-deprived cells (**Fig. 6f**) and recapitulated a partial cell-cycle exit phenotype similar to that observed after partial degradation of DNA2^{dd} (**Fig. 6g**). Concomitantly, DNA2 T655A-expressing cells exhibited a reduced number of G2 γ H2AX/RPA foci compared to DNA2-deprived cells 8 h after S phase, and cells that had transitioned to the next G1 phase were largely devoid of γ H2AX/RPA foci (**Fig. 6h**). In contrast, expression of DNA2 L48P did not rescue cell proliferation, cell-cycle exit, or γ H2AX/RPA foci formation in DNA2-deprived cells (**Fig. 6f-h**); while the OB domain of DNA2 folds against the nuclease domain, we did not find evidence of elevated RPA foci in S phase^{7,70}, indicating that the DNA2 L48P mutant does not behave like a nuclease-deficient/helicase-proficient DNA2 D277A mutant (**Extended Data Fig. 10c**). These data validate the pathogenicity of the DNA2 T655A and L48P patient mutations and indicate they result in partial and complete loss of DNA2 function, respectively.”*

“Similarly, expression of patient-derived variant DNA2 T655A resulted in stochastic cell-cycle exit.”

- Fig. 5d, representative of how many experiments?

Authors’ response: In the revised version of the MS, the data for cyclin B1 in DNA2-deprived cells exhibiting RPA-ssDNA accumulations is shown as mean values with

standard deviation from three independent experiments (please see **Fig. 5e**, corresponding to **Fig. 5d** in the previous version of the MS).

-Fig. 5f, same as for 5e (i.e. 1,000 cells for 3 independent experiments)? (Or is this 1,000 cells for one experiment or 3x 1,000 cells?) Fig. 5h, is this a single experiment?

Authors' response: We apologize for not being clear on this point in the figure legend to former **Fig. 5e, f**, now presented as **Fig. 6, panels a, b**. Former **Fig. 5e**, $n \geq 1000$ cells for each of three repeats; former **Fig. 5f**, $n \geq 1000$ cells for a representative example of three repeats per IAA concentration; **Fig. 5h**, $n \geq 60$ across two independent experiments. The figure legend for what is now **Fig. 6** in the revised version of the MS has been amended accordingly.

- Note that in support of increased senescence with DNA2 deficiency, this has been shown using primary fibroblasts from one PD patient with DNA2 mutations (ref 5).

Authors' response: We thank the referee for pointing this out and now refer to the fact that, in good agreement with our results, senescent cells have been observed among cultured fibroblasts derived from a primordial dwarfism patient with *DNA2* mutations (Shaheen R et al. (2014) *Genome Res* 24:291-9. doi: 10.1101/gr.160572.113). Now referred to in the text of the revised version of the MS on **page 14**, from **line 332ff.**:

“Case reports indicate low-level expression of wild-type DNA2 in DNA2-linked microcephalic primordial dwarfism, Seckel syndrome, and Rothmund-Thomson syndrome, and senescent cells have been observed among cultured patient fibroblasts.⁵⁻⁷”

4. Provide more detail for statistical analyses

- ED Fig. 1, were these experiments performed once? Can it be shown that the partial suppression of increased Pol delta synthesis in the *dna2-2 cdc27-D1* strains is statistically significant?

Authors' response: In **Extended Data Fig. 1**, we use Pu-Seq as a direct read-out of D-loop DNA synthesis efficiency by DNA polymerase δ and how wild-type levels are affected in *cdc27-D1* single-mutant cells. In **Fig. 1e**, we show that *cdc27-D1* abolishes the formation of *RTS1*-dependent *ade+* recombinants. This is consistent with data from budding yeast showing that efficient D-loop DNA synthesis requires the C-terminal domain of Cdc27 homolog Pol32 (Lydeard J R et al. (2007) *Nature* 448:820-823, doi:10.1038/nature06047); however, because this dependence on the Cdc27 C-terminal domain had not been formally demonstrated in fission yeast, we undertook the Pu-Seq experiments as an additional control. As expected, we find that D-loop DNA synthesis efficiency is down in the presence of C-terminally truncated Cdc27, demonstrating that a defect in recombination-dependent DNA synthesis underpins the loss of *ade+* recombinants at *RTS1* in *cdc27-D1* cells (**Fig. 1e**), and indicating that excessive *ade+* recombinants, which are suppressed in the presence of *cdc27-D1* (**Fig. 1e**), arise via HoRRer. The *cdc27-D1* data presented in **Extended Data Fig. 1** is a representative of two Pu-Seq runs; in parallel we have investigated a related C-terminal truncation mutant, *cdc27-D3* (Tanaka H et al. (2004) *Nucleic Acids Res.* 32:6367-77.

doi: 10.1093/nar/gkh963.) lacking the coding sequence for the Cdc27 C-terminal PCNA interaction site, which in budding yeast is regarded as most important for supporting D-loop synthesis. The *cdc27-D3* Pu-Seq tracks closely match the results for *cdc27-D1* presented in **Extended Data Fig. 1**. Importantly, a recent Whitby lab publication (Xu Y et al. (2025) Mol Cell 85:91-106.e5. doi: 10.1016/j.molcel.2024.10.026) has independently confirmed that *cdc27-D1* has a defect in recombination-dependent DNA synthesis using an assay monitoring BIR efficiency, which, like HoRReR, is reliant on DNA synthesis in the context of a D-loop. We now cite this new supporting data and have amended the MS text accordingly on **page 4, line 92ff.:**

*“Across organisms, recombination-dependent DNA synthesis requires DNA polymerase δ (Pol δ) subunit Cdc27 (Pol32 and POLD3 in budding yeast and human, respectively).⁴¹⁻⁴³ Thus, cells harbouring the truncated *cdc27-D1* (ref. 44) version of the essential *cdc27+* gene (**Fig. 1b**) are strongly defective for break-induced replication (BIR)⁴⁵, a mechanism which cells deploy to recover broken RFs by recombination-dependent DNA synthesis.⁴³ Analogously, we determined by polymerase usage sequencing (Pu-Seq)⁴⁶ that HoRReR-mediated DNA synthesis (where Pol δ synthesizes the Watson and Crick strands)⁴⁷ at *RTS1* was markedly reduced), in cells harbouring *cdc27-D1* (**Extended Data Fig. 1**); concomitantly, *ade+* recombinants were largely abolished (**Fig. 1e**).”*

- Show exact P-values for each comparison.

Authors' response: In the revised version of the MS, p values rather than stars are shown across all figures.

- Indicate statistical test performed in legends, as well as relevant n per sample and what these represent (be clear whether n is per experiment or total)

Authors' response: In the revised version of the MS, this information is now provided in each figure legend and has been removed from the Methods section for clarity.

- I may be wrong, but for Fig. 1f, is a t-test really the appropriate test?

Authors' response: We thank the reviewer for spotting this and have now re-blotted the data for enhanced clarity (showing all data points) and performed a Mann-Whitby *U* test to account for the non-normal distribution of cell-length data. This is now specified in the figure legend in the revised version of the MS. The conclusions drawn from **Fig. 1f** remain the same, *dna2-2* exhibits a highly significant cell-length increase over wild-type and there is no significant increase in cell length for *dna2-2 cdc27-D1* over *cdc27-D1*.

5. Other

- Re: ED Fig. 3b (line 170-174), can it really be concluded that “cells had fully completed bulk DNA replication prior to the induction of DNA2 degradation”

Authors' response: In this experiment, DIA was added when cells were already in very late S phase. Thus, DNA2 degradation was induced when bulk DNA synthesis had been completed. In the revised version of the MS, this experiment was replaced with a more complete analysis of the relationship between the absence of DNA2 for increasing periods of S phase and the formation of HoRReR-dependent RPA-ssDNA

accumulations in G2. Our findings indicate that while DNA2 promotes the recovery of stalled forks during S phase, its actions are required up until very late S/G2 phase to limit persistent replication intermediates with the potential to undergo HoRRer to an extent that allows the majority of cells to progress to mitosis and into the next G1. These data are presented in new **Extended Data Fig. 7** and referred to in the text on **page 10, line 227ff.:**

*“When we treated RPE-1 OsTIR1 DNA2^{dd} cells with DIA at varying timepoints before late S-phase, we observed that the number of FANCD2 and RPA foci persisting 8 h after late S phase correlated with the proportion of S phase that cells had traversed after the induction of DNA2^{dd} degradation. Thus, if DNA2 degradation was induced prior to S phase, cells exhibited the expected level of RPA-ssDNA accumulations (median 12 if DIA was added 12 h before late S phase). When DNA2^{dd} was degraded in the first half of S phase, RPA-ssDNA accumulations were moderately reduced (median 11 if DIA was added 8 h before late S phase) and steadily declined as cells traversed less of S phase without DNA2^{dd}. If DIA was added 2 h before late S phase, the median of RPA-ssDNA foci per G2 cell was reduced to 3 and over 70% of cells progressed to the next G1 phase (**Extended Data Fig. 7**). These data show that DNA2 is required throughout S phase to avoid a build-up of stalled replication intermediates accumulating RPA-ssDNA in G2 phase of the cell cycle.”*

- In Fig. 5i and ED Fig. 5c, micronuclei per cell are plotted. However, increased number of MN per cell can indicate a small number of cells with many MN, or more cells with (one or a few) MN. It is therefore more common to quantify and plot % cells with MN, and probably better to do so here as well.

Authors' response: We thank the reviewer for the suggestion and have re-plotted the data in **Fig. 5i** (now presented in **Extended Data Fig. 10b**) and **Extended Data Fig. 5c** (now **Extended Data Fig. 8c**) in the revised version of the MS.

- It would be helpful to add to the legend of ED Fig. 1 that all strains were dna2-2.

Authors' response: The genotypes in **Extended Data Fig. 1** are correctly annotated (please also see reply to **point 4, questions 1** above).

Referee #3:

In the manuscript, “DNA2 enables proliferation by restricting homologous recombination-restarted replication”, Ulrich Rass and colleagues investigate the essential function of the nuclease-helicase DNA2. DNA2 is set apart from most DNA repair proteins by its essential nature suggesting the protein will take care of a problem that is fundamental and encountered during every cell cycle. In this manuscript, the essential function of DNA2 is investigated in fission yeast and human cell lines. Using the fission yeast RTS1 replication barrier systems the authors implicate DNA2 in recombination-dependent replication restart of stalled replication forks (HoRRer). They link this function to the essential function of DNA2, as DNA2 phenotypes were found to be suppressed by mutation of the CDC27 subunit of polymerase delta, which is critical for recombination-dependent replication restart. They then move on to experiments with non-cancerous human cell-lines, where they show that targeted degradation of DNA2 leads (i) to accumulation of RAD51-dependent replication intermediates with (ii) exposed single-stranded DNA (ssDNA) which in turn leads to (iii) ATR-dependent checkpoint hyperactivation and (iv) cell cycle exit and senescence. Lastly, they explore whether reduced levels of DNA2, as seen in genetic disease such as DNA2-linked microcephalic primordial dwarfism, triggers a similar response and find that it sporadically causes checkpoint hyperactivation and cell cycle exit, potentially explaining patient phenotypes.

This is a well-designed study with very interesting results, which will have strong impact on the field and as such would profit greatly from being published in a journal such as Nature. Currently, there are still a number of points which need support by additional experiments and also the manuscript needs improvement in some places.

Major Points:

#1 – Fig. 1 focuses on fission yeast and shows an increase in recombination-dependent replication using the RTS1-system. While this system is elegant, this part of the paper should be better connected to the rest of the paper. A major result of Fig. 1 is that a mutation in the polymerase delta subunit *cdc27* suppresses the *dna2-2* mutant phenotype. I therefore suggest testing for suppression of the DNA2-degron phenotype by POLD2 knockdown in the human cell culture experiments.

Authors' response: We thank **referee 3** for the suggestion and have now conducted an siRNA-mediated depletion of POLD3 in DNA2-deprived cells. Consistent with our findings of elevated Cdc27-dependent HoRRer at stalled replication forks in fission yeast (**Fig. 1**), we find that POLD3 depletion significantly suppresses RPA foci intensity and EdU incorporation at stalled replication intermediates in DIA-treated RPE-1 *OsTIR1 DNA2^{dd}* cells (please see **new panel h in revised Fig. 4**). This provides strong additional evidence that that excessive HoRRer occurs in fission and human cells upon DNA2 dysfunction. To better connect the findings in fission yeast and human cells, we now introduce the dependence of recombination-dependent DNA synthesis on POLD3 homologs across organisms on **page 4 (line 92ff.)** and describe the POLD3-depletion experiment on **page 11 (line 265ff.)** as follows:

“Across organisms, recombination-dependent DNA synthesis requires DNA polymerase δ (Pol δ) subunit Cdc27 (Pol32 and POLD3 in budding yeast and human, respectively).⁴¹⁻⁴³ Thus, cells harbouring the truncated *cdc27-D1* (ref. 44) version of the essential *cdc27+* gene (**Fig. 1b**) are strongly defective for break-induced replication (BIR)⁴⁵, a mechanism which cells deploy to recover broken RFs by recombination-dependent DNA synthesis.⁴³ Analogously, we determined by polymerase usage sequencing (Pu-Seq)⁴⁶ that HoRRer-mediated DNA synthesis (where Pol δ synthesizes the Watson and Crick strands)⁴⁷ at *RTS1* was markedly reduced), in cells harbouring *cdc27-D1* (**Extended Data Fig. 1**); concomitantly, *ade+* recombinants were largely abolished (**Fig. 1e**).”

“(…) DNA synthesis at RPA foci in DNA2-deprived cells was strongly suppressed by siRNA-mediated depletion of Pol δ subunit POLD3 on which D-loop DNA synthesis depends⁴¹⁻⁴³ (**Fig. 4h**).

Taken together, these data indicate a shift in RF recovery towards HoRRer when DNA2-dependent fork processing and reactivation²⁸ is not available.”

#2 – Fig. 2: DNA2 degradation needs better characterization, including quantification and a more detailed time-course.

Authors’ response: We have replaced the western blot showing the degradation of DNA2^{dd} in DIA-treated RPE-1 *OSTIR1 DNA2^{dd}* cells to better demonstrate the timeframe for total loss of DNA2^{dd} by adding additional time-points into the analysis (figure legend and methods amended accordingly). The blot shows that while protein levels drop immediately after DIA addition, approx. 40% of DNA2^{dd} compared to t=0 (untreated cells) is retained 2 h after DIA addition. Importantly, 4 h after DIA addition DNA2^{dd} levels have fallen below the detection limit. We use this 4 h timeframe as a reference point for DNA2 loss after DIA addition to inform our experimental set-ups as indicated in the work-flow diagrams (please also see **response to Minor Point #1** below).

#3 – Fig. 2e shows only normalized cell proliferation. Please also show non-normalized cell proliferation to allow better judgement of effect sizes.

Authors’ response: We appreciate the data in **Fig. 1e** was slightly convoluted and determining the level to which ATRi rescues cell proliferation in **Fig. 2e** required a comparison to **Fig. 2c** in the original version of the MS. For an improved representation of effect size, we have now re-plotted the data without the normalisation of cell number to DIA-only treated cells. For example, DIA-treated cells at 0 μ M ATRi show an average cell count of 5281 on day three compared to an average of 30139 cells when left untreated (i.e, approx. 20%, similar to **Fig. 2c**). This rises to 11694 cells on average (approx. 40%) at an ATRi dose of 1 μ M (before ATRi itself becomes toxic to cells at higher doses), in line with an additional cell division as observed by live cell imaging upon ATRi treatment in **Fig. 2f**. We agree this is perhaps the best way to present this data, and the representations is now consistent with the way proliferation data is presented throughout the MS.

#4 – Fig. 4e shows that RAD51 inhibition partially suppresses the ssDNA-accumulation phenotype observed upon DNA2 depletion. Given that the effect size is moderate, I would suggest to also test genetic knock-down of RAD51 in this context.

Authors' response: RAD51 fulfils multiple enzymatic and non-enzymatic functions in interphase to mediate DNA replication fork protection and recovery. Accordingly, the physical absence of RAD51 results in fork degradation and DNA gaps in replicating DNA (e.g., Schlacher K et al. (2012) *Cancer Cell* 22:106-16. doi: 10.1016/j.ccr.2012.05.015; Kolinjivadi AM (2017) *Mol Cell* 2017 67:867-881.e7. doi: 10.1016/j.molcel.2017.07.001). Consistent with DNA damage arising from the absence of RAD51, RAD51 depletion by siRNA (as previously described in Mocanu C et al. (2022) *Cell Rep* 39:110701. doi: 10.1016/j.celrep.2022.110701) in RPE-1 *OsTIR1 DNA2^{dd}* cells resulted in RPA foci formation and profound changes in the cell-cycle distribution of cells (<5% cells in S phase 48 h into siRNA-mediated RAD51 depletion) independently of DIA-mediated degradation of DNA2. A similar depletion of the S phase pool in a DIA-independent manner is also observed at higher doses of RAD51 inhibitor B02. We therefore titrated B02 to determine a dose with no overt effect on cell cycle distribution (6.25 μ M; please see **Figure B** below), which we then used to perform the experiment in **Fig. 4e** (now shown as **Fig. 4g** in the revised version of the MS). Thus, the low-dose small molecule inhibitor approach allows circumventing inherent caveats associated with complete loss of RAD51 or high inhibitor doses and offers the most reliable method for analysing the effect of RAD51 on DIA-induced RPA foci in RPE-1 *OsTIR1 DNA2^{dd}* cells. The partial reduction in RPA-ssDNA observed at the B02 dose employed indicate that RAD51 inhibition may not be complete, yet sufficiently strong to observe a significant reduction in RPA-ssDNA. Importantly, our new results showing a strong suppression of the ssDNA-accumulation phenotype in DNA2-deprived cells upon POLD3 depletion (please see **response to Major Point #1** above) independently support that the observed RPA-ssDNA accumulations arise by HoRReR at stalled replication forks.

Figure B: The effect of RAD51 inhibitor B02 on the cell cycle distribution. Flow cytometric analysis with PI staining of RPE-1 *OsTIR1 DNA2^{dd}* cells treated with the indicated doses of B02 for 18 h in the presence or absence of DIA. Please note that B02 doses of 6.25 μ M or below have no significant effect on cell-cycle distribution compared to control.

#5 – In general, the manuscript would profit from a comparative analysis of the two rescue conditions (HR-inhibition vs checkpoint inhibition). Moreover, the authors had also previously shown *dna2* rescue by checkpoint inhibition in budding yeast, but these results are hardly discussed in the current paper. I think the paper would overall profit

from a more detailed discussion of similarities and differences of DNA2 phenotypes in different organisms.

Authors' response: We agree that expanding on previous results in budding yeast underpinning to the hypothesis of elevated HoRReR as a source of checkpoint signalling in Dna2-mutant cells is beneficial. Similarly, an enhanced discussion of rescue conditions of Dna2/DNA2 dysfunction in yeast and human helps to showcase the striking conservation of the essential function of DNA2 in limiting recombination-dependent fork restart across eukaryotes. In the revised version of the MS, we now lead into the Discussion (para 1) with a brief description of our previous findings in budding yeast and have extended para 2 to highlight that interference with checkpoint signalling or HoRReR can circumvent senescence and effect mitotic progression in DNA2-deprived human cells. The first two paragraphs of the Discussion in the revised version of the MS now reads (**page 16, line 386ff.**):

“Our data reveal the essential role of DNA2 in eukaryotes. Previously, we have shown that Dna2 dysfunction in budding yeast exposes cells to a terminal G2/M checkpoint arrest after transient RF arrest.^{39,40} The block to M phase entry was alleviated by disruption of the DNA damage checkpoint or mutations compromising recombination-dependent DNA synthesis; in either case, cells progressed into mitosis with incompletely replicated chromosomes.^{39,40} These findings led us to propose that elevated HoRReR in lieu of Dna2-mediated RF recovery results in a fatal checkpoint response triggered by recombination-dependent fork restart intermediates.³⁸ Here, we show that Dna2 dysfunction results in a dramatic rise of HoRReR-induced direct repeat recombination at the site-specific replication barrier RTS1 in fission yeast, demonstrating that the actions of Dna2 restrict recombination at stalled RFs. Thus, Dna2-dependent fork processing is indispensable to prevent the accumulation of recombinogenic chicken foot structures, which is consistent with our previous findings of elevated fork reversal in fission yeast upon loss of Dna2.² We further show that genetic perturbation of recombination-dependent D-loop DNA synthesis rescues excessive recombination, checkpoint activation, and cell-cycle arrest associated with Dna2 dysfunction. Thus, Dna2 dysfunction causes inviability by durable G2/M checkpoint arrest in distantly related yeast species through unrestricted HoRReR.

In DNA2-deprived human cells, we found persistent replication intermediates marked by γ H2AX and FANCD2. This is consistent with increased fork reversal in DNA2-depleted human cells²⁸, but markedly different from previous reports^{4,8,19,28,49,50} of rampant DNA damage associated with gradual depletion of DNA2. Importantly, replication intermediates that accumulate upon DNA2 degradation engaged in RAD51 and POLD3-dependent DNA synthesis, leading to substantial RPA-ssDNA in G2. The accumulation of RPA-ssDNA was invariably associated with a checkpoint-dependent cell-cycle exit from G2 into senescence, a phenotype akin to the sudden and permanent G2/M arrest upon induction of Dna2-deficiency in yeast^{30,71}. Disruption of checkpoint signalling by ATR inhibition or p21 depletion in DNA2-deprived cells enabled mitotic progression, albeit at the expense of micronuclei formation indicative of unfaithful chromosome replication in the absence of DNA2. Similarly, interference with RF reversal, a prerequisite for HoRReR, by depletion of FBH1, diminished RPA-ssDNA accumulations upon DNA2 loss and allowed a subset of cells to progress through mitosis. These striking parallels between the yeast and human systems support a unified model where DNA2-

dependent fork processing plays a critical role in RF recovery and acts as an essential suppressor of HoRReR-induced cessation of cell-cycle progression across eukaryotes (Fig. 6)."

#6 - Fig. 5e shows a dose-response curve for proliferation over IAA concentrations. The authors need to show information on DNA2 levels from the same experiment and show whether there is correlation of DNA2 levels and phenotype.

Authors' response: In the revised version of the MS, we have included a western blot analysis (**Extended Data Fig. 10a**) of the dose-response curve for proliferation over IAA concentration on RPE-1 *OSTIR1 DNA2^{dd}* cells (formerly **Fig. 5e**, presented as **Fig. 6a** in the revised MS). As expected, DNA2 remains detectable at 0.08 and 0.2 μ M IAA, concentrations associated with no or mild cell-cycle exit phenotypes. From 0.7 μ M IAA, DNA2 becomes undetectable, and cells start showing a near-complete cell-cycle exit phenotype. These data confirm a correlation between diminishing DNA2 levels and cell-cycle exit.

#7 – In the same vein, the authors claim that their DNA2-degron model mimics patient scenarios, but they do not show in the paper that DNA2 protein levels are actually similar.

Authors' response: Case reports of *DNA2*-linked primordial dwarfism show that patients harbour intronic splice-site mutations, partial gene deletions, or missense mutations leading to amino acid changes in *DNA2*. Larger gene deletions and missense mutations always occur *in trans* with a splice-site mutations allowing for expression of residual wild-type *DNA2*. Thus, while all case reports agree that *DNA2* levels are significantly reduced in patients, cells must be able to express residual levels of *DNA2* consistent with the essential requirement for *DNA2* for cell proliferation. However, there is not sufficient data available from patient-derived cells to determine in precise quantitative terms what constitutes a critical pathological level of *DNA2* in cells. In fact, the analysis of *DNA2* in patient-derived fibroblasts by western blotting has yielded varying results from an estimated 50% reduction to *DNA2* being undetectable (Shaheen R et al. (2014) *Genome Res* 24:291-9. doi: 10.1101/gr.160572.113; Di Lazzaro Filho R et al (2023) *J Med Genet* 60:1127-1132, doi:10.1136/jmg-2022-109119). This could relate to differences in the patient and control cell lines and/or method of detection. Perhaps more importantly, measurements in fibroblasts may not provide an adequate proxy of critical *DNA2* expression levels on an organismal level during development. Accordingly, we do not attempt to relate diminished *DNA2* levels upon titration of IAA to *DNA2* levels observed in patient cell lines in quantitative terms. Nonetheless, our partial degradation of *DNA2* provides proof-of-principle that lower *DNA2* levels, as observed and/or expected from all identified compound *DNA2* mutations in patients, result in stochastic cell-cycle exit.

Following suggestions of **referee 2**, we also provide new experimental data on the effect of pathophysiologically-relevant *DNA2* missense mutations identified in primordial dwarfism patients in the revised version of the MS. Interestingly, *DNA2* K654E (Tarnauskaitė Ž et al (2019) *Hum Mutat* 40:1063-1070. doi: 10.1002/humu.23776) provided a moderate rescue of cell proliferation upon expression in *DNA2*-deprived RPE-1 *OsTIR1 DNA2^{dd}* cells. Recapitulating the stochastic cell-cycle exit phenotype observed upon partial degradation of *DNA2*, *DNA2* K654E expression

resulted in a dichotomy of phenotype where cells either produce RPA-ssDNA accumulations and enter senescence or continue into mitosis. Regardless of the precise residual activity level inherent to DNA2 K654E, these findings strongly corroborate our model that a critical level of DNA2 activity exists which allows sufficient residual proliferative potential for human development while DNA2 levels below this threshold invariably lead to senescence and do not support life. The data is presented in **new Fig. 6e-h** and described on **page 15** of the revised version of the MS (**line 364ff.**):

“We expressed the patient-derived DNA2 variants in DIA-treated RPE-1 OsTIR1 DNA2^{dd} cells (Fig. 6e). DNA2 T655A expression provided a mild suppression of the cell proliferation defect in DNA2-deprived cells (Fig. 6f) and recapitulated a partial cell-cycle exit phenotype similar to that observed after partial degradation of DNA2^{dd} (Fig. 6g). Concomitantly, DNA2 T655A-expressing cells exhibited a reduced number of G2 γ H2AX/RPA foci compared to DNA2-deprived cells 8 h after S phase, and cells that had transitioned to the next G1 phase were largely devoid of γ H2AX/RPA foci (Fig. 6h).”

#8 – The paper would profit from a discussion of similarities and differences between the recombination-dependent replication described here and MIDAS.

Authors’ response: MiDAS (mitotic DNA synthesis) is defined as a form of recombination-dependent DNA synthesis that is only activated after cells enter mitosis. MiDAS can be observed under specific experimental conditions in cells exposed to replication stress, where unresolved replication intermediates that are carried over into mitosis collapse to form DNA double-strand breaks; these breaks give rise to break-induced replication (BIR) (Minocherhomji S et al. (2015) Nature528:286-90. doi: 10.1038/nature16139). Because the HoRRer events we describe in the current MS occur in interphase and the finding that affected cells never reach mitosis, we had not discussed MiDAS specifically in the previous version of the MS. However, we agree that MiDAS is a topical variation of BIR, and we have now expanded the text where we distinguish HoRRer from BIR by showing that recombination-dependent RPA-ssDNA formation in DNA2-deprived cells occurs at stalled replication intermediates independently of DNA double-strand break formation to refer to MiDAS. Introducing MiDAS as a mitosis-specific BIR-type reaction helps clarify that the HoRRer events causing cell cycle exit from G2 in DNA2-deprived cells are distinct from MiDAS events, both on account of the different cell-cycle phases in which they occur, and the respective recombination-dependent DNA synthesis subways employed. The text on **page 13 (line 308ff.)** of the revised version of the MS has been amended accordingly to read:

“Because RPA-ssDNA accumulation in a RAD51/POLD3-dependent manner (see Fig. 4) could be compatible with either HoRRer at stalled RFs or BIR, which mediates recombination-dependent DNA synthesis after fork breakage and DNA double-strand break formation in interphase or in mitosis (known as mitotic DNA synthesis, or MiDAS)⁶⁶, we probed for DNA double-strand break-specific phosphorylation of RPA32 on S4 and S8 (pRPA)^{67,68}. Only a small subset of cells (approx. 5% of RPE-1 OsTIR1 DNA2^{dd} cells 18 h after DIA addition) exhibited pRPA foci, and these were only detected in cells in which cyclin B1 had translocated into the nucleus or been degraded (Extended Data Fig. 8d).

This shows that RAD51/POLD3-dependent RPA-ssDNA formation and the commitment to cell-cycle exit is initiated at stalled replication intermediates in a manner that is independent of DNA double-strand break formation and consistent with HoRRer.”

Minor Points:

#1 - The authors use graded shading of “DNA2 degradation” in several figure legends, but it is unclear whether (i) this corresponds to actual levels and (ii) it is not adjusted to the changing time scales in the experiment.

Authors’ response: In the revised version of the MS, we have redrawn all workflow diagrams and, in each, indicate the 4-hr timepoint at which DNA2 becomes undetectable (as determined by western blot in **new panel a of Fig. 2**), at the correct scale.

#2 – Abstract, “Three decades after DNA2 was first described...”: I would argue that the historical perspective is not sufficiently important for the paper to be featured this prominently in the abstract.

Authors’ response: We have removed this half-sentence from the revised version of the MS.

#3 – Sentence line 96 ff.: This sentence seems to have unnecessarily complicated grammar.

Authors’ response: We have amended the text in the revised version of the MS to read:

“Combining dna2-2 with cdc27-D1 reversed the dramatic increase in RTS1-dependent ade⁺ recombinants associated with Dna2 dysfunction, confirming that HoRRer underpins excessive ade⁺ recombinants in Dna2-defective cells (Fig. 1e).”

Response to Referees Nature MS 2024-10-21223

Hudson et al.: “DNA2 enables growth by restricting homologous recombination-restarted replication”

Our sincere thanks to all three referees for their support and suggestions throughout this review process. We are grateful for their positive assessment of our revised MS. Below, we address the final points raised.

Referee #1:

I read the revised version of the manuscript by Hudson, Appanah and colleagues, and I am very pleased by the additional work done to arrange comments and suggestions from the two other reviewers and myself. In my view, the additional experiments using DNA2 mutants, depletion of fork reversal enzymes and careful analysis of phenotypes improved the manuscript and made the authors' assumptions more solid. While I feel that an attempt to directly evaluate the relevance of fork collapse post-fork reversal in the initiation of HR would have helped authors' conclusions, I am very convinced by the claims and found the revised manuscript improved.

The revised version of the manuscript retained its logical writing and accuracy in reporting methods and statistical details. I noticed also that authors improved presentation of the statistical details in figures vis-a-vis specific comments from another reviewer.

I have no additional comments but I would suggest authors to include in the abstract some mention to the results from DNA2 disease mutants and also to mitigate the statement about the growth failure in DNA2-linked dwarfism as, even if of high relevance, authors' findings still do not provide a solid molecular foundation to this clinical phenotype.

Similarly, I would suggest changing the sentence in the discussion that refers to the same link in a way that delivers a potential explanation rather than an established connection.

Authors' response: We thank **referee 1** for the suggestion to directly relate to our new experiments showing that the expression of patient-derived DNA2 variants results in stochastic cell-cycle exit in the Abstract. This has now been implemented, and we refer to our results as suggesting a *conceptual framework* to rationalize the link between DNA2 mutations and primordial dwarfism:

“Stochastic entry into senescence stifles the proliferative potential of cells following expression of a Seckel syndrome patient-derived DNA2 hypomorph or partial degradation of DNA2, providing a conceptual framework to explain global growth failure in DNA2-linked primordial dwarfism disorders.”

In the Discussion section we have changed the wording from “*compelling rationale*” to “*compelling model*” where we discuss these results in the context of human disease.

Referee #2:

My compliments to the authors for thoroughly addressing all reviewers' comments. This is a very nice piece of work that will be of interest to Nature readers.

There is one relatively small thing I feel would be helpful to add. Although I'm absolutely convinced that it is indeed possible to identify late S-phase cells using EdU incorporation patterns, it is still not entirely clear to me from ref. 54 and the rebuttal what the exact pattern is supposed to be. Is it simply those cells that display EdU foci (and not pan-nuclear EdU staining, or absence of staining), or is the number of EdU foci also important? And to what extent is DAPI content taken into account?

Authors' response: EdU incorporation patterns in cells at different stages of the cell cycle are well established and can be seen in Fig. 2F of former ref. 54 (now ref. 47). Late S-phase EdU incorporation is easily distinguished from the typical pan-nuclear EdU incorporation pattern exhibited if cells receive continuous EdU exposure at any time during S phase (early/mid) prior to their passage through late S-phase. A late EdU incorporation pattern is best described as being patchy (rather than focal) and often peripheral, with signals emanating from late-replicating and heterochromatic DNA. Examples of this in our MS can be seen in **Fig. 3b** and **Fig. 5d** (note the much more strongly EdU-labelled mid-S phase cell on the left boundary of **Fig. 5d** for comparison). DAPI measurements are not required to identify late S-phase cells by EdU incorporation patterns in this experimental set-up.

Therefore, can the authors please: 1) clearly define (in their methods section) how they decided on which cells were late S-phase and which ones not; and potentially 2) provide example images of cells with late S-phase EdU incorporation patterns (the latter may not be necessary if the description is clear enough and it is simply the presence of any number of EdU foci in cells that have 4N DNA content).

Authors' response: We have included the following explanation in the **Methods** section on **Immunofluorescence microscopy** and refer to the example of a late S-phase EdU incorporation that can be seen in **Fig. 3d**:

*“To identify DNA2-deprived cells at different times after traversing late S phase, we treated non-synchronous OsTIR1 DNA2^{dd} cells with DIA for 14 h before adding EdU to the culture medium to continuously label replicating DNA. 2, 4, and 8 h later, cells were fixed and only cells with a typical patchy and peripheral late S phase EdU incorporation pattern⁴⁷ (as seen in cells marked with arrowheads in **Fig. 3d**) were selected (i.e., cells that had been in late S phase 2, 4, and 8 h earlier).”*

Referee #3:

I congratulate Rass and colleagues to the revised version of their manuscript “DNA2 enables proliferation by restricting homologous recombination-restarted replication”, which has addressed all points raised in my previous review. I think it is suitable for publication in Nature.

Authors' response: We thank **referee 3** for their kind words.